# Is Attention All You Need for Temporal Link Prediction? A Lightweight Alternative via Learnable Positional Encoding and MLPs

## Abstract

Link prediction is of key importance in many real-world applications like social network analysis and recommender systems. To leverage the expressive power for achieving SOTA performance, many recent works adapt the attention mechanism to the structured data for link prediction, in which dense or relational attention is often unaffordable on large-scale structured data. Moreover, in a realistic setting, the time-evolving topological and feature information can raise more challenging questions about the efficiency and effectiveness of attention mechanisms. In spite of the expressive power, we discern that the attention mechanism may not always be as irreplaceable as expected for temporal graph representation learning, at least not for temporal link prediction tasks. Formally, we discover that some deliberately-designed simple positional encoding can enable MLPs to exploit attributed graph information to achieve SOTA performance than complex graph transformers. Hence, we propose a simple temporal link prediction model, named SimpleTLP. In detail, for SimpleTLP, we first propose to adapt Fourier Transform on temporal graphs for learning informative positional encoding, then we (1) prove this learning scheme can make positional encoding preserve the temporal graph topology from the spatial-temporal spectral viewpoint, (2) verify MLPs can fully exploit the expressiveness and reach and even surpass Transformers on that encoding, (3) change different initial positional encoding inputs to show robustness, (4) analyze the theoretical complexity and obtain less empirical running time than SOTA baselines, and (5) demonstrate its temporal link prediction out-performance in a comprehensive way on 13 classic datasets and with 10 algorithms in both transductive and inductive settings using 3 different sampling strategies. Also, SimpleTLP obtains the leading performance in the large-scale TGB benchmark (the newest TGB 2.0).

## 1 Introduction

Link prediction is an important research topic in the graph learning community (Kumar et al., 2020), especially in complex temporal networks (Lu & Zhou, 2010; Martínez et al., 2017). Learning from the time-evolving topological structures and time-evolving node and edge features, and determining whether the link (i.e., edge or interaction between two nodes) exists at a certain timestamp are challenging but have attracted much research interest like dynamic protein-protein interactions (Taylor et al., 2009; Gupta et al., 2015), recommender systems (Huang et al., 2005), social network analysis (Daud et al., 2020), and many more.

Recently, with the success of the attention mechanism (Vaswani et al., 2017), graph transformers have been proposed for the structured data to obtain the out-performance in many graph applications for their powerful expressiveness, like GraphGPS (Rampásek et al., 2022) for graph classification, TokenGT (Kim et al., 2022) for node and graph classification, and DyGFormer (Yu et al., 2023) for link prediction. However, to preserve the long-distance relation into the representation vectors, graph transformers usually need high time complexity due to the dense attention or relational attention (Müller et al., 2023; Min et al., 2022), i.e., each pair of nodes need to be attended with $O(n^2)$ time complexity, which can even approach $O(n^3)$ in a more complex relational setting (Diao

& Loynd, 2023). A detailed theoretical time complexity and empirical running time comparison with SOTAs can be found in Appendix E.2 and Appendix E.3.

The above complexity analysis places the burden on the temporal link prediction task from at least two aspects (Kazemi et al., 2020). First, to make the current link prediction, the decision models need to consider the interaction history, and the corresponding spatial-temporal structure will force the attention mechanism to become more complex to attend to more interactions. Second, besides the spatial-temporal structure, the node and edge features also evolve over time, and the prolonged features also increase the size of neural architectures. Promisingly, some nascent work (Cong et al., 2023) proposed to only use Multilayer Perceptrons (MLPs) to accomplish the temporal link prediction. Although (Cong et al., 2023) does not reach the superior performance of the graph transformers (Yu et al., 2023), it paves the way for exploring the ability of MLPs in the temporal link prediction task.

In light of the above discussion, an intriguing question emerges: instead of a pre-defined fixed positional encoding, **can we design a learnable positional encoding along with time such that a simple neural architecture can achieve a similar or even better performance than graph transformers?** Thus, in this paper, we design a simple temporal link prediction model, named SimpleTLP. In SimpleTLP, we propose two novel techniques: Learnable Positional Encoding Module (LPE) and Node-Link-Positional Encoder based on Discrete Fourier Transform.

With those proposed techniques, our goal is that even using MLPs can exploit temporal graph information and achieve competitive performance compared with graph transformers in the temporal link prediction task. To demonstrate this, we first prove that the proposed positional encoding method in SimpleTLP can preserve the temporal graph topology from the spatial-temporal spectral viewpoint and the corresponding theoretical time complexity, as shown in Section 4. Moreover, in Section 5, we evaluate the empirical performance of SimpleTLP extensively, including (1) the comprehensive effectiveness comparison with 13 classic datasets, 10 algorithms, 2 learning settings (transductive and inductive), and 3 sampling strategies, where our SimpleTLP performs the best across the board; (2) the verification of the role of MLPs and Transformers upon SimpleTLP; (3) the robustness of SimpleTLP in terms of different initial positional encoding inputs; (4) various parameter analysis and ablation studies; and (5) the leading performance in the large-scale TGB open benchmark (Gastinger et al., 2024).

## 2 PRELIMINARIES

Here, we introduce preliminaries to pave the way for deriving our SimpleTLP in the next section.

**Temporal Graph Snapshot**. A snapshot of a temporal graph $G$ is a collection of temporal interactions (i.e., edges) at time $t \in \{1, \ldots, T\}$, which is defined as $G^t = \{(u, v, t) | u \in V^t; v \in V^t; (u, v, t) \in E^t\}$. Each event $(u, v, t)$ denotes an interaction between nodes $u$ and $v$ that occurs at time $t$, and the node and edge sets of $G^t$ are denoted as $V^t$ and $E^t$ respectively. Additionally, for node $u \in V^t$, we denote the node features as $\mathbf{s}_u^t \in \mathbb{R}^{d_N}$, and the edge features associated with an event $(u, v, t)$ as $\mathbf{e}_{u,v}^t \in \mathbb{R}^{d_E}$, where $d_N$ and $d_E$ are the dimensions of the node and edge features.

**Temporal Neighbors**. The temporal neighborhood of a node $u \in V^t$ at time $t \in \{1, \ldots, T\}$, denoted as $\mathcal{N}_u^t$, is the set of all interactions that involve $u$ at time $t$. Formally, $\mathcal{N}_u^t$ can be mathematically expressed as $\mathcal{N}_u^t = \{(u, v, t) | v \in V^t; (u, v, t) \in E^t\}$.

**Time Encoder**. A time encoder is a function to obtain the vector representation of time $t$ (Xu et al., 2020; Wang et al., 2021b; Cong et al., 2023). Let $d_T$ be the dimension of the time encoding vector. We define our time encoding function as $f_T : \mathbb{R}^+ \to \mathbb{R}^{d_T}$. In SimpleTLP, for $t \in \mathbb{R}^+$, $f_T(t) = \cos(t \cdot \omega)$, where $\omega \in \mathbb{R}^{d_T}$ with the $i^{\text{th}}$ entry computed as $\omega_i = \alpha^{-(i-1)/\beta}$, and $\alpha, \beta \in \mathbb{R}$ are hyper-parameters. In order to enhance the ability of $d_T$ in distinguishing all timestamps, given that $t_{max}$ is the maximum timestamp being considered, $\alpha$ and $\beta$ should be selected in such a way that $t_{max} \cdot \alpha^{-(i-1)/\beta} \to 0$ as $i$ goes towards $d_T$. Moreover, $\alpha, \beta$, and $\omega$ are fixed during the training process of SimpleTLP.

**Positional Encoding for Graphs**. Originated from the transformer (Vaswani et al., 2017), positional encoding is proposed to add node feature information to support the attention mechanism by indicating the relative position of a node within the input graph (Dwivedi et al., 2023), and the common positional encoding methods include computing the Laplacian eigenvector or Personalized PageRannk vector for each node (Rampásek et al., 2022; Chen et al., 2023). Mathematically, given an $n$-node (static)

graph $G = (V, E)$ and its adjacency matrix $\mathbf{A} \in \mathbb{R}^{n \times n}$, the normalized Laplacian of $G$ is computed as $\mathbf{L} = \mathbf{I}_n - \mathbf{D}^{-1/2} \mathbf{A} \mathbf{D}^{-1/2}$, where $\mathbf{I}_n$ is the $n \times n$ identity matrix, and $\mathbf{D}$ is the diagonal degree matrix of $G$. Let the eigen-decomposition of $\mathbf{L}$ be $\mathbf{L} = \mathbf{U} \mathbf{\Lambda} \mathbf{U}^\top$, where $\mathbf{\Lambda} = \mathrm{diag}(\lambda_1, \lambda_2, \ldots, \lambda_n)$ is a diagonal matrix consisting of $\mathbf{L}$'s eigenvalues and $\mathbf{U}$ consists $n$ eigenvectors of $\mathbf{L}$ as its columns. Let $\mathbf{u}_i \in \mathbb{R}^n$ be the $i^{\text{th}}$ eigenvector of $\mathbf{L}$ and its $j^{\text{th}}$ entry denoted as $\mathbf{u}_{i,j}$. Then, the positional encoding of node $i \in V$ is defined as $\mathbf{p}_i = [\mathbf{u}_{1,i}, \ldots, \mathbf{u}_{n,i}]^\top \in \mathbb{R}^n$. In practice, the dimension $d_P$ of positional encoding is usually set to be less than $n$ for computational efficiency (Dwivedi & Bresson, 2020), i.e., $d_P \ll n$ and $\mathbf{p}_i = [\mathbf{u}_{1,i}, \ldots, \mathbf{u}_{d_P,i}]^\top \in \mathbb{R}^{d_P}$.

**Discrete Fourier Transform**. To pave the way for learnable positional encoding, we need to first introduce the basics of Discrete Fourier Transform (DFT) (Sundararajan, 2001). In brief, DFT is designed to convert a finite-length sequence of samples into another sequence in the frequency domain (with the same length). Formally, given a sequence with length $N$, $\{\mathbf{x}_j\}_{j=1}^N$, the DFT operation $\mathcal{F}(\{\mathbf{x}_j\}_{j=1}^N)$ converts $\{\mathbf{x}_j\}_{j=1}^N$ to the complex-valued sequence $\{\mathbf{X}_j\}_{j=1}^N$ as $\mathbf{X}_j = \sum_{k=1}^N \mathbf{x}_k e^{-i2\pi \frac{j}{N} k}$. Also, to re-construct the original sequence $\{\mathbf{x}_j\}_{j=1}^N$, inverse DFT (IDFT) $\mathcal{F}^{-1}(\{\mathbf{X}_j\}_{j=1}^N)$ is defined as $\mathbf{x}_j = \sum_{k=1}^N \mathbf{X}_k e^{i2\pi \frac{j}{N} k}$.

# 3 PROPOSED METHOD: SIMPLETLP

To begin with, we focus on a general research question, i.e., *given two nodes $u$ and $v$, how can we predict whether the link (interaction or edge) exists between $u$ and $v$ at a certain timestamp $t$, effectively and efficiently*?

To answer this question, we propose a simple temporal link prediction model named SimpleTLP. It retrieves the interaction history of two nodes and decides whether the current link exists or not. Without heavy neural architectures like dense attention (Rampásek et al., 2022) or relational attention (Diao & Loynd, 2023), SimpleTLP only depends on two of our proposed simple yet effective techniques, i.e., *Learnable Positional Encoding Module* (LPE) and *Node-Link-Positional Encoder*, such that SimpleTLP can preserve the spatial-temporal topology rigorously (as shown in Section 4) and outperform attention-based state-of-the-art link prediction methods (as shown in Section 5).

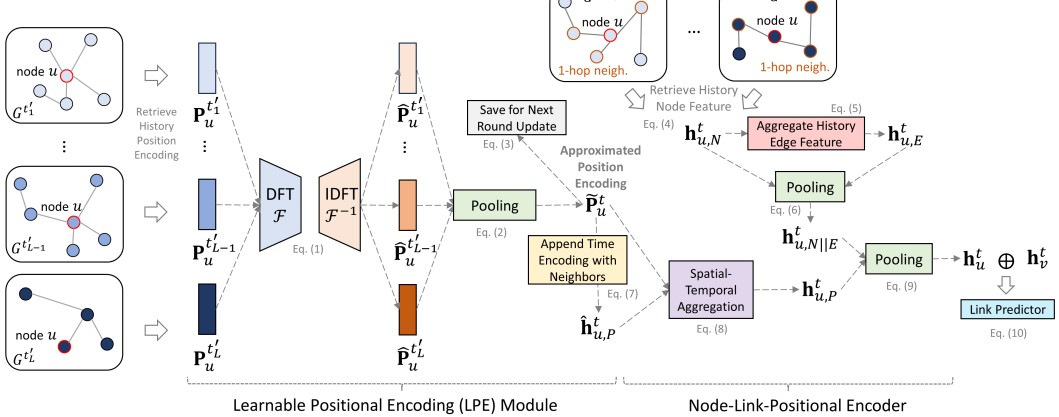

Figure 1: Overall Framework of SimpleTLP

The overall framework of SimpleTLP is shown in Figure 1 about how to retrieve and learn from history to make the current decision with two proposed techniques. Next, we will introduce the details of the proposed techniques, i.e., *Learnable Positional Encoding Module* in Section 3.1 and *Node-Link-Positional Encoder* in Section 3.2.

## 3.1 LEARNABLE POSITIONAL ENCODING (LPE) MODULE

The main idea of Learnable Positional Encoding (LPE) is to leverage historical positional encoding to determine whether the link exists at the current timestamp. However, learning from the history data

in an effective and efficient way and only through positional encoding is challenging, because SOTA methods usually need a complex attention mechanism (Poursafaei et al., 2022; Yu et al., 2023).

To this end, we aim to only use positional encoding to carry informative and useful knowledge to record what had happened in the past (e.g., what interactions occurred and what the node and edge features looked like around the interaction). Existing simple hand-crafted positional encoding can be efficient but tend to induce suboptimal performance (Wang et al., 2021b; Cong et al., 2023).

Thus, we propose a "**learnable**" positional encoding module (LPE), enabling it to reflect complex and non-linear knowledge. As shown in Figure 1, in our SimpleTLP, LPE estimates (or approximates) the positional encoding at current time $t$ by applying the Fourier Transform (with learnable parameters) on the past positional encoding. It is worth noting that obtaining the positional encoding through LPE: (1) records the complex spatial-temporal information (proved from the spectral viewpoint in Section 4), (2) and avoids computing the Laplacian eigendecomposition for each timestamp from scratch, which is computationally expensive (e.g., $O(|V|^3)$ (Chen et al., 2023) or $O(|E|^{3/2})$ (Dwivedi et al., 2023))

Because of no peeking at the current graph topology structure and features, our LPE is an iterative updating module. Next, we introduce LPE in two-folds: (1) given the past positional encoding, how LPE approximates the current positional encoding (that will be used to determine links by *Node-Link-Positional Encoder* in Section 3.2), and (2) how to store the approximate positional encoding and update/optimize it in the future round with current topology structure and features observed.

**Approximating current position encoding**. First of all, we denote the positional encoding of $u$ to be learned for time $t$ as $\mathbf{p}_u^t \in \mathbb{R}^{d_P}$. Suppose there are $L$ most recent distinct timestamps prior to $t$, i.e., $t'_1, \ldots, t'_L$ (where $t'_1 < \cdots < t'_L < t$) , and we can retrieve the already learned $\{\mathbf{p}_u^{t'_1}, \ldots, \mathbf{p}_u^{t'_L}\}$ that is regarded as a $L$-length sequence. Additionally, considering computational efficiency, we only retrieve $L$ most recent timestamps. Then, in this LPE module, we leverage the Discrete Fourier Transform as a way to capture the temporal dependencies among $\mathbf{p}_u^{t'_1}, \ldots, \mathbf{p}_u^{t'_L}$ and derive an estimated positional encoding of $u$ at time $t$ denoted by $\widetilde{\mathbf{p}}_u^t \in \mathbb{R}^{d_P}$.

To be specific, we start by applying Discrete Fourier Transform (DFT) on the sequence $\{\mathbf{p}_u^{t'}\}_{t'=t'_1}^{t'_L}$ to obtain the corresponding sequence in the frequency domain. Then, we employ a complex-valued learnable filter, $\mathbf{W}_{filter} \in \mathbb{C}^{d_P \times L}$, to filter out noises of the sequence in the frequency domain, and finally transform back to the original domain. The process can be mathematically expressed as

$$\{\widehat{\mathbf{p}}_u^{t'_j}\}_{j=1}^L = \mathcal{F}^{-1}(\mathbf{W}_{filter} \odot \mathcal{F}(\{\mathbf{p}_u^{t'_j}\}_{j=1}^L)) \tag{1}$$

Then, we obtain the approximate positional encoding $\widetilde{\mathbf{p}}_u^t$ by applying sum-pooling with learnable weights on $\{\widehat{\mathbf{p}}_u^{t'_j}\}_{j=1}^L$

$$\widetilde{\mathbf{p}}_u^t = \begin{bmatrix} \widehat{\mathbf{p}}_u^{t'_1} & \ldots & \widehat{\mathbf{p}}_u^{t'_L} \end{bmatrix} \mathbf{W}_P^{sum} \tag{2}$$

where $\mathbf{W}_P^{sum} \in \mathbb{R}^{L \times 1}$. Note that the approximate positional encoding $\widetilde{\mathbf{p}}_u^t$ will be sent to the *Node-Link-positional Encoder* (in Section. 3.2) for making predictions for current time $t$.

**Storing approximate positional encoding for future retrieval and optimization**. Since SimpleTLP does not peek at the current topology structure and features when making the prediction for now, i.e., only using approximate positional encoding $\widetilde{\mathbf{p}}_u^t$, we want to clarify how the current approximate positional encoding get updated and optimized in the next round.

After making the current link predictions at time $t$ (as shown in Section 3.2), SimpleTLP can then "peek" (or "reveal") the ground-truth links at time $t$, such that it can update the approximation $\widetilde{\mathbf{p}}_u^t$ into updated $\mathbf{p}_u^t$ for the future retrieval and predictions.

Then, we need to update the positional encoding of $u$ at $t$. We sample the most $K$ recent interactions, *including those that occur at time $t$*. Suppose they are indexed as $\{(\bar{u}_1, u, \bar{t}_1), \ldots, (\bar{u}_K, u, \bar{t}_K)\}$ where $\bar{t}_1 \leq \cdots \leq \bar{t}_K \leq t$. Leveraging the approximate positional encoding from $\bar{u}_1, \ldots, \bar{u}_K$, $\mathbf{p}_u^t$ is derived as

$$\mathbf{p}_u^t = \widetilde{\mathbf{p}}_u^t + tanh(\mathbf{W}_P^{(self)}\widetilde{\mathbf{p}}_u^t + \mathbf{W}_P^{(2)}\text{RELU}(\mathbf{W}_P^{(1)}\widehat{\mathbf{q}}_u^t)) \in \mathbb{R}^{d_P} \tag{3}$$

where $\widehat{\mathbf{q}}_u^t = \sum_{j=1}^K [(f_T(t - \bar{t}_j))^\top \;||\; (\widetilde{\mathbf{p}}_{\bar{u}_j}^t)^\top]^\top \in \mathbb{R}^{d_T + d_P}$, $f_T$ is the time encoder, and $\mathbf{W}_P^{(1)} \in \mathbb{R}^{d_P \times (d_P + d_T)}, \mathbf{W}_P^{(2)} \in \mathbb{R}^{d_P \times d_P}, \mathbf{W}_P^{(self)} \in \mathbb{R}^{d_P \times d_P}$ are learnable MLP parameters.

At the first time $t_0$, we denote the initial positional encoding as the Laplacian of the initial graph snapshot $G^{t_0}$. Formally, let $\mathbf{L}$ be the Laplacian of $G^{t_0}$ and the eigen-decomposition of $\mathbf{L}$ be $\mathbf{U}\Lambda\mathbf{U}^T$, then $\mathbf{p}_u^{t_0} = [\mathbf{u}_{1,u}, \ldots, \mathbf{u}_{d_P,u}]^T \in \mathbb{R}^{d_P}$, where $\mathbf{u}_i$ is the $i$-th eigenvector of $\mathbf{L}$, as defined in Section 2.

## 3.2 Node-Link-Positional Encoder

For node $u$, the Node-Link-Positional Encoder is designed to aggregate its historical node and edge features with the positional encoding of its recent 1-hop neighbors prior to time $t$, and, eventually, derive a compact representation associated with $u$ that encodes all the information.

In general, Node-Link-Positional Encoder works as follows. We first use $\mathbf{h}_{u,N}^t$ and $\mathbf{h}_{u,E}^t$ to denote the node and link encoding of recent 1-hop neighbors of node $u$. Then, we apply MLP transformations on $\mathbf{h}_{u,N}^t$ and $\mathbf{h}_{u,E}^t$ to obtain the encoding, denoted by $\mathbf{h}_{u,N\|E}^t$, which represents the combination of node-link information. For adding positional information, we use $\mathbf{h}_{u,P}^t$ to denote the aggregated positional encoding of recent 1-hop neighbors, and further transform them by MLP layers. Eventually, $\mathbf{h}_{u,N\|E}^t$ and $\mathbf{h}_{u,P}^t$ are combined by a 1-layer MLP to derive a temporal representation at time $t$ for $u$, denoted by $\mathbf{h}_u^t$, which will be sent together with $\mathbf{h}_v^t$ to the link predictor in Section 3.3 to determine whether the link exists between nodes $u$ and $v$ at time $t$.

The detailed computational procedures of the Node-Link-Positional Encoder are explained below.

First, the node encoding of recent 1-hop neighbors of $u$ can be obtained by aggregating features of nodes that interact with $u$ in the time interval $[t - t_{gap}; t]$, where $t_{gap}$ is a dataset-dependent hyper-parameter. Mathematically, this process can be described as

$$\mathbf{h}_{u,N}^t = \mathbf{s}_u^t + \text{Mean}(\{\mathbf{s}_{u'}^{t'} | (u', u, t') \in \mathcal{N}_u^{t - t_{gap}} \cup \cdots \cup \mathcal{N}_u^t\}; t' \in [t - t_{gap}, t]) \in \mathbb{R}^{d_N} \quad (4)$$

where $\mathbf{s}_u^t$ is the input node features, and $\text{Mean}(\cdot)$ is the mean pooling operation.

Then, we derive the link encoding for $u$ by extracting the $K$ most recent interactions involving $u$ that occur prior to time $t$, combine the edge features and the time encoding of these interactions, and further apply MLP transformations on these representations. Specifically, suppose the most $K$ recent interactions of $u$ are indexed as $\{(u_1, u, t_1'), \ldots, (u_K, u, t_K')\}$, where $t_1' \leq \cdots \leq t_K' < t$. Let $\mathbf{H}_{u,E}^t \in \mathbb{R}^{K \times (d_T + d_E)}$ be a matrix, whose $k^{\text{th}}$ row is defined as $[\mathbf{H}_{u,E}^t]_k = [(f_T(t - t_k'))^\top \;||\; (\mathbf{e}_{u,u_k'}^t)^\top] \in \mathbb{R}^{1 \times (d_T + d_E)}$, where $[\cdot||\cdot]$ denotes concatenation, $f_T$ is the time encoder, and $e$ denotes the edge feature. If $u$ does not involve at least $K$ interactions before $t$ then $\mathbf{H}_{u,E}^t$ is zero-padded so that the matrix has $K$ rows. We first apply a 1-layer MLP on the last dimension of $\mathbf{H}_{u,E}^t$, sum-pool over $K$ rows of $\mathbf{H}_{u,E}^t$ with learnable weights to obtain a vector of dimension $d_T + d_E$, and finalize the encoding with an additional 1-layer MLP transformation as

$$\mathbf{h}_{u,E}^t = \mathbf{W}_{link}^{(2)}(\text{RELU}((\mathbf{H}_{u,E}^t \mathbf{W}_{link}^{(1)})^\top \mathbf{W}_{link}^{sum})) \in \mathbb{R}^{d_T + d_E} \quad (5)$$

where $\mathbf{W}_{link}^{(1)}, \mathbf{W}_{link}^{(2)} \in \mathbb{R}^{(d_T + d_E) \times (d_T + d_E)}$, and $\mathbf{W}_{link}^{sum} \in \mathbb{R}^{K \times 1}$.

After that, $\mathbf{h}_{u,N\|E}^t$ is obtained by a 1-layer MLP transformation as

$$\mathbf{h}_{u,N\|E}^t = \mathbf{W}_{N\|E}([(\mathbf{h}_{u,N}^t)^\top \;||\; (\mathbf{h}_{u,E}^t)^T])^\top \in \mathbb{R}^{d_N} \quad (6)$$

where $\mathbf{W}_{N\|E} \in \mathbb{R}^{d_N \times (d_N + d_T + d_E)}$.

Next, for historical positional information, we apply sum aggregation over the positional information from the aforementioned $K$ most recent interactions involving $u$ as

$$\widehat{\mathbf{h}}_{u,P}^t = \sum_{j=1}^K [(f_T(t - t_j'))^\top \;||\; (\widetilde{\mathbf{p}}_{u_j}^t)^\top]^\top \in \mathbb{R}^{d_T + d_P} \quad (7)$$

where $f_T$ is the time encoder. Then, $\mathbf{h}_{u,P}^t$ is obtained through the neighborhood-aggregated positional encoding $\widehat{\mathbf{h}}_{u,P}^t$ and $\widetilde{\mathbf{p}}_u^t$ as

$$\mathbf{h}_{u,P}^t = \widetilde{\mathbf{p}}_u^t + tanh(\mathbf{W}_P^{(self)} \widetilde{\mathbf{p}}_u^t + \mathbf{W}_P^{(2)} \text{RELU}(\mathbf{W}_P^{(1)} \widehat{\mathbf{h}}_{u,P}^t)) \in \mathbb{R}^{d_P} \quad (8)$$

where $\mathbf{W}_P^{(1)} \in \mathbb{R}^{d_P \times (d_P + d_T)}, \mathbf{W}_P^{(2)} \in \mathbb{R}^{d_P \times d_P}, \mathbf{W}_P^{(self)} \in \mathbb{R}^{d_P \times d_P}$, and $tanh$ is the hyperbolic tangent function.

Finally, the temporal representation at time $t$ of $u$, $\mathbf{h}_u^t$, is derived as

$$\mathbf{h}_u^t = \mathbf{W}[(\mathbf{h}_{u,N\|E}^t)^\top \| (\mathbf{h}_{u,P}^t)^\top]^\top \in \mathbb{R}^{d_N} \tag{9}$$

where $\mathbf{W} \in \mathbb{R}^{d_N \times (d_N + d_P)}$.

### 3.3 Optimization

With the two embeddings $\mathbf{h}_u^t$ and $\mathbf{h}_v^t$, we can now let SimpleTLP decide whether a link exists between nodes $u$ and $v$ at time $t$ and train SimpleTLP against the ground truth connections before $t$.

**Link predictor**. Given a node pair $(u, v)$ and timestamp $t$, SimpleTLP's goal is to predict whether there is a link between $(u, v)$ at time $t$. Specifically, the Link Predictor would give the probability $\hat{y} \in (0, 1)$ of whether $u$ and $v$ interact at time $t$ by

$$\hat{y} = sigmoid(\mathbf{W}^{(2)}\text{RELU}(\mathbf{W}^{(1)}[(\mathbf{h}_u^t)^\top \| (\mathbf{h}_v^t)^\top]^\top)) \in (0, 1) \tag{10}$$

where $sigmoid$ denotes the sigmoid function mapping all real values to the range $(0, 1)$, and $\mathbf{W}^{(1)} \in \mathbb{R}^{(2d_N) \times d_N}, \mathbf{W}^{(2)} \in \mathbb{R}^{d_N \times 1}$ are 1-layer MLPs.

**Loss functions**. To begin with, we point out the necessary components of the objective loss function for the training process of SimpleTLP, give formal definitions for those components, and derive the final loss function used in the training process of SimpleTLP.

Suppose there are $B$ positive (i.e., existing edges) samples $(u_1^{(pos)}, v_1^{(pos)}, t_1), \ldots, (u_B^{(pos)}, v_B^{(pos)}, t_B)$ and $B$ negative (i.e., non-existing edges) samples $(u_1^{(neg)}, v_1^{(neg)}, t_1), \ldots, (u_B^{(neg)}, v_B^{(neg)}, t_B)$. Thus the ground truth label for the positive samples should be 1, and 0 for the negative samples. As the link prediction can be regarded as a binary classification problem, we employ the binary cross-entropy loss function to obtain the link prediction loss $\mathcal{L}_{lp}$ as

$$\mathcal{L}_{lp} = \frac{-1}{2B}\left[\left(\sum_{i=1}^{B} \log\left(\hat{y}_{u_i^{(pos)}, v_i^{(pos)}}\right)\right) + \left(\sum_{i=1}^{B}(1 - \log\left(\hat{y}_{u_i^{(neg)}, v_i^{(neg)}}\right))\right)\right] \tag{11}$$

where $\hat{y}_{u_i^{(pos)}, v_i^{(pos)}}$ denotes the predicted probability of a link existing between $u_i^{(pos)}, v_i^{(pos)}$ at time $t_i$, as defined in Eq. 10, and $\hat{y}_{u_i^{(neg)}, v_i^{(neg)}}$ is the predicted probability of a link existing between $u_i^{(neg)}, v_i^{(neg)}$ at time $t_i$. Thus, ideally, $\hat{y}_{u_i^{(pos)}, v_i^{(pos)}}$ should be close to 1, and $\hat{y}_{u_i^{(neg)}, v_i^{(neg)}}$ should be close to zero.

In addition, we introduce the positional encoding loss $\mathcal{L}_{pe}$ to assess the learned positional encoding, defined in Eq. 2. Specifically, we would expect $\widetilde{\mathbf{p}}_{u_i^{(pos)}}^{t_i}$ be close to $\widetilde{\mathbf{p}}_{v_i^{(pos)}}^{t_i}$, as there is a link between $u_i^{(pos)}, v_i^{(pos)}$ at time $t_i$. Thus, the difference $||\widetilde{\mathbf{p}}_{u_i^{(pos)}}^{t_i} - \widetilde{\mathbf{p}}_{v_i^{(pos)}}^{t_i}||_2$ (where $|| \cdot ||_2$ denotes the 2-norm of a real-valued vector) should be small, so that the learned positional encoding correctly reflects the nature of positional encoding. Moreover, in order to avoid SimpleTLP giving similar positional encoding to all nodes to minimize the aforementioned 2-norm difference of positive samples, we add another constraint, i.e., maximizing the difference $\alpha_{neg}||\widetilde{\mathbf{p}}_{u_i^{(neg)}}^{t_i} - \widetilde{\mathbf{p}}_{v_i^{(neg)}}^{t_i}||_2$, where $\alpha_{neg}$ is a positive scaling factor. As $(u_1^{(neg)}, v_1^{(neg)}, t_1), \ldots, (u_B^{(neg)}, v_B^{(neg)}, t_B)$ are not all negative (i.e., not connected), we let $\alpha_{neg} < 1$ to avoid SimpleTLP emphasizing the negativity of these $B$ negative samples. Therefore, we define the positional encoding loss as

$$\mathcal{L}_{pe} = \frac{1}{B}\left[\left(\sum_{i=1}^{B} ||\widetilde{\mathbf{p}}_{u_i^{(pos)}}^{t_i} - \widetilde{\mathbf{p}}_{v_i^{(pos)}}^{t_i}||_2\right) - \alpha_{neg}\left(\sum_{i=1}^{B} ||\widetilde{\mathbf{p}}_{u_i^{(neg)}}^{t_i} - \widetilde{\mathbf{p}}_{v_i^{(neg)}}^{t_i}||_2\right)\right] \tag{12}$$

Finally, the overall loss function is

$$\mathcal{L} = (1 - \alpha_{pe})\mathcal{L}_{lp} + \alpha_{pe}\mathcal{L}_{pe} \tag{13}$$

where $\alpha_{pe} \in (0, 1)$ is the weight for $\mathcal{L}_{pe}$.

The algorithmic training procedures of SimpleTLP can be found in Appendix A.

## 4 THEORETICAL ANALYSIS

In this section, we first theoretically show that, at time $t$, the approximated positional encoding $\widetilde{\mathbf{p}}^t$, defined by Eq. 2, can be a good approximation of the positional encoding of graph snapshot $G^t$ by establishing a bound for the approximation error.

**Theorem 1.** *Suppose the temporal graph $G$ is "slowly changing" (i.e., the ground truth positional encoding $\mathbf{p}^0 \approx \cdots \approx \mathbf{p}^{t-1}$), then the corresponding approximated positional encoding of SimpleTLP satisfies*

$$||\widetilde{\mathbf{p}}_u^t - \widetilde{\mathbf{p}}_u^{t''}||_2 \leq \epsilon \tag{14}$$

*where $t'' > t$ is the future distinct timestamp of $t$, and $\epsilon$ is a bounding term that depends on $L$, which is the number of recent graph snapshots utilized in obtaining the approximated positional encoding, as stated in Eqs. 1 and 2. (Proof in Appendix B)*

Briefly, Theorem 1 indicates that when the graph is slowly changing, the positional encoding learned at the previous timestamp can well preserve the next close future timestamp's positional encoding. In other words, as Theorem 1 suggests, the positional encoding function of SimpleTLP can be a good representation of the positional information of future graph snapshots without leveraging the information about the future graph topology.

Note that the above statements do not necessarily mean that SimpleTLP does not need to update the positional encoding in every case. Actually, SimpleTLP indeed updates the positional encoding when new interaction happens. The role of Theorem 1 is to demonstrate the effectiveness of our positional encoding in the inductive setting, where the future scenario is usually not given or difficult to observe.

**Complexity Analysis**. Next, we provide the analysis on SimpleTLP's computational complexity. SimpleTLP consists of two main components: LPE Module and Node-Link-Positional Encoder. In LPE Module, the approximated positional encoding is derived by applying Discrete Fourier Transform on a $L$-length sequence of previous positional encoding, so the complexity is $\mathcal{O}(L\log(L))$, achieved by utilizing Fast Fourier Transform. Then, we update the positional encoding by sampling the $K$ most recent interactions, adding $\mathcal{O}(K)$ time. Therefore, for $n$ nodes, LPE Module costs $\mathcal{O}(n(L\log(L) + K))$. For Node-Link-Positional Encoder, for each node we retrieve node-level information from its neighborhood in the interval $[t-t_{gap};t]$ and sample its $K$ most recent interactions, so the complexity is $\mathcal{O}(n\cdot(t_{gap}+K))$, which can be achieved by pre-computing all temporal neighbors in the range $[t-t_{gap};t]$ of each node. Thus, the total time complexity is $\mathcal{O}(n(t_{gap} + K + L\log(L)))$, which scales linearly with the number of nodes, as $L, K, t_{gap}$ are constants independent of $n$.

Compared to recent SOTAs, DyGFormer (Yu et al., 2023) and FreeDyG (Tian et al., 2024), SimpleTLP is more efficient as both of these methods employ neighborhood co-occurrence for each node pair of interaction, making their worst-case complexity scales with the number of edges, $|E|$, while SimpleTLP's complexity scales with $n$, the number of nodes, and its quite common that $n << |E|$, showing SimpleTLP's advantage in efficiency. Detailed theoretical time complexity comparison can be found in Appendix E.2, and detailed empirical running time comparison can be found in Appendix E.3

## 5 EXPERIMENTS

Here, we put the main results showing the outperformance of SimpleTLP in classic and large-scale datasets and demonstrating the substitutability of transformers by our lightweight solution. We leave the effectiveness comparison in more learning and sampling settings in Appendix E.1, running time comparison with SOTAs in Appendix E.3, LPE ablation studies wrt input node and edge features in Appendix E.4, analysis of important hyperparameters in Appendix E.5, robustness of SimpleTLP with different initial positional encoding input in Appendix E.6, and detailed reproducibility in Appendix F.

### 5.1 EXPERIMENTAL SETTINGS

**Datasets and baselines.** We assess the ability of SimpleTLP in performing link prediction with 13 datasets covering various domains and collected by (Poursafaei et al., 2022): *Wikipedia, Reddit, MOOC, LastFM, Enron, Social Evo., UCI, Flights, Can. Parl., US Legis., UN Trade, UN Vote,* and

*Contact.* Details about the dataset statistics are shown in Appendix D. We compare SimpleTLP with 8 state-of-the-art baselines, including JODIE (Kumar et al., 2019), DyRep (Trivedi et al., 2019), TGAT (Xu et al., 2020), TGN (Rossi et al., 2020), CAWN (Wang et al., 2021b), TCL (Wang et al., 2021a), GraphMixer (Cong et al., 2023), DyGFormer (Yu et al., 2023), and FreeDyG (Tian et al., 2024). Across all 13 datasets, training/validation/testing sets are following the standard library (Yu et al., 2023) by chronological splits with ratios 70%/15%/15%. The large-scale datasets with pre-defined splits are publicly available at TGB Benchmark 2.0 (Gastinger et al., 2024)

**Evaluation metrics and settings.** Following existing works (Xu et al., 2020; Rossi et al., 2020; Yu et al., 2023) in evaluating models for link prediction, we assess SimpleTLP under two settings: **transductive** setting and **inductive** setting. Under the transductive setting, models will predict the link occurrence in future timestamps between nodes that had been observed during the training process, while the inductive setting involves predicting future links between unseen nodes in the training process. For each setting, we use two evaluation metrics: Average Precision (AP) and Area Under the Receiver Operating Characteristic Curve (AUC-ROC). In addition, following (Poursafaei et al., 2022), for more robust evaluation, each baseline is assessed with three negative sampling strategies (NSS): random, historical, and inductive (Details in Appendix F).

## 5.2 TEMPORAL LINK PREDICTION PERFORMANCE ON 13 CLASSIC DATASETS

Due to the limited space, we present all baseline methods with respect to AP and AUC-ROC metrics with *random* negative sampling strategy here for the transductive setting in Table 1 and the inductive setting in Table 2. The performance of *historical* and *inductive* negative sampling strategies on both learning settings are placed in Appendix E.1. The first and second results are highlighted with **Red** and **Blue**. For the *random* negative sampling strategy, in Tables 1 and 2, SimpleTLP achieves competitive results over 13 datasets and outperforming most of the baselines with average rankings close to 1 under both settings. Moreover, the second-best results of SimpleTLP closely approach the corresponding best results by a small gap. Notably, on some datasets, such as *UN Trade, UN Vote*, the difference between the second-best and SimpleTLP's performance is substantial, suggesting that SimpleTLP makes great improvements over existing methods. Regarding the *historical* and *inductive* sampling strategies, SimpleTLP also attains competitive results, as suggested in Tables 7, 8, 9, and 10, at most cases, SimpleTLP surpasses the second-best by a substantial margin.

Table 1: Performance comparison in the *transductive setting* with *random negative sampling strategy*.

| Metric | Datasets | JODIE | DyRep | TGAT | TGN | CAWN | TCL | GraphMixer | DyGFormer | FreeDyG | SimpleTLP |
|---|---|---|---|---|---|---|---|---|---|---|---|
| AP | Wikipedia | 96.50 ± 0.14 | 94.86 ± 0.06 | 96.94 ± 0.06 | 98.45 ± 0.06 | 98.76 ± 0.03 | 96.47 ± 0.16 | 97.25 ± 0.03 | 99.03 ± 0.02 | 99.26 ± 0.01 | 99.34 ± 0.04 |
| | Reddit | 98.31 ± 0.14 | 98.22 ± 0.04 | 98.52 ± 0.02 | 98.63 ± 0.06 | 99.11 ± 0.01 | 97.53 ± 0.02 | 97.31 ± 0.01 | 99.22 ± 0.01 | 99.48 ± 0.01 | 99.37 ± 0.04 |
| | MOOC | 80.23 ± 2.44 | 81.97 ± 0.49 | 85.84 ± 0.15 | 89.15 ± 1.60 | 80.15 ± 0.25 | 82.38 ± 0.24 | 82.78 ± 0.15 | 87.52 ± 0.49 | 89.61 ± 0.19 | 86.94 ± 0.34 |
| | LastFM | 70.85 ± 2.13 | 71.92 ± 2.21 | 73.42 ± 0.21 | 77.07 ± 3.97 | 86.99 ± 0.06 | 67.27 ± 2.16 | 75.61 ± 0.24 | 93.00 ± 0.12 | 92.15 ± 0.16 | 96.06 ± 0.30 |
| | Enron | 84.77 ± 0.30 | 82.38 ± 3.36 | 71.12 ± 0.97 | 86.53 ± 1.11 | 89.56 ± 0.09 | 79.70 ± 0.71 | 82.25 ± 0.16 | 92.47 ± 0.12 | 92.51 ± 0.05 | 93.96 ± 0.36 |
| | Social Evo. | 89.89 ± 0.55 | 88.87 ± 0.30 | 93.16 ± 0.17 | 93.57 ± 0.17 | 84.96 ± 0.09 | 93.13 ± 0.16 | 93.37 ± 0.07 | 94.73 ± 0.01 | 94.91 ± 0.01 | 92.22 ± 0.25 |
| | UCI | 89.43 ± 1.09 | 65.14 ± 2.30 | 79.63 ± 0.70 | 92.34 ± 1.04 | 95.18 ± 0.06 | 89.57 ± 1.63 | 93.25 ± 0.57 | 95.79 ± 0.17 | 96.28 ± 0.11 | 96.67 ± 0.47 |
| | Flights | 95.60 ± 1.73 | 95.29 ± 0.72 | 94.03 ± 0.18 | 97.95 ± 0.14 | 98.51 ± 0.01 | 91.23 ± 0.02 | 90.99 ± 0.05 | 98.91 ± 0.01 | 98.12 ± 0.19 | 98.94 ± 0.10 |
| | Can. Parl. | 69.26 ± 0.31 | 66.54 ± 2.76 | 70.73 ± 0.72 | 70.88 ± 2.34 | 69.82 ± 2.34 | 68.67 ± 2.67 | 77.04 ± 0.46 | 97.36 ± 0.45 | 72.22 ± 2.47 | 98.24 ± 0.11 |
| | US Legis. | 75.05 ± 1.52 | 75.34 ± 0.39 | 68.52 ± 3.16 | 75.99 ± 0.58 | 70.58 ± 0.48 | 69.59 ± 0.48 | 70.74 ± 1.02 | 71.11 ± 0.59 | 69.94 ± 0.59 | 76.74 ± 0.60 |
| | UN Trade | 64.94 ± 0.31 | 63.21 ± 0.93 | 61.47 ± 0.18 | 65.03 ± 1.37 | 65.39 ± 0.12 | 62.21 ± 0.03 | 62.61 ± 0.27 | 66.46 ± 1.29 | - | 75.84 ± 2.08 |
| | UN Vote | 63.91 ± 0.81 | 62.81 ± 0.80 | 52.21 ± 0.98 | 65.72 ± 2.17 | 52.84 ± 0.10 | 51.90 ± 0.30 | 52.11 ± 0.16 | 55.55 ± 0.42 | 51.99 ± 1.13 | 73.40 ± 0.53 |
| | Contact | 95.31 ± 1.33 | 95.98 ± 0.15 | 96.28 ± 0.09 | 96.89 ± 0.56 | 90.26 ± 0.28 | 92.44 ± 0.12 | 91.92 ± 0.03 | 98.29 ± 0.01 | 97.99 ± 0.03 | 98.16 ± 0.09 |
| | Avg. Rank | 6.77 | 7.38 | 7.23 | 4.08 | 5.38 | 8.46 | 6.77 | 2.77 | 3.85 | 1.85 |
| ROC-AUC | Wikipedia | 96.33 ± 0.07 | 94.37 ± 0.09 | 96.67 ± 0.07 | 98.37 ± 0.07 | 98.54 ± 0.04 | 95.84 ± 0.18 | 96.92 ± 0.03 | 98.91 ± 0.02 | 99.41 ± 0.01 | 99.48 ± 0.03 |
| | Reddit | 98.31 ± 0.05 | 98.17 ± 0.05 | 98.47 ± 0.02 | 98.60 ± 0.06 | 99.01 ± 0.01 | 97.42 ± 0.02 | 97.17 ± 0.02 | 99.15 ± 0.01 | 99.50 ± 0.01 | 99.49 ± 0.03 |
| | MOOC | 83.81 ± 2.09 | 85.03 ± 0.58 | 87.11 ± 0.19 | 91.21 ± 1.15 | 80.38 ± 0.26 | 83.12 ± 0.18 | 84.01 ± 0.17 | 87.91 ± 0.58 | 89.93 ± 0.35 | 89.28 ± 0.28 |
| | LastFM | 70.49 ± 1.66 | 71.16 ± 1.89 | 71.59 ± 0.18 | 78.47 ± 2.94 | 85.92 ± 0.10 | 64.06 ± 1.16 | 73.53 ± 0.12 | 93.05 ± 0.10 | 93.42 ± 0.15 | 97.52 ± 0.17 |
| | Enron | 87.96 ± 0.52 | 84.89 ± 3.00 | 68.89 ± 1.10 | 88.32 ± 0.99 | 90.45 ± 0.14 | 75.74 ± 0.72 | 84.38 ± 0.21 | 93.33 ± 0.13 | 94.01 ± 0.11 | 95.98 ± 0.23 |
| | Social Evo. | 92.05 ± 0.46 | 90.76 ± 0.21 | 94.76 ± 0.16 | 95.39 ± 0.17 | 87.34 ± 0.08 | 94.84 ± 0.17 | 95.23 ± 0.07 | 96.30 ± 0.01 | 96.59 ± 0.04 | 94.33 ± 0.21 |
| | UCI | 90.44 ± 0.49 | 68.77 ± 2.34 | 78.53 ± 0.74 | 92.03 ± 1.13 | 93.87 ± 0.08 | 87.82 ± 1.36 | 91.81 ± 0.67 | 94.49 ± 0.26 | 95.00 ± 0.21 | 97.62 ± 0.23 |
| | Flights | 96.21 ± 1.42 | 95.95 ± 0.62 | 94.13 ± 0.17 | 98.22 ± 0.13 | 98.45 ± 0.01 | 91.21 ± 0.02 | 91.13 ± 0.01 | 98.93 ± 0.01 | 98.03 ± 0.17 | 99.38 ± 0.04 |
| | Can. Parl. | 78.21 ± 0.23 | 73.35 ± 3.67 | 75.69 ± 0.78 | 76.99 ± 1.80 | 75.70 ± 3.27 | 72.46 ± 3.23 | 83.17 ± 0.53 | 97.76 ± 0.41 | 81.09 ± 2.28 | 98.97 ± 0.06 |
| | US Legis. | 82.85 ± 1.07 | 82.28 ± 0.32 | 75.84 ± 1.99 | 83.34 ± 0.43 | 77.16 ± 0.39 | 76.27 ± 0.63 | 76.96 ± 0.79 | 77.90 ± 0.58 | 77.26 ± 0.50 | 83.89 ± 0.43 |
| | UN Trade | 69.62 ± 0.44 | 67.44 ± 0.83 | 64.01 ± 0.12 | 69.10 ± 1.67 | 68.54 ± 0.18 | 64.72 ± 0.05 | 65.52 ± 0.51 | 70.20 ± 1.44 | - | 83.17 ± 1.28 |
| | UN Vote | 68.53 ± 0.95 | 67.18 ± 1.04 | 52.83 ± 1.12 | 69.71 ± 2.65 | 53.09 ± 0.22 | 51.88 ± 0.36 | 52.46 ± 0.27 | 57.12 ± 0.62 | 52.79 ± 1.56 | 79.59 ± 0.02 |
| | Contact | 96.66 ± 0.89 | 96.48 ± 0.14 | 96.95 ± 0.08 | 97.54 ± 0.35 | 89.99 ± 0.34 | 94.15 ± 0.09 | 93.94 ± 0.02 | 98.53 ± 0.01 | 98.39 ± 0.02 | 98.72 ± 0.06 |
| | Avg. Rank | 6.08 | 7.31 | 7.46 | 3.92 | 6.0 | 8.69 | 7.15 | 3.0 | 3.69 | 1.69 |

## 5.3 DEMONSTRATING THE SUBSTITUTABILITY OF TRANSFORMER IN TEMPORAL LINK PREDICTION TASKS

Here, we empirically show that simple architectures like MLPs can fully harness our learnable positional encoding to finish temporal link prediction tasks.

Table 2: Performance Comparison in the *inductive setting* with *random negative sampling strategy*.

| Metric | Datasets | JODIE | DyRep | TGAT | TGN | CAWN | TCL | GraphMixer | DyGFormer | FreeDyG | SimpleTLP |
|---|---|---|---|---|---|---|---|---|---|---|---|
| AP | Wikipedia | 94.82 ± 0.20 | 92.43 ± 0.37 | 96.22 ± 0.07 | 97.83 ± 0.04 | 98.24 ± 0.03 | 96.22 ± 0.17 | 96.65 ± 0.02 | 98.59 ± 0.03 | 98.97 ± 0.01 | 99.15 ± 0.04 |
| | Reddit | 96.50 ± 0.13 | 96.09 ± 0.11 | 97.09 ± 0.04 | 97.50 ± 0.07 | 98.62 ± 0.01 | 94.09 ± 0.07 | 95.26 ± 0.02 | 98.84 ± 0.02 | 98.91 ± 0.01 | 98.02 ± 0.09 |
| | MOOC | 79.63 ± 1.92 | 81.07 ± 0.44 | 85.50 ± 0.19 | 89.04 ± 1.17 | 81.42 ± 0.24 | 80.60 ± 0.22 | 81.41 ± 0.21 | 86.96 ± 0.43 | 87.75 ± 0.62 | 88.49 ± 0.24 |
| | LastFM | 81.61 ± 3.82 | 83.02 ± 1.48 | 78.63 ± 0.31 | 81.45 ± 4.29 | 89.42 ± 0.07 | 73.53 ± 1.66 | 82.11 ± 0.42 | 94.23 ± 0.09 | 94.89 ± 0.01 | 96.25 ± 0.43 |
| | Enron | 80.72 ± 1.39 | 74.55 ± 3.95 | 67.05 ± 1.51 | 77.94 ± 1.02 | 86.35 ± 0.51 | 76.14 ± 0.79 | 75.88 ± 0.48 | 89.76 ± 0.34 | 89.69 ± 0.17 | 89.17 ± 0.88 |
| | Social Evo. | 91.96 ± 0.48 | 90.04 ± 0.47 | 91.41 ± 0.16 | 90.77 ± 0.86 | 79.94 ± 0.18 | 91.55 ± 0.09 | 91.86 ± 0.06 | 93.14 ± 0.04 | 94.76 ± 0.05 | 91.71 ± 0.32 |
| | UCI | 79.86 ± 1.48 | 57.48 ± 1.87 | 79.54 ± 0.48 | 88.12 ± 2.05 | 92.73 ± 0.06 | 87.36 ± 2.03 | 91.19 ± 0.42 | 94.54 ± 0.12 | 94.85 ± 0.10 | 94.60 ± 0.41 |
| | Flights | 94.74 ± 0.37 | 92.88 ± 0.73 | 88.73 ± 0.33 | 95.03 ± 0.60 | 97.06 ± 0.02 | 83.41 ± 0.07 | 83.03 ± 0.05 | 97.79 ± 0.02 | 96.42 ± 0.32 | 97.43 ± 0.15 |
| | Can. Parl. | 53.92 ± 0.94 | 54.02 ± 0.76 | 55.18 ± 0.79 | 54.10 ± 0.93 | 55.80 ± 0.69 | 54.30 ± 0.66 | 55.91 ± 0.82 | 87.74 ± 0.71 | 52.96 ± 1.05 | 92.25 ± 0.24 |
| | US Legis. | 54.93 ± 2.29 | 57.28 ± 0.71 | 51.00 ± 3.11 | 58.63 ± 0.37 | 53.17 ± 1.20 | 52.59 ± 0.97 | 50.71 ± 0.76 | 54.28 ± 2.87 | 53.05 ± 2.59 | 61.98 ± 1.31 |
| | UN Trade | 59.65 ± 0.77 | 57.02 ± 0.69 | 61.03 ± 0.18 | 58.31 ± 3.15 | 65.24 ± 0.21 | 62.21 ± 0.12 | 62.17 ± 0.31 | 64.55 ± 0.62 | - | 76.80 ± 2.01 |
| | UN Vote | 56.64 ± 0.96 | 54.62 ± 2.22 | 52.24 ± 1.46 | 58.85 ± 2.51 | 49.94 ± 0.45 | 51.60 ± 0.97 | 50.68 ± 0.44 | 55.93 ± 0.39 | 50.51 ± 1.02 | 75.09 ± 0.34 |
| | Contact | 94.34 ± 1.45 | 92.18 ± 1.41 | 95.87 ± 0.11 | 93.82 ± 0.99 | 89.55 ± 0.30 | 91.11 ± 0.12 | 90.59 ± 0.05 | 98.03 ± 0.02 | 97.66 ± 0.04 | 97.25 ± 0.07 |
| | Avg. Rank | 6.38 | 7.54 | 7.08 | 5.31 | 5.38 | 7.54 | 6.92 | 2.62 | 4.15 | 2.08 |
| ROC-AUC | Wikipedia | 94.33 ± 0.27 | 91.49 ± 0.45 | 95.90 ± 0.09 | 97.72 ± 0.03 | 98.03 ± 0.04 | 95.57 ± 0.20 | 96.30 ± 0.04 | 98.48 ± 0.03 | 99.01 ± 0.02 | 99.30 ± 0.03 |
| | Reddit | 96.52 ± 0.13 | 96.05 ± 0.12 | 96.98 ± 0.04 | 97.39 ± 0.07 | 98.42 ± 0.02 | 93.80 ± 0.07 | 94.97 ± 0.05 | 98.71 ± 0.01 | 98.84 ± 0.01 | 98.49 ± 0.06 |
| | MOOC | 83.16 ± 1.30 | 84.03 ± 0.49 | 86.84 ± 0.17 | 91.24 ± 0.99 | 81.86 ± 0.25 | 81.43 ± 0.19 | 82.77 ± 0.24 | 87.62 ± 0.51 | 87.01 ± 0.74 | 90.63 ± 0.21 |
| | LastFM | 81.13 ± 3.39 | 82.24 ± 1.51 | 76.99 ± 0.29 | 82.61 ± 3.15 | 87.82 ± 0.12 | 70.84 ± 0.85 | 80.37 ± 0.18 | 94.08 ± 0.08 | 94.32 ± 0.03 | 97.66 ± 0.21 |
| | Enron | 81.96 ± 1.34 | 76.34 ± 4.20 | 64.63 ± 1.74 | 78.83 ± 1.11 | 87.02 ± 0.50 | 72.33 ± 0.99 | 76.51 ± 0.71 | 90.69 ± 0.26 | 89.51 ± 0.20 | 92.30 ± 0.57 |
| | Social Evo. | 93.70 ± 0.29 | 91.18 ± 0.49 | 93.41 ± 0.19 | 93.43 ± 0.59 | 84.73 ± 0.27 | 93.71 ± 0.18 | 94.09 ± 0.07 | 95.29 ± 0.03 | 96.41 ± 0.07 | 94.09 ± 0.24 |
| | UCI | 78.80 ± 0.94 | 58.08 ± 1.81 | 77.64 ± 0.38 | 86.68 ± 2.29 | 90.40 ± 0.11 | 84.49 ± 1.82 | 89.30 ± 0.57 | 92.63 ± 0.13 | 93.01 ± 0.08 | 95.88 ± 0.17 |
| | Flights | 95.21 ± 0.32 | 93.56 ± 0.70 | 88.64 ± 0.35 | 95.92 ± 0.43 | 96.86 ± 0.02 | 82.48 ± 0.01 | 82.27 ± 0.06 | 97.80 ± 0.02 | 96.05 ± 0.29 | 98.46 ± 0.07 |
| | Can. Parl. | 53.81 ± 1.14 | 55.27 ± 0.49 | 56.51 ± 0.75 | 55.86 ± 0.75 | 58.83 ± 1.13 | 55.83 ± 1.07 | 58.32 ± 1.08 | 89.33 ± 0.48 | 52.89 ± 1.61 | 95.06 ± 0.11 |
| | US Legis. | 58.12 ± 2.35 | 61.07 ± 0.56 | 48.27 ± 3.50 | 62.38 ± 0.48 | 51.49 ± 1.13 | 50.43 ± 1.48 | 47.20 ± 0.89 | 53.21 ± 3.04 | 53.01 ± 3.14 | 66.18 ± 1.19 |
| | UN Trade | 62.28 ± 0.50 | 58.82 ± 0.98 | 62.72 ± 0.12 | 59.99 ± 3.50 | 67.05 ± 0.21 | 63.76 ± 0.07 | 63.48 ± 0.37 | 67.25 ± 1.05 | - | 83.95 ± 1.76 |
| | UN Vote | 58.13 ± 1.43 | 55.13 ± 3.46 | 51.83 ± 1.35 | 61.23 ± 2.71 | 48.34 ± 0.76 | 50.51 ± 1.05 | 50.04 ± 0.86 | 56.73 ± 0.69 | 49.97 ± 1.22 | 82.01 ± 0.21 |
| | Contact | 95.37 ± 0.92 | 91.89 ± 0.38 | 96.53 ± 0.10 | 94.84 ± 0.75 | 89.07 ± 0.34 | 93.05 ± 0.09 | 92.83 ± 0.05 | 98.30 ± 0.02 | 98.11 ± 0.02 | 97.99 ± 0.07 |
| | Avg. Rank | 6.38 | 7.54 | 7.08 | 4.92 | 5.77 | 7.77 | 7.08 | 2.62 | 4.31 | 1.54 |

By performing an ablation study of using Transformers (instead of MLPs) to help SimpleTLP learn positional encodings, as shown in Table 3, we can discern (1) in most cases (i.e., 9 of 13 datasets), our vanilla SimpleTLP can outperform, and the gain is not marginal; (2) in a few cases, adding the attention mechanism on SimpleTLP to replace MLPs can boost the performance to a small extent, and it is prone to cost more computational memory, like OOM in the Flights dataset.

Table 3: Comparison of replacing MLPs in SimpleTLP with Transformers.

| Dataset | Method | Transductive AP | Transductive ROC-AUC | Inductive AP | Inductive ROC-AUC |
|---|---|---|---|---|---|
| Wikipedia | SimpleTLP + Transformer | 98.50 ± 0.73 | 98.85 ± 0.50 | 98.11 ± 0.83 | 98.49 ± 0.62 |
| | SimpleTLP | **99.34 ± 0.04** | **99.48 ± 0.03** | **99.15 ± 0.04** | **99.30 ± 0.03** |
| Reddit | SimpleTLP + Transformer | 98.90 ± 0.10 | 99.13 ± 0.08 | 97.75 ± 0.26 | 98.29 ± 0.16 |
| | SimpleTLP | **99.37 ± 0.04** | **99.49 ± 0.03** | **98.02 ± 0.09** | **98.49 ± 0.06** |
| MOOC | SimpleTLP + Transformer | 82.46 ± 3.61 | 85.38 ± 2.93 | 82.57 ± 4.24 | 85.80 ± 3.18 |
| | SimpleTLP | **86.94 ± 0.34** | **89.28 ± 0.28** | **88.49 ± 0.24** | **90.63 ± 0.21** |
| LastFM | SimpleTLP + Transformer | 95.50 ± 0.70 | 97.19 ± 0.50 | 96.07 ± 0.36 | 97.62 ± 0.25 |
| | SimpleTLP | **96.06 ± 0.30** | **97.52 ± 0.17** | **96.25 ± 0.43** | **97.66 ± 0.21** |
| Enron | SimpleTLP + Transformer | 92.56 ± 0.53 | 95.02 ± 0.33 | 86.41 ± 1.15 | 90.70 ± 0.72 |
| | SimpleTLP | **93.96 ± 0.36** | **95.98 ± 0.23** | **89.17 ± 0.88** | **92.30 ± 0.57** |
| SocialEvo | SimpleTLP + Transformer | 90.26 ± 0.69 | 92.61 ± 0.58 | 90.75 ± 0.31 | 93.50 ± 0.30 |
| | SimpleTLP | **92.22 ± 0.25** | **94.33 ± 0.21** | **91.71 ± 0.32** | **94.09 ± 0.24** |
| UCI | SimpleTLP + Transformer | 96.06 ± 0.53 | 97.12 ± 0.41 | 93.00 ± 1.30 | 94.59 ± 0.85 |
| | SimpleTLP | **96.67 ± 0.47** | **97.62 ± 0.23** | **94.60 ± 0.41** | **95.88 ± 0.17** |
| Flights | SimpleTLP + Transformer | OOM | OOM | OOM | OOM |
| | SimpleTLP | 98.94 ± 0.10 | 99.38 ± 0.04 | 97.43 ± 0.15 | 98.46 ± 0.07 |
| Can. Parl. | SimpleTLP + Transformer | **98.36 ± 0.12** | **99.01 ± 0.07** | **92.59 ± 0.32** | **95.37 ± 0.25** |
| | SimpleTLP | 98.24 ± 0.11 | 98.97 ± 0.06 | 92.25 ± 0.24 | 95.06 ± 0.11 |
| US Legis. | SimpleTLP + Transformer | **76.84 ± 0.61** | **84.16 ± 0.40** | **64.07 ± 1.31** | **68.13 ± 1.78** |
| | SimpleTLP | 76.74 ± 0.60 | 83.89 ± 0.43 | 61.98 ± 1.31 | 66.18 ± 1.19 |
| UN Trade. | SimpleTLP + Transformer | **77.45 ± 3.27** | **83.94 ± 2.23** | **79.83 ± 3.28** | **85.95 ± 2.13** |
| | SimpleTLP | 75.84 ± 2.08 | 83.17 ± 1.28 | 76.80 ± 2.01 | 83.95 ± 1.76 |
| UN Vote | SimpleTLP + Transformer | **74.65 ± 0.46** | **80.73 ± 0.40** | **76.10 ± 0.88** | **82.89 ± 0.64** |
| | SimpleTLP | 73.40 ± 0.03 | 79.59 ± 0.02 | 75.09 ± 0.34 | 82.01 ± 0.21 |
| Contact | SimpleTLP + Transformer | 97.32 ± 0.15 | 98.12 ± 0.10 | 96.05 ± 0.41 | 97.18 ± 0.21 |
| | SimpleTLP | **98.16 ± 0.09** | **98.72 ± 0.06** | **97.25 ± 0.07** | **97.99 ± 0.07** |

Based on the above analysis, our design gets verified in-depth: the proposed learnable positional encoding is effective and friendly to lightweight neural architectures, and even simple MLPs can fully exploit its expressive power.

## 5.4 TEMPORAL LINK PREDICTION PERFORMANCE ON LARGE-SCALE BENCHMARK TGB 2.0

In this section, we assess the scalability of our method by evaluating SimpleTLP on large-scale temporal graphs from the TGB Benchmark 2.0 (Gastinger et al., 2024) with link prediction task. Specifically, we report SimpleTLP 's performance on *tgbl-review* and *tgbl-coin* datasets in Table 4, following the given pre-defined splits and metrics. Moreover, *tgbl-review* has **352,637 nodes** and **4,873,540 edges**, while *tgbl-coin* has **638,486 nodes** and **22,809,486 edges**.

As shown in Table 4, our method achieves competitive performance, ranking the 2nd place on *tgbl-review* [1] and the 3rd place on *tgbl-coin* [2], according to the worldwide open leaderboard.

Table 4: *Test MRR* and *Validation MRR* in the worldwide leaderboard of TGB datasets

| | tgbl-review | | | | tgbl-coin | | |
|---|---|---|---|---|---|---|---|
| Rank | Method | Test MRR | Validation MRR | Rank | Method | Test MRR | Validation MRR |
| 1 | GraphMixer | $0.521 \pm 0.015$ | $0.428 \pm 0.019$ | 1 | TNCN | $0.762 \pm 0.004$ | $0.740 \pm 0.002$ |
| 2 | **SimpleTLP** | **$0.411 \pm 0.011$** | **$0.330 \pm 0.017$** | 2 | DyGFormer | $0.752 \pm 0.004$ | $0.730 \pm 0.002$ |
| 3 | TNCN | $0.377 \pm 0.010$ | $0.325 \pm 0.003$ | 3 | **SimpleTLP** | **$0.711 \pm 0.0185$** | **$0.674 \pm 0.021$** |
| 4 | TGAT | $0.355 \pm 0.012$ | $0.324 \pm 0.006$ | 4 | TGN | $0.586 \pm 0.037$ | $0.607 \pm 0.014$ |
| 5 | TGN | $0.349 \pm 0.020$ | $0.313 \pm 0.012$ | 5 | EdgeBank(tw) | 0.580 | 0.492 |
| 6 | NAT | $0.341 \pm 0.020$ | $0.302 \pm 0.011$ | 6 | DyRep | $0.452 \pm 0.046$ | $0.512 \pm 0.014$ |
| 7 | DyGFormer | $0.224 \pm 0.015$ | $0.219 \pm 0.017$ | 7 | EdgeBank(unlimited) | 0.359 | 0.315 |
| 8 | DyRep | $0.220 \pm 0.030$ | $0.216 \pm 0.031$ | | | | |
| 9 | CAWN | $0.193 \pm 0.001$ | $0.200 \pm 0.001$ | | | | |
| 10 | TCL | $0.193 \pm 0.009$ | $0.199 \pm 0.007$ | | | | |
| 11 | EdgeBank(tw) | 0.025 | 0.024 | | | | |
| 12 | EdgeBank(unlimited) | 0.023 | 0.023 | | | | |

To further demonstrate the efficiency of SimpleTLP, we provide runtime comparison between our method some of the SOTAs, DyGFormer (Yu et al., 2023) and FreeDyG (Tian et al., 2024), on *US Legis* and *UN Trade* datasets in Table 11 in Appendix E.3.

## 6 RELATED WORK

Most temporal graph learning methods are designed to learn node temporal representations at a certain timestamp, and then obtain the link prediction between a pair of nodes $(u, v)$ at time $t$. They typically start by concatenating the temporal representations of $u$, $v$ at time $t$, and then process the concatenation with MLP layers to estimate the link occurrence probability. Usually, the temporal representations of nodes are obtained by aggregating intrinsic node/edge features from recent temporal neighbors and interactions, which are then processed using more complex architectures. For example, TGN (Rossi et al., 2020) and JODIE (Kumar et al., 2019) employ the recurrent architecture, TGAT (Xu et al., 2020) utilizes the self-attention mechanism, DyRep (Trivedi et al., 2019) uses Temporal Point Processes, and more recently, GraphMixer (Cong et al., 2023) leverages Mixer-MLP layers, while DyGFormer (Yu et al., 2023) harnesses the power of Transformer and FreeDyG (Tian et al., 2024) utilizes Fourier Transform to encode neighborhood co-occurrence information. From another aspect, some methods aggregate information from higher-order graph structures. For example, TGN (Rossi et al., 2020) and TGAT (Xu et al., 2020) both aggregate information from $k$-hop neighborhoods, and CAWN (Wang et al., 2021b) samples random walks and encode them to derive the node representations. Our SimpleTLP achieves effectiveness with efficiency. First, our SimpleTLP only considers 1-hop neighborhood for information aggregation and processes representations with simple MLP transformations. More importantly, SimpleTLP integrates the domain of node and edge features by integrating evolving positional encoding, which is informative to support task out-performance, and the acquisition is easy and does not really rely on heavy attention mechanisms.

## 7 CONCLUSION

In this paper, we propose a simple temporal link prediction model, SimpleTLP, which only relies on learnable positional encoding and MLPs to achieve out-performance over SOTA graph transformers in link prediction tasks. In addition to theoretical analysis of the effectiveness and efficiency, we design comprehensive experiments to demonstrate the empirical performance of SimpleTLP.

---

[1] https://tgb.complexdatalab.com/docs/leader_linkprop/#tgbl-review-v2
[2] https://tgb.complexdatalab.com/docs/leader_linkprop/#tgbl-coin-v2

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

# A  PSEUDO-CODE FOR SIMPLETLP

In this section, we provide the pseudo-code for SimpleTLP. Specifically, Algorithm. 1 summarizes the computational process of the Node-Link-Positional Encoder, Algorithm. 2 demonstrates the computation of the objective loss function, and finally, Algorithm. 3 briefly describe the training procedure of SimpleTLP as a neural architecture.

---

**Algorithm 1** Temporal Representation

---

**Require:** node $u$, time $t$
**Ensure:** Temporal Representation for $u$ at $t$

$\{\widehat{\mathbf{p}}_u^{t'_j}\}_{j=1}^L \leftarrow$ Eq. 1
$\widetilde{\mathbf{p}}_u^t \leftarrow 1$ Eq. 2
$\mathbf{h}_{u,N}^t \leftarrow n$ Eq. 4
$\mathbf{H}_{u,E}^t \leftarrow$ Eq. 5
$\mathbf{h}_{u,E}^t \leftarrow$ Eq. 5
$\mathbf{h}_{u,N\|E}^t \leftarrow$ Eq. 6
$\widehat{\mathbf{h}}_{u,P}^t \leftarrow$ Eq. 7
$\mathbf{h}_{u,P}^t \leftarrow$ Eq. 8
$\mathbf{h}_u^t \leftarrow$ Eq. 9
**return** $\mathbf{h}_u^t$

---

**Algorithm 2** Loss Function

---

**Require:** $B$ positive samples, $B$ negative samples
**Ensure:** $\mathcal{L}$

$\mathcal{L}_{lp} \leftarrow 0$
$\mathcal{L}_{pe} \leftarrow 0$
**for** $i \in [1, \ldots, B]$ **do**

   $\mathbf{h}_{u_i^{(pos)}}^t \leftarrow$ Algorithm. 1$(u_i^{(pos)}, t)$

   $\mathbf{h}_{v_i^{(pos)}}^t \leftarrow$ Algorithm. 1$(v_i^{(pos)}, t)$

   $\mathbf{h}_{u_i^{(neg)}}^t \leftarrow$ Algorithm. 1$(u_i^{(neg)}, t)$

   $\mathbf{h}_{u_i^{(neg)}}^t \leftarrow$ Algorithm. 1$(v_i^{(neg)}, t)$

   $\hat{y}_i^{(pos)} \leftarrow$ Eq. 10 with $\mathbf{h}_{u_i^{(pos)}}^t, \mathbf{h}_{v_i^{(pos)}}^t$

   $\hat{y}_i^{(neg)} \leftarrow$ Eq. 10 with $\mathbf{h}_{u_i^{(neg)}}^t, \mathbf{h}_{v_i^{(neg)}}^t$

   $\mathcal{L}_{lp} \leftarrow \mathcal{L}_{lp} + \log(\hat{y}_i^{(pos)}) + (1 - \log(\hat{y}_i^{(neg)}))$
   $\mathcal{L}_{pe} \leftarrow \mathcal{L}_{pe} + ||\widetilde{\mathbf{p}}_{u_i^{(pos)}}^{t_i} - \widetilde{\mathbf{p}}_{v_i^{(pos)}}^{t_i}||_2 - \alpha_{neg}||\widetilde{\mathbf{p}}_{u_i^{(neg)}}^{t_i} - \widetilde{\mathbf{p}}_{v_i^{(neg)}}^{t_i}||_2$

**end for**
$\mathcal{L}_{lp} \leftarrow -\mathcal{L}_{lp}/(2B)$
$\mathcal{L}_{pe} \leftarrow \mathcal{L}_{pe}/B$
$\mathcal{L} \leftarrow$ Eq. 13

---

---

**Algorithm 3** Training Pipeline of SimpleTLP

---

**Require:** Data-loader over $G$, number of epochs (# Epochs)
**Ensure:** Link Predictions
  Initialize the PE of the initial snapshot with its Laplacian PE: $\mathbf{p}^{t_0} \leftarrow$ Laplacian PE of $G^{t_0}$
  **for** epoch in $[1, \ldots, \#$ Epochs$]$ **do**
    **for** ($B$ positive samples, $B$ negative samples) in Data-loader **do**
      $\mathcal{L} \leftarrow$ Algorithm. 2($B$ positive samples, $B$ negative samples)
      Back-propagate $\mathcal{L}$
      **for** $t$ in $[t_1, \ldots, t_B]$ **do**
        **if** Finish link prediction at time $t$ ($t \in [t_1, t_B]$) **then**
          reveal new links and update positional encoding: $\forall u, \mathbf{p}_u^t \leftarrow$ Eq. 3
        **end if**
      **end for**
    **end for**
  **end for**

---

## B    PROOF OF THEOREM 1

We start our proof for Eq. 14 by first stating the following Lemma about the eigenvalues of $n$-node ring graph, denoted by $R_n$.

**Lemma 1.** *The real-valued part of Fourier basis is the eigenvector of the Laplacian of the ring graph. Specifically, for a $n$-node ring graph $R_n$, the $k-$th eigenvector of its Laplacian, $\mathbf{L}(R_n)$, is*

$$\mathbf{u}_k^{ring} = \begin{bmatrix} \cos(2\pi \cdot 0/n) \\ \cos(2\pi \cdot 1/n) \\ \ldots \\ \cos(2\pi \cdot (n-1))/n \end{bmatrix} \in \mathbb{R}^n \tag{15}$$

*and its corresponding eigenvalue is $\lambda_k^{ring} = 2 - \cos(2\pi k/n)$.*

Lemma. 1 is motivated by Lemma 5.2.1 from the lecture [3]. Based on that, we can then extend to prove our Theorem. 1 as follows.

*Proof.* For short of notation, in the rest of this proof, suppose the positional encoding of snapshots at $L+1$ distinct timestamps before $t$ are denoted as $\mathbf{p}^0, \mathbf{p}^1, \ldots, \mathbf{p}^L$, which are the Laplacian eigenvector of graph snapshots at $t_0, \ldots, t_L$, and $\widetilde{\mathbf{p}}_u^t$ and $\widetilde{\mathbf{p}}_u^{t'}$ are denoted as $\widetilde{\mathbf{p}}^{L+1}$ and $\widetilde{\mathbf{p}}^{L+2}$, respectively. Again, for short of notation, we omit the subscript $u$ in the positional encoding notations. In addition, we denote $\mathbf{L}(R_n)$ as the (un-normalized) Laplacian of a $n$-node ring graph, $R_n$.

We start by giving the definition of "slow-changing" network. We suppose that our network is "slow-changing", in terms of graph topology, thus, we can assume the eigenvector of the Laplacian does not vary across timestamps, i.e., $\mathbf{p}^0 \approx \cdots \approx \mathbf{p}^L$, so now we focus on bounding the left-hand side of Eq. 14.

---

[3] https://www.cs.yale.edu/homes/spielman/561/lect05-15.pdf

We start by considering the following equation:

$$
\widetilde{\mathbf{p}}^{L+1} - \widetilde{\mathbf{p}}^{L+2} + \widetilde{\mathbf{p}}^{L+1} - \widetilde{\mathbf{p}}^{L} + \left( \begin{bmatrix} \mathbf{p}^0 - \mathbf{p}^L & 0 & 0 & \cdots & \mathbf{p}^{L+1} - \mathbf{p}^1 \end{bmatrix} \right) \mathbf{F} \mathbf{W}_{filter} \mathbf{F}^{-1} \begin{bmatrix} \alpha_1 \\ \cdots \\ \alpha_{t-1} \end{bmatrix} =
$$

$$
= \left( \begin{bmatrix} \mathbf{p}^1 & \mathbf{p}^2 & \cdots & \mathbf{p}^L \end{bmatrix} - \begin{bmatrix} \mathbf{p}^2 & \mathbf{p}^3 & \cdots & \mathbf{p}^{L+1} \end{bmatrix} \right.
$$
$$
+ \begin{bmatrix} \mathbf{p}^1 & \mathbf{p}^2 & \cdots & \mathbf{p}^L \end{bmatrix} - \begin{bmatrix} \mathbf{p}^0 & \mathbf{p}^1 & \cdots & \mathbf{p}^{L-1} \end{bmatrix}
$$
$$
+ \begin{bmatrix} \mathbf{p}^0 - \mathbf{p}^L & 0 & 0 & \cdots & \mathbf{p}^{L+1} - \mathbf{p}^1 \end{bmatrix} \right) \mathbf{F} \mathbf{W}_{filter} \mathbf{F}^{-1} \begin{bmatrix} \alpha_1 \\ \cdots \\ \alpha_{t-1} \end{bmatrix}
$$

$$
= \left( \begin{bmatrix} \mathbf{p}^1 - \mathbf{p}^2 + \mathbf{p}^1 - \mathbf{p}^0 & \cdots & \mathbf{p}^i - \mathbf{p}^{i+1} + \mathbf{p}^i - \mathbf{p}^{i-1} & \cdots & \mathbf{p}^L - \mathbf{p}^{L+1} + \mathbf{p}^L - \mathbf{p}^{L-1} \end{bmatrix} \right.
$$
$$
+ \begin{bmatrix} \mathbf{p}^0 - \mathbf{p}^L & 0 & 0 & \cdots & \mathbf{p}^{L+1} - \mathbf{p}^1 \end{bmatrix} \right) \mathbf{F} \mathbf{W}_{filter} \mathbf{F}^{-1} \begin{bmatrix} \alpha_1 \\ \cdots \\ \alpha_{t-1} \end{bmatrix}
$$

$$
= \left( \begin{bmatrix} \mathbf{p}^1 - \mathbf{p}^2 + \mathbf{p}^1 - \mathbf{p}^L & \cdots & \mathbf{p}^i - \mathbf{p}^{i+1} + \mathbf{p}^i - \mathbf{p}^{i-1} & \cdots & \mathbf{p}^L - \mathbf{p}^1 + \mathbf{p}^L - \mathbf{p}^{L-1} \end{bmatrix} \right) \mathbf{F} \mathbf{W}_{filter} \mathbf{F}^{-1} \begin{bmatrix} \mathbf{p}^1 \\ \cdots \\ \mathbf{p}^L \end{bmatrix}
$$

$$
= \begin{bmatrix} \mathbf{p}^1 & \cdots & \mathbf{p}^L \end{bmatrix} L(R_L) \mathbf{F} \mathbf{W}_{filter} \mathbf{F}^{-1} \begin{bmatrix} \alpha_1 \\ \cdots \\ \alpha_L \end{bmatrix}
\tag{16}
$$

The eigen-decomposition of $L(R_L)$ is $L(R_L) = F \Lambda F^{-1}$, where $\Lambda$ is a diagonal matrix with the $i-$th entri is $\lambda_i^{ring}$. Thus

$$
\begin{bmatrix} \mathbf{p}^1 & \cdots & \mathbf{p}^L \end{bmatrix} L(R_L) \mathbf{F} \mathbf{W}_{filter} \mathbf{F}^{-1} \begin{bmatrix} \alpha_1 \\ \cdots \\ \alpha_L \end{bmatrix}
$$
$$
= \begin{bmatrix} \mathbf{p}^1 & \cdots & \mathbf{p}^L \end{bmatrix} \mathbf{F} \Lambda \mathbf{F}^{-1} \mathbf{F} \mathbf{W}_{filter} \mathbf{F}^{-1} \begin{bmatrix} \alpha_1 \\ \cdots \\ \alpha_L \end{bmatrix}
\tag{17}
$$
$$
= \begin{bmatrix} \mathbf{p}^1 & \cdots & \mathbf{p}^L \end{bmatrix} \mathbf{F} (\Lambda \mathbf{W}_{filter}) \mathbf{F}^{-1} \begin{bmatrix} \alpha_1 \\ \cdots \\ \alpha_L \end{bmatrix}
$$

For simplicity, let $\mathbf{W}_{filter}$ be a diagonal matrix, thus $(\Lambda \mathbf{W}_{filter})$ is a diagonal matrix.
Therefore, we obtain

$$
\begin{bmatrix} \mathbf{p}^1 & \cdots & \mathbf{p}^L \end{bmatrix} \mathbf{F} (\Lambda \mathbf{W}_{filter}) \mathbf{F}^{-1} \begin{bmatrix} \alpha_1 \\ \cdots \\ \alpha_L \end{bmatrix}
$$
$$
= \begin{bmatrix} \mathbf{p}^1 & \cdots & \mathbf{p}^L \end{bmatrix} \sum_{i=1}^{L} (\Lambda W_{filter})_i \mathbf{F}_{:,i} (\mathbf{F}_{:,i}^{-1})^T \begin{bmatrix} \alpha_1 \\ \cdots \\ \alpha_L \end{bmatrix}
\tag{18}
$$

Thus,

$$\widetilde{\mathbf{p}}^{L+1} - \widetilde{\mathbf{p}}^{L+2} + \widetilde{\mathbf{p}}^{L+1} - \widetilde{\mathbf{p}}^{L} + \Big( \begin{bmatrix} \mathbf{p}^0 - \mathbf{p}^L & 0 & 0 & \ldots & \mathbf{p}^{L+1} - \mathbf{p}^1 \end{bmatrix} \Big) \mathbf{F} \mathbf{W}_{filter} \mathbf{F}^{-1} \begin{bmatrix} \alpha_1 \\ \ldots \\ \alpha_{t-1} \end{bmatrix} =$$

$$= \begin{bmatrix} \mathbf{p}^1 & \ldots & \mathbf{p}^L \end{bmatrix} \sum_{i=1}^{L} (\Lambda W_{filter})_i \mathbf{F}_{:,i} (\mathbf{F}_{:,i}^{-1})^T \begin{bmatrix} \alpha_1 \\ \ldots \\ \alpha_L \end{bmatrix}$$

$$= (\sum_{i=1}^{L} (\Lambda \mathbf{W}_{filter})_i)(\sum_{j=1}^{L} \alpha_j \mathbf{p}^j)$$

$$\Rightarrow \widetilde{\mathbf{p}}^{L+1} - \widetilde{\mathbf{p}}^{L+2} = \widetilde{\mathbf{p}}^{L} - \widetilde{\mathbf{p}}^{L+1} - \Big( \begin{bmatrix} \mathbf{p}^0 - \mathbf{p}^L & 0 & 0 & \ldots & \mathbf{p}^{L+1} - \mathbf{p}^1 \end{bmatrix} \Big) \mathbf{F} \mathbf{W}_{filter} \mathbf{F}^{-1} \begin{bmatrix} \alpha_1 \\ \ldots \\ \alpha_{t-1} \end{bmatrix}$$

$$+ (\sum_{i=1}^{L} (\Lambda \mathbf{W}_{filter})_i)(\sum_{j=1}^{L} \alpha_j \mathbf{p}^j) \tag{19}$$

Therefore,

$$||\widetilde{\mathbf{p}}^{L+1} - \widetilde{\mathbf{p}}^{L+2}||_2 \leq ||(\sum_{i=1}^{L} (\Lambda \mathbf{W}_{filter})_i)(\sum_{j=1}^{L} \alpha_j \mathbf{p}^j)||_2 + ||\widetilde{\mathbf{p}}^{L+1} - \widetilde{\mathbf{p}}^{L}||_2$$

$$\leq ||(\sum_{i=1}^{L} (\Lambda \mathbf{W}_{filter})_i)(\sum_{j=1}^{L} \alpha_j \mathbf{p}^j)||_2 + ||(\sum_{i=1}^{L} (\Lambda \mathbf{W}_{filter})_i)(\sum_{j=0}^{L-1} \alpha_j \mathbf{p}^j)||_2 + ||\widetilde{\mathbf{p}}^{L} - \widetilde{\mathbf{p}}^{L-1}||_2 \tag{20}$$

$$\leq \ldots$$

$$\leq (\sum_{i=1}^{L} (\Lambda \mathbf{W}_{filter})_i)||(\sum_{j=1}^{L} \alpha_j \mathbf{p}^j)||_2 + (\sum_{i=1}^{L} (\Lambda \mathbf{W}_{filter})_i)|| \sum_{j=0}^{L-1} (L-1-j)\alpha_{j+1} \mathbf{p}^j||_2$$

where the first inequality holds due to the assumption of "slow-changing" network.

For simplicity, we assume $\alpha_1, \alpha_2, \ldots, \alpha_{t-1}$ and the length of positional encoding $P^{(t)}$ is normalized, i.e $\alpha_1 + \cdots + \alpha_{t-1} = 1$

Thus,

$$(20) \leq \Big( \sum_{i=1}^{L} |(\Lambda \mathbf{W}_{filter})_i| \Big) \cdot (2L - 2) \tag{21}$$

Since $L \ll N$, thus choosing $t$ small enough and choosing the filter such that $\sum_{i=1}^{L} (\Lambda \mathbf{W}_{filter})_i < 1$ shows that the past positional encoding can be a well approximation of the positional encoding of the future snapshot.

$\square$

## C   MORE COMPARISON WITH RELATED WORKS

In this section, we provide comparative discussion and experimental comparison between our method SimpleTLP and PINT (Souza et al., 2022), a method for temporal link prediction that also leverage positional features.

To be more specific, PINT (Souza et al., 2022) proposes relative positional features that count the number of temporal walks of a given length between 2 nodes. As stated in PINT (Souza

et al., 2022), temporal walks are defined as follows. An $(L-1)$-length temporal walk is $W = \{(w_1, t_1), (w_2, t_2), \ldots, (w_L, t_L)\}$, where $t_1 > t_2 > \ldots t_L$ and $(w_{i-1}, w_i, t_i)$ is an interaction in the given temporal graph. In summary, PINT proposes a relative positional encoding that represents pair-wise node "distance" (here the "distance" is the number of $L$-length temporal walks, assuming $L$ is given). On the other hand, our SimpleTLP learnable positional encodings (LPE) fall into the category of global positional encodings, which encode the position of the node relative to the global structure of the graph.

Table 5: Performance comparison between PINT and SimpleTLP in *transductive setting* and *inductive setting* with random negative sampling strategy

| Dataset | Method | Transductive AP | Transductive ROC-AUC | Inductive AP | Inductive ROC-AUC |
|---------|--------|-----------------|----------------------|--------------|-------------------|
| Enron | PINT | 81.60 ± 0.67 | 84.50 ± 0.60 | 67.01 ± 2.76 | 70.41 ± 2.34 |
| | SimpleTLP (Ours) | **93.96 ± 0.36** | **95.98 ± 0.23** | **89.17 ± 0.88** | **92.30 ± 0.57** |
| UCI | PINT | 95.77 ± 0.11 | 94.89 ± 0.15 | 94.14 ± 0.05 | 92.51 ± 0.07 |
| | SimpleTLP (Ours) | **96.67 ± 0.47** | **97.62 ± 0.23** | **94.60 ± 0.41** | **95.88 ± 0.17** |

Next, we present the empirical comparision between SimpleTLPand PINT (Souza et al., 2022) in Table 5. As shown in the Table 5, our SimpleTLP outperforms PINT in all experimental settings (transductive and inductive). Regarding PINT's empirical evaluation, we adopt PINT's official implementation (via the Github link provided in the (Souza et al., 2022)) and use the exact hyperparameters presented in the PINT paper to reproduce their results on these datasets. Moreover, PINT (Souza et al., 2022) and SimpleTLP use the same data split.

# D DATASETS

Table 6: Statistics of the datasets.

| Datasets | Domains | #Nodes | #Links | #N&L Feat | Bipartite | Duration | Unique Steps | Time Granularity |
|----------|---------|--------|--------|-----------|-----------|----------|--------------|------------------|
| Wikipedia | Social | 9,227 | 157,474 | – & 172 | True | 1 month | 152,757 | Unix timestamps |
| Reddit | Social | 10,984 | 672,447 | – & 172 | True | 1 month | 669,065 | Unix timestamps |
| MOOC | Interaction | 7,144 | 411,749 | – & 4 | True | 17 months | 345,600 | Unix timestamps |
| LastFM | Interaction | 1,980 | 1,293,103 | – & – | True | 1 month | 1,283,614 | Unix timestamps |
| Enron | Social | 184 | 125,235 | – & – | False | 3 years | 22,632 | Unix timestamps |
| Social Evo. | Proximity | 74 | 2,099,519 | – & 2 | False | 8 months | 565,932 | Unix timestamps |
| UCI | Social | 1,899 | 59,835 | – & – | False | 196 days | 58,911 | Unix timestamps |
| Flights | Transport | 13,169 | 1,927,145 | – & 1 | False | 4 months | 122 | days |
| Can. Parl. | Politics | 734 | 74,478 | – & 1 | False | 14 years | 14 | years |
| US Legis. | Politics | 225 | 60,396 | – & 1 | False | 12 congresses | 12 | congresses |
| UN Trade | Economics | 255 | 507,497 | – & 1 | False | 32 years | 32 | years |
| UN Vote | Politics | 201 | 1,035,742 | – & 1 | False | 72 years | 72 | years |
| Contact | Proximity | 692 | 2,426,279 | – & 1 | False | 1 month | 8,064 | 5 minutes |

Here, in Table 6, we report a detailed regarding 13 datasets in our empirical assessment, which includes the domain, number of nodes, number of links, dimension of raw node and edge features. The statistics of Benchmark TGB 2.0 are available at `https://tgb.complexdatalab.com/docs/linkprop/`.

# E MORE EXPERIMENTAL RESULTS

## E.1 EMPIRICAL RESULTS FOR DIFFERENT NEGATIVE SAMPLING STRATEGIES

We provide empirical results for (1) historical NSS for transductive setting in Table 7 and inductive setting in Table 8, (2) inductive NSS under transductive setting in Table 9 and inductive setting in Table 10.

Table 7: Performance comparison in the *transductive setting* with *historical negative sampling strategy*.

| Metric | Datasets | JODIE | DyRep | TGAT | TGN | CAWN | TCL | GraphMixer | DyGFormer | FreeDyG | SimpleTLP |
|---|---|---|---|---|---|---|---|---|---|---|---|
| AP | Wikipedia | 83.01 ± 0.66 | 79.93 ± 0.56 | 87.38 ± 0.22 | 86.86 ± 0.33 | 71.21 ± 1.67 | 89.05 ± 0.39 | 90.90 ± 0.10 | 82.23 ± 2.54 | 91.59 ± 0.57 | 99.10 ± 0.51 |
| | Reddit | 80.03 ± 0.36 | 79.83 ± 0.31 | 79.55 ± 0.20 | 81.22 ± 0.61 | 80.82 ± 0.45 | 77.14 ± 0.16 | 78.44 ± 0.18 | 81.57 ± 0.67 | 85.67 ± 1.01 | 93.25 ± 1.13 |
| | MOOC | 78.94 ± 1.25 | 75.60 ± 1.12 | 82.19 ± 0.62 | 87.06 ± 1.93 | 74.05 ± 0.95 | 77.06 ± 0.41 | 77.77 ± 0.92 | 85.85 ± 0.66 | 86.71 ± 0.81 | 96.52 ± 1.58 |
| | LastFM | 74.35 ± 3.81 | 74.92 ± 2.46 | 71.59 ± 0.24 | 76.87 ± 4.64 | 69.86 ± 0.43 | 59.30 ± 2.31 | 72.47 ± 0.49 | 81.57 ± 0.48 | 79.71 ± 0.51 | 94.88 ± 1.35 |
| | Enron | 69.85 ± 2.70 | 71.19 ± 2.76 | 64.07 ± 1.05 | 73.91 ± 1.76 | 64.73 ± 0.36 | 70.66 ± 0.39 | 77.98 ± 0.92 | 75.63 ± 0.73 | 78.87 ± 0.82 | 95.42 ± 0.53 |
| | Social Evo. | 87.44 ± 6.78 | 93.29 ± 0.43 | 95.01 ± 0.44 | 94.45 ± 0.56 | 85.53 ± 0.38 | 94.74 ± 0.31 | 94.93 ± 0.31 | 97.38 ± 0.14 | 97.79 ± 0.23 | 90.19 ± 2.01 |
| | UCI | 75.24 ± 5.80 | 55.10 ± 3.14 | 68.27 ± 1.37 | 80.43 ± 2.12 | 65.30 ± 0.43 | 80.25 ± 2.74 | 84.11 ± 1.35 | 82.17 ± 0.82 | 86.10 ± 1.19 | 80.76 ± 1.88 |
| | Flights | 66.48 ± 2.59 | 67.61 ± 0.99 | 72.38 ± 0.18 | 66.70 ± 1.64 | 64.72 ± 0.97 | 70.68 ± 0.24 | 71.47 ± 0.26 | 66.59 ± 0.49 | 66.13 ± 1.23 | 82.02 ± 2.39 |
| | Can. Parl. | 51.79 ± 0.63 | 63.31 ± 1.23 | 67.13 ± 0.84 | 68.42 ± 3.07 | 66.53 ± 2.77 | 65.93 ± 3.00 | 74.34 ± 0.87 | 97.00 ± 0.31 | 72.22 ± 2.47 | 75.03 ± 1.76 |
| | US Legis. | 51.71 ± 5.76 | 86.88 ± 2.25 | 62.14 ± 6.60 | 74.00 ± 7.57 | 68.82 ± 8.23 | 80.53 ± 3.95 | 81.65 ± 1.02 | 85.30 ± 3.88 | 69.94 ± 0.59 | 62.66 ± 3.54 |
| | UN Trade | 61.39 ± 1.83 | 59.19 ± 1.07 | 55.74 ± 0.91 | 58.44 ± 5.51 | 55.71 ± 0.38 | 55.90 ± 1.17 | 57.05 ± 1.22 | 64.41 ± 1.40 | - | 80.00 ± 1.55 |
| | UN Vote | 70.02 ± 0.81 | 69.30 ± 1.12 | 52.96 ± 2.14 | 69.37 ± 3.93 | 51.26 ± 0.04 | 52.30 ± 2.35 | 51.20 ± 1.60 | 60.84 ± 1.58 | 50.61 ± 2.46 | 61.43 ± 1.07 |
| | Contact | 95.31 ± 2.13 | 96.39 ± 0.20 | 96.05 ± 0.52 | 93.05 ± 2.35 | 84.16 ± 0.49 | 93.86 ± 0.21 | 93.36 ± 0.41 | 97.57 ± 0.06 | 97.32 ± 0.48 | 80.24 ± 1.98 |
| | Avg. Rank | 6.62 | 6.0 | 6.31 | 4.85 | 8.62 | 6.62 | 5.08 | 3.38 | 4.23 | 3.31 |
| ROC-AUC | Wikipedia | 80.77 ± 0.73 | 77.74 ± 0.33 | 82.87 ± 0.22 | 82.74 ± 0.32 | 67.84 ± 0.64 | 85.76 ± 0.46 | 87.68 ± 0.17 | 78.80 ± 1.95 | 82.78 ± 0.30 | 99.21 ± 0.48 |
| | Reddit | 80.52 ± 0.32 | 80.15 ± 0.18 | 79.33 ± 0.16 | 81.11 ± 0.19 | 80.27 ± 0.30 | 76.49 ± 0.16 | 77.80 ± 0.12 | 80.54 ± 0.29 | 85.92 ± 0.10 | 96.40 ± 0.59 |
| | MOOC | 82.75 ± 0.83 | 81.06 ± 0.94 | 80.81 ± 0.67 | 88.00 ± 1.80 | 71.57 ± 1.07 | 72.09 ± 0.56 | 76.68 ± 1.40 | 87.04 ± 0.35 | 88.32 ± 0.99 | 97.69 ± 0.94 |
| | LastFM | 75.22 ± 2.36 | 74.65 ± 1.98 | 64.27 ± 0.26 | 77.97 ± 3.04 | 67.88 ± 0.24 | 47.24 ± 3.13 | 64.21 ± 0.73 | 78.78 ± 0.35 | 73.53 ± 0.12 | 97.66 ± 1.96 |
| | Enron | 75.39 ± 2.37 | 74.69 ± 3.55 | 61.85 ± 1.43 | 77.09 ± 2.22 | 65.10 ± 0.34 | 67.95 ± 0.88 | 75.27 ± 1.14 | 76.55 ± 0.52 | 75.74 ± 0.72 | 95.79 ± 0.33 |
| | Social Evo. | 90.06 ± 3.15 | 93.12 ± 0.34 | 93.08 ± 0.59 | 94.71 ± 0.53 | 87.43 ± 0.15 | 93.44 ± 0.68 | 94.39 ± 0.31 | 97.28 ± 0.07 | 97.42 ± 0.02 | 93.60 ± 2.27 |
| | UCI | 78.64 ± 3.50 | 57.91 ± 3.12 | 58.89 ± 1.57 | 77.25 ± 2.68 | 57.86 ± 0.15 | 72.25 ± 3.46 | 77.54 ± 2.02 | 76.97 ± 0.24 | 80.38 ± 0.26 | 90.59 ± 1.60 |
| | Flights | 68.97 ± 1.87 | 69.43 ± 0.90 | 72.20 ± 0.16 | 68.39 ± 0.95 | 66.11 ± 0.71 | 70.57 ± 0.18 | 70.37 ± 0.23 | 68.09 ± 0.43 | 67.83 ± 1.46 | 92.84 ± 1.21 |
| | Can. Parl. | 62.44 ± 1.11 | 70.16 ± 1.70 | 70.86 ± 0.94 | 73.23 ± 3.08 | 72.06 ± 3.94 | 69.95 ± 3.70 | 79.03 ± 1.01 | 97.61 ± 0.40 | 77.59 ± 6.22 | 87.22 ± 1.85 |
| | US Legis. | 67.47 ± 6.40 | 91.44 ± 1.18 | 73.47 ± 5.25 | 83.53 ± 4.53 | 78.62 ± 7.46 | 83.97 ± 3.71 | 85.17 ± 0.70 | 90.77 ± 1.96 | 63.49 ± 13.04 | 76.64 ± 3.38 |
| | UN Trade | 68.92 ± 1.40 | 64.36 ± 1.40 | 60.37 ± 0.68 | 63.93 ± 5.41 | 63.09 ± 0.74 | 61.43 ± 1.04 | 63.20 ± 1.54 | 73.86 ± 1.13 | - | 87.67 ± 1.80 |
| | UN Vote | 76.84 ± 1.01 | 74.72 ± 1.43 | 53.95 ± 3.15 | 73.40 ± 5.20 | 51.27 ± 0.33 | 52.29 ± 2.39 | 52.61 ± 1.44 | 64.27 ± 1.78 | 52.76 ± 2.85 | 72.92 ± 1.73 |
| | Contact | 96.35 ± 0.92 | 96.00 ± 0.23 | 95.39 ± 0.43 | 93.76 ± 1.29 | 83.06 ± 0.32 | 93.34 ± 0.19 | 93.14 ± 0.34 | 97.17 ± 0.05 | 97.12 ± 0.24 | 90.99 ± 1.80 |
| | Avg. Rank | 5.38 | 5.69 | 6.92 | 4.31 | 8.54 | 7.15 | 5.69 | 3.69 | 4.92 | 2.69 |

Table 8: Performance comparison in the *inductive setting* with *historical negative sampling strategy*.

| Metric | Datasets | JODIE | DyRep | TGAT | TGN | CAWN | TCL | GraphMixer | DyGFormer | FreeDyG | SimpleTLP |
|---|---|---|---|---|---|---|---|---|---|---|---|
| AP | Wikipedia | 68.69 ± 0.39 | 62.18 ± 1.27 | 84.17 ± 0.22 | 81.76 ± 0.32 | 67.27 ± 1.63 | 82.20 ± 2.18 | 87.60 ± 0.30 | 71.42 ± 4.43 | 82.78 ± 0.30 | 98.45 ± 0.88 |
| | Reddit | 62.34 ± 0.54 | 61.60 ± 0.72 | 63.47 ± 0.36 | 64.85 ± 0.85 | 63.67 ± 0.41 | 60.83 ± 0.25 | 64.50 ± 0.26 | 65.37 ± 0.60 | 66.02 ± 0.41 | 85.74 ± 2.46 |
| | MOOC | 63.22 ± 1.55 | 62.93 ± 1.24 | 76.73 ± 0.29 | 77.07 ± 3.41 | 74.68 ± 0.68 | 74.27 ± 0.53 | 74.00 ± 0.97 | 80.82 ± 0.30 | 81.63 ± 0.33 | 95.17 ± 2.78 |
| | LastFM | 70.39 ± 4.31 | 71.45 ± 1.76 | 71.27 ± 0.25 | 66.65 ± 6.11 | 71.33 ± 0.47 | 69.82 ± 0.32 | 76.42 ± 0.22 | 76.35 ± 0.52 | 77.28 ± 0.21 | 92.96 ± 1.08 |
| | Enron | 65.86 ± 3.71 | 62.08 ± 2.27 | 61.40 ± 1.31 | 62.91 ± 1.16 | 60.70 ± 0.36 | 67.11 ± 0.62 | 72.37 ± 1.37 | 67.07 ± 0.62 | 73.01 ± 0.88 | 89.37 ± 1.01 |
| | Social Evo. | 88.51 ± 0.87 | 88.72 ± 1.10 | 93.97 ± 0.54 | 90.66 ± 1.62 | 79.83 ± 0.38 | 94.10 ± 0.31 | 94.01 ± 0.47 | 96.82 ± 0.16 | 96.69 ± 0.14 | 85.13 ± 1.05 |
| | UCI | 63.11 ± 2.27 | 52.47 ± 2.06 | 70.52 ± 0.93 | 70.78 ± 0.78 | 64.54 ± 0.47 | 76.71 ± 1.00 | 81.66 ± 0.49 | 72.13 ± 1.87 | 82.35 ± 0.39 | 75.97 ± 1.13 |
| | Flights | 61.01 ± 1.65 | 62.83 ± 1.31 | 64.72 ± 0.36 | 59.31 ± 1.43 | 56.82 ± 0.57 | 64.50 ± 0.25 | 65.28 ± 0.24 | 57.11 ± 0.21 | 56.72 ± 1.30 | 71.54 ± 2.12 |
| | Can. Parl. | 52.60 ± 0.88 | 52.28 ± 0.52 | 56.72 ± 0.47 | 54.42 ± 0.77 | 57.14 ± 0.07 | 55.71 ± 0.74 | 55.84 ± 0.73 | 87.40 ± 0.85 | 52.15 ± 0.84 | 68.91 ± 1.63 |
| | US Legis. | 52.94 ± 2.11 | 62.10 ± 1.41 | 51.83 ± 3.95 | 61.18 ± 1.10 | 55.56 ± 1.71 | 53.87 ± 1.41 | 52.03 ± 1.02 | 56.31 ± 3.46 | 53.63 ± 4.38 | 49.51 ± 1.09 |
| | UN Trade | 55.46 ± 1.19 | 55.49 ± 0.84 | 55.28 ± 0.71 | 52.80 ± 3.19 | 55.00 ± 0.38 | 55.76 ± 1.03 | 54.94 ± 0.97 | 53.20 ± 1.07 | - | 79.09 ± 1.86 |
| | UN Vote | 61.04 ± 1.30 | 60.22 ± 1.78 | 53.05 ± 3.10 | 63.74 ± 3.00 | 47.98 ± 0.84 | 48.09 ± 0.43 | 52.63 ± 1.26 | 50.23 ± 1.68 | 61.89 ± 1.28 | - |
| | Contact | 90.42 ± 2.34 | 89.22 ± 0.66 | 94.15 ± 0.45 | 88.13 ± 1.50 | 74.20 ± 0.80 | 90.44 ± 0.17 | 89.91 ± 0.36 | 93.56 ± 0.52 | 93.26 ± 1.42 | 79.08 ± 2.24 |
| | Avg. Rank | 6.85 | 6.85 | 5.31 | 5.85 | 7.54 | 5.23 | 4.92 | 4.38 | 4.77 | 3.31 |
| ROC-AUC | Wikipedia | 61.86 ± 0.53 | 57.54 ± 1.09 | 78.38 ± 0.20 | 75.75 ± 0.29 | 62.04 ± 0.65 | 79.79 ± 0.96 | 82.87 ± 0.21 | 68.33 ± 2.82 | 82.08 ± 0.32 | 98.59 ± 0.82 |
| | Reddit | 61.69 ± 0.39 | 60.45 ± 0.37 | 64.43 ± 0.27 | 64.55 ± 0.50 | 64.94 ± 0.21 | 61.43 ± 0.26 | 64.27 ± 0.13 | 64.81 ± 0.25 | 66.79 ± 0.31 | 92.04 ± 1.87 |
| | MOOC | 64.48 ± 1.64 | 64.23 ± 1.29 | 74.08 ± 0.27 | 77.69 ± 3.55 | 71.68 ± 0.94 | 74.88 ± 0.32 | 72.53 ± 0.84 | 80.77 ± 0.63 | 81.52 ± 0.37 | 96.85 ± 1.59 |
| | LastFM | 68.44 ± 3.26 | 68.79 ± 1.08 | 69.89 ± 0.28 | 66.99 ± 5.62 | 67.69 ± 0.24 | 55.88 ± 1.85 | 70.07 ± 0.20 | 70.73 ± 0.37 | 72.63 ± 0.16 | 96.34 ± 1.68 |
| | Enron | 65.32 ± 3.57 | 61.50 ± 2.50 | 57.84 ± 2.18 | 62.68 ± 1.09 | 62.25 ± 0.40 | 64.06 ± 1.02 | 68.20 ± 1.62 | 65.78 ± 0.42 | 70.09 ± 0.65 | 92.14 ± 0.70 |
| | Social Evo. | 88.53 ± 0.55 | 87.93 ± 1.05 | 91.87 ± 0.72 | 92.10 ± 1.22 | 83.54 ± 0.25 | 93.62 ± 0.35 | 96.91 ± 0.09 | 96.94 ± 0.17 | 90.08 ± 2.91 |  |
| | UCI | 60.24 ± 1.94 | 51.25 ± 2.37 | 62.32 ± 1.18 | 62.69 ± 0.90 | 56.39 ± 0.10 | 70.46 ± 1.94 | 75.98 ± 0.84 | 65.55 ± 1.01 | 76.01 ± 0.75 | 84.63 ± 1.23 |
| | Flights | 60.72 ± 1.29 | 61.99 ± 1.39 | 63.38 ± 0.26 | 59.66 ± 1.04 | 56.58 ± 0.44 | 63.48 ± 0.23 | 63.30 ± 0.19 | 56.05 ± 0.21 | 55.20 ± 1.16 | 85.60 ± 1.47 |
| | Can. Parl. | 51.62 ± 1.00 | 52.38 ± 0.46 | 58.30 ± 0.61 | 55.64 ± 0.54 | 60.11 ± 0.48 | 57.30 ± 1.03 | 56.68 ± 1.20 | 88.68 ± 0.74 | 51.20 ± 1.99 | 80.73 ± 1.09 |
| | US Legis. | 58.12 ± 2.94 | 67.94 ± 0.98 | 49.99 ± 4.88 | 64.87 ± 1.65 | 54.41 ± 1.31 | 52.12 ± 2.13 | 49.28 ± 0.86 | 56.57 ± 3.22 | 53.75 ± 5.92 | 55.74 ± 1.53 |
| | UN Trade | 58.73 ± 1.19 | 57.90 ± 1.33 | 59.74 ± 0.59 | 55.61 ± 3.54 | 60.95 ± 0.80 | 61.12 ± 0.97 | 59.88 ± 1.17 | 58.46 ± 1.65 | - | 86.26 ± 1.78 |
| | UN Vote | 65.16 ± 1.28 | 63.98 ± 2.12 | 51.73 ± 4.12 | 68.59 ± 3.11 | 48.01 ± 1.77 | 54.66 ± 2.11 | 45.49 ± 0.42 | 53.85 ± 2.02 | 48.98 ± 2.44 | 72.98 ± 1.22 |
| | Contact | 90.80 ± 1.18 | 88.88 ± 0.68 | 93.76 ± 0.41 | 88.84 ± 1.39 | 74.79 ± 0.37 | 90.37 ± 0.16 | 90.04 ± 0.29 | 94.14 ± 0.26 | 94.27 ± 0.77 | 88.69 ± 2.24 |
| | Avg. Rank | 6.54 | 7.46 | 5.77 | 5.92 | 7.08 | 5.54 | 5.23 | 4.38 | 4.62 | 2.46 |

Table 9: Performance comparison in the *transductive setting* with *inductive negative sampling strategy*.

| Metric | Datasets | JODIE | DyRep | TGAT | TGN | CAWN | TCL | GraphMixer | DyGFormer | FreeDyG | SimpleTLP |
|---|---|---|---|---|---|---|---|---|---|---|---|
| AP | Wikipedia | 75.65 ± 0.79 | 70.21 ± 1.58 | 87.00 ± 0.16 | 85.62 ± 0.44 | 74.06 ± 2.62 | 86.76 ± 0.72 | 88.59 ± 0.17 | 78.29 ± 5.38 | 90.05 ± 0.79 | 98.66 ± 0.94 |
| | Reddit | 86.98 ± 0.16 | 86.30 ± 0.26 | 89.59 ± 0.24 | 88.10 ± 0.24 | 91.67 ± 0.24 | 87.45 ± 0.29 | 85.26 ± 0.11 | 91.11 ± 0.40 | 90.74 ± 0.17 | 98.15 ± 0.48 |
| | MOOC | 65.23 ± 2.19 | 61.66 ± 0.95 | 75.95 ± 0.64 | 77.50 ± 2.91 | 73.51 ± 0.94 | 74.65 ± 0.54 | 74.27 ± 0.92 | 81.24 ± 0.69 | 83.01 ± 0.87 | 97.49 ± 0.99 |
| | LastFM | 62.67 ± 4.49 | 64.41 ± 2.70 | 71.13 ± 0.17 | 65.95 ± 5.98 | 67.48 ± 0.77 | 58.21 ± 0.89 | 68.12 ± 0.33 | 73.97 ± 0.50 | 72.19 ± 0.24 | 93.92 ± 1.10 |
| | Enron | 68.96 ± 0.98 | 67.79 ± 1.53 | 63.94 ± 1.36 | 70.89 ± 2.72 | 75.15 ± 0.58 | 71.29 ± 0.32 | 75.01 ± 0.79 | 77.41 ± 0.89 | 77.81 ± 0.65 | 98.98 ± 0.53 |
| | Social Evo. | 89.82 ± 4.11 | 93.28 ± 0.48 | 94.84 ± 0.44 | 95.13 ± 0.56 | 88.32 ± 0.27 | 94.90 ± 0.36 | 94.72 ± 0.33 | 97.68 ± 0.10 | 97.57 ± 0.15 | 90.53 ± 1.88 |
| | UCI | 65.99 ± 1.40 | 54.79 ± 1.76 | 68.67 ± 0.84 | 70.94 ± 0.71 | 64.61 ± 0.48 | 76.01 ± 1.11 | 80.10 ± 0.51 | 72.25 ± 1.71 | 82.35 ± 0.73 | 74.08 ± 1.84 |
| | Flights | 69.07 ± 4.02 | 70.57 ± 1.82 | 75.48 ± 0.26 | 71.09 ± 2.72 | 69.18 ± 1.52 | 74.62 ± 0.18 | 74.87 ± 0.21 | 70.92 ± 1.78 | 65.56 ± 1.10 | 80.75 ± 1.51 |
| | Can. Parl. | 48.42 ± 0.66 | 58.61 ± 0.86 | 68.82 ± 1.21 | 65.34 ± 2.87 | 67.75 ± 1.00 | 65.85 ± 1.75 | 69.48 ± 0.63 | 95.44 ± 0.57 | 59.83 ± 5.15 | 73.79 ± 1.26 |
| | US Legis. | 50.27 ± 5.13 | 83.44 ± 1.16 | 61.91 ± 5.82 | 67.57 ± 6.47 | 65.81 ± 8.52 | 78.15 ± 3.34 | 79.63 ± 0.84 | 81.25 ± 3.62 | 51.64 ± 5.39 | 57.67 ± 3.10 |
| | UN Trade | 60.42 ± 1.48 | 60.19 ± 1.24 | 60.61 ± 1.24 | 61.04 ± 6.01 | 62.54 ± 0.67 | 61.06 ± 1.74 | 60.15 ± 1.29 | 55.79 ± 1.02 | - | 80.96 ± 1.12 |
| | UN Vote | 67.79 ± 1.46 | 67.53 ± 1.98 | 52.89 ± 1.61 | 67.63 ± 2.67 | 52.19 ± 0.34 | 50.62 ± 0.82 | 51.60 ± 0.73 | 51.91 ± 0.84 | 51.14 ± 1.56 | 59.42 ± 3.73 |
| | Contact | 93.43 ± 1.78 | 94.18 ± 0.10 | 94.35 ± 0.48 | 90.18 ± 3.28 | 89.31 ± 0.27 | 91.35 ± 0.21 | 90.87 ± 0.35 | 94.75 ± 0.28 | 94.89 ± 0.89 | 80.15 ± 1.48 |
| | Avg. Rank | 7.69 | 7.23 | 5.08 | 5.38 | 6.46 | 5.69 | 5.38 | 3.92 | 4.85 | 3.31 |
| ROC-AUC | Wikipedia | 70.96 ± 0.78 | 67.36 ± 0.96 | 81.93 ± 0.22 | 80.97 ± 0.31 | 70.95 ± 0.95 | 82.19 ± 0.48 | 84.28 ± 0.30 | 75.09 ± 3.70 | 82.74 ± 0.33 | 98.78 ± 0.91 |
| | Reddit | 83.51 ± 0.15 | 82.90 ± 0.31 | 87.13 ± 0.20 | 84.56 ± 0.24 | 88.04 ± 0.29 | 84.67 ± 0.29 | 82.21 ± 0.13 | 86.23 ± 0.51 | 84.38 ± 0.21 | 98.51 ± 0.40 |
| | MOOC | 66.63 ± 2.30 | 63.26 ± 1.01 | 73.18 ± 0.33 | 77.44 ± 2.86 | 70.32 ± 1.43 | 70.36 ± 0.37 | 72.45 ± 0.72 | 80.76 ± 0.76 | 78.47 ± 0.94 | 98.31 ± 0.59 |
| | LastFM | 61.32 ± 3.49 | 62.15 ± 2.12 | 63.99 ± 0.21 | 65.46 ± 4.27 | 67.92 ± 0.44 | 46.93 ± 2.59 | 60.22 ± 0.32 | 69.25 ± 0.36 | 72.30 ± 0.59 | 97.08 ± 1.47 |
| | Enron | 70.92 ± 1.05 | 68.73 ± 1.34 | 60.45 ± 2.12 | 71.34 ± 2.46 | 75.17 ± 0.50 | 67.64 ± 0.86 | 71.53 ± 0.85 | 74.07 ± 0.64 | 77.27 ± 0.61 | 95.79 ± 0.33 |
| | Social Evo. | 90.01 ± 3.19 | 93.07 ± 0.38 | 92.94 ± 0.61 | 95.24 ± 0.56 | 89.93 ± 0.15 | 93.44 ± 0.72 | 94.22 ± 0.32 | 97.51 ± 0.06 | 98.47 ± 0.02 | 93.73 ± 2.22 |
| | UCI | 64.14 ± 1.26 | 54.25 ± 2.01 | 60.80 ± 1.01 | 64.11 ± 1.04 | 58.06 ± 0.26 | 70.05 ± 1.86 | 74.59 ± 0.74 | 65.96 ± 1.18 | 75.39 ± 0.57 | 84.09 ± 1.61 |
| | Flights | 69.99 ± 3.10 | 71.13 ± 1.55 | 73.47 ± 0.18 | 71.63 ± 1.72 | 69.70 ± 0.75 | 72.54 ± 0.19 | 72.21 ± 0.21 | 69.53 ± 1.17 | 64.04 ± 0.55 | 92.32 ± 1.51 |
| | Can. Parl. | 52.88 ± 0.80 | 63.53 ± 0.65 | 72.47 ± 1.18 | 69.57 ± 2.81 | 72.93 ± 1.78 | 69.47 ± 2.12 | 70.52 ± 0.94 | 96.70 ± 0.59 | 65.58 ± 5.71 | 85.80 ± 1.51 |
| | US Legis. | 59.05 ± 5.52 | 89.44 ± 0.71 | 71.62 ± 5.42 | 78.12 ± 4.46 | 76.45 ± 7.02 | 82.54 ± 3.91 | 84.22 ± 0.91 | 87.96 ± 1.80 | 60.71 ± 9.72 | 70.81 ± 3.72 |
| | UN Trade | 66.82 ± 1.27 | 65.60 ± 1.28 | 66.13 ± 0.78 | 66.37 ± 5.39 | 71.73 ± 0.74 | 67.80 ± 1.21 | 66.53 ± 1.22 | 62.56 ± 1.51 | - | 87.92 ± 1.78 |
| | UN Vote | 73.73 ± 1.61 | 72.80 ± 2.16 | 53.04 ± 2.58 | 72.69 ± 3.72 | 52.75 ± 0.90 | 52.02 ± 1.64 | 51.89 ± 0.74 | 53.37 ± 1.26 | 51.95 ± 2.14 | 70.70 ± 3.58 |
| | Contact | 94.47 ± 1.08 | 94.23 ± 0.18 | 94.10 ± 0.41 | 91.64 ± 1.72 | 87.68 ± 0.24 | 91.23 ± 0.19 | 90.96 ± 0.27 | 95.01 ± 0.15 | 95.41 ± 0.49 | 90.68 ± 1.59 |
| | Avg. Rank | 6.92 | 7.0 | 5.85 | 5.23 | 6.23 | 5.92 | 5.69 | 4.23 | 5.15 | 2.77 |

Table 10: Performance comparison in the *inductive setting* with *inductive negative sampling strategy*.

| Metric | Datasets | JODIE | DyRep | TGAT | TGN | CAWN | TCL | GraphMixer | DyGFormer | FreeDyG | SimpleTLP |
|---|---|---|---|---|---|---|---|---|---|---|---|
| ind | Wikipedia | 68.70 ± 0.39 | 62.19 ± 1.28 | 84.17 ± 0.22 | 81.77 ± 0.32 | 67.24 ± 1.63 | 82.20 ± 2.18 | 87.60 ± 0.29 | 71.42 ± 4.43 | 87.54 ± 0.26 | 98.45 ± 0.88 |
| | Reddit | 62.32 ± 0.54 | 61.58 ± 0.72 | 63.40 ± 0.36 | 64.84 ± 0.84 | 63.65 ± 0.41 | 60.81 ± 0.26 | 64.49 ± 0.25 | 65.35 ± 0.60 | 64.98 ± 0.20 | 85.76 ± 2.46 |
| | MOOC | 63.22 ± 1.55 | 62.92 ± 1.24 | 76.72 ± 0.30 | 77.07 ± 3.40 | 74.69 ± 0.68 | 74.28 ± 0.53 | 73.99 ± 0.97 | 80.82 ± 0.30 | 81.41 ± 0.31 | 95.17 ± 2.78 |
| | LastFM | 70.39 ± 4.31 | 71.45 ± 1.75 | 76.28 ± 0.25 | 69.46 ± 4.65 | 71.33 ± 0.47 | 65.78 ± 0.65 | 76.42 ± 0.22 | 76.35 ± 0.52 | 77.01 ± 0.43 | 92.96 ± 1.08 |
| | Enron | 65.86 ± 3.71 | 62.08 ± 2.27 | 61.40 ± 1.30 | 62.90 ± 1.16 | 60.72 ± 0.36 | 67.11 ± 0.62 | 72.37 ± 1.38 | 67.07 ± 0.62 | 72.85 ± 0.81 | 89.36 ± 1.01 |
| | Social Evo. | 88.51 ± 0.87 | 88.72 ± 1.10 | 93.97 ± 0.54 | 90.65 ± 1.62 | 79.83 ± 0.39 | 94.10 ± 0.32 | 94.01 ± 0.47 | 96.82 ± 0.17 | 96.91 ± 0.12 | 85.13 ± 1.05 |
| | UCI | 63.16 ± 2.27 | 52.47 ± 2.09 | 70.49 ± 0.93 | 70.73 ± 0.79 | 64.54 ± 0.47 | 76.65 ± 0.99 | 81.64 ± 0.49 | 72.13 ± 1.86 | 82.06 ± 0.58 | 75.97 ± 1.13 |
| | Flights | 61.01 ± 1.66 | 62.83 ± 1.31 | 64.72 ± 0.37 | 59.32 ± 1.45 | 56.82 ± 0.56 | 64.50 ± 0.25 | 65.29 ± 0.24 | 57.11 ± 0.20 | 56.70 ± 1.17 | 69.14 ± 1.19 |
| | Can. Parl. | 52.58 ± 0.86 | 52.24 ± 0.28 | 56.46 ± 0.50 | 54.18 ± 0.73 | 57.06 ± 0.08 | 55.46 ± 0.69 | 55.76 ± 0.65 | 87.22 ± 0.82 | 52.33 ± 0.97 | 69.59 ± 1.45 |
| | US Legis. | 52.94 ± 2.11 | 62.10 ± 1.41 | 51.83 ± 3.95 | 61.18 ± 1.10 | 55.56 ± 1.71 | 53.87 ± 1.41 | 52.03 ± 1.02 | 56.31 ± 3.46 | 53.63 ± 4.38 | 49.51 ± 1.09 |
| | UN Trade | 55.43 ± 1.20 | 55.42 ± 0.87 | 55.58 ± 0.68 | 52.80 ± 3.24 | 54.97 ± 0.38 | 55.66 ± 0.98 | 54.88 ± 1.01 | 52.56 ± 1.70 | - | 79.12 ± 1.84 |
| | UN Vote | 61.17 ± 1.33 | 60.29 ± 1.79 | 53.08 ± 3.10 | 63.71 ± 2.97 | 48.01 ± 0.82 | 54.13 ± 2.16 | 48.10 ± 0.40 | 52.61 ± 1.25 | 50.21 ± 1.74 | 61.88 ± 1.27 |
| | Contact | 90.43 ± 2.33 | 89.22 ± 0.65 | 94.14 ± 0.45 | 88.12 ± 1.50 | 74.19 ± 0.81 | 90.43 ± 0.17 | 89.91 ± 0.36 | 93.55 ± 0.52 | 93.26 ± 1.41 | 79.09 ± 1.25 |
| | Avg. Rank | 6.85 | 7.08 | 5.23 | 5.77 | 7.54 | 5.23 | 4.92 | 4.46 | 4.62 | 3.31 |
| ind | Wikipedia | 61.87 ± 0.53 | 57.54 ± 1.09 | 78.38 ± 0.20 | 75.76 ± 0.29 | 62.02 ± 0.65 | 79.79 ± 0.96 | 82.88 ± 0.21 | 68.33 ± 2.82 | 83.17 ± 0.31 | 98.59 ± 0.82 |
| | Reddit | 61.69 ± 0.39 | 60.44 ± 0.37 | 64.39 ± 0.27 | 64.55 ± 0.50 | 64.91 ± 0.21 | 61.36 ± 0.26 | 64.27 ± 0.13 | 64.80 ± 0.25 | 64.51 ± 0.19 | 92.05 ± 1.87 |
| | MOOC | 64.48 ± 1.64 | 64.22 ± 1.29 | 74.07 ± 0.27 | 77.68 ± 3.55 | 71.69 ± 0.94 | 69.83 ± 0.32 | 72.52 ± 0.84 | 80.77 ± 0.63 | 75.81 ± 0.69 | 96.85 ± 1.59 |
| | LastFM | 68.44 ± 3.26 | 68.79 ± 1.08 | 69.89 ± 0.28 | 66.99 ± 5.61 | 67.68 ± 0.24 | 55.88 ± 1.85 | 70.07 ± 0.20 | 70.73 ± 0.37 | 71.42 ± 0.33 | 96.34 ± 1.68 |
| | Enron | 65.32 ± 3.57 | 61.50 ± 2.50 | 57.83 ± 2.18 | 62.68 ± 1.09 | 62.27 ± 0.40 | 64.05 ± 1.02 | 68.19 ± 1.63 | 65.79 ± 0.42 | 68.79 ± 0.91 | 92.14 ± 0.70 |
| | Social Evo. | 88.53 ± 0.55 | 87.93 ± 1.05 | 91.88 ± 0.72 | 92.10 ± 1.22 | 83.54 ± 0.24 | 93.28 ± 0.60 | 93.21 ± 0.35 | 96.91 ± 0.09 | 96.79 ± 0.17 | 90.08 ± 3.91 |
| | UCI | 60.27 ± 1.94 | 51.26 ± 2.40 | 62.29 ± 1.17 | 62.66 ± 0.91 | 56.39 ± 0.11 | 70.42 ± 1.93 | 75.97 ± 0.85 | 65.58 ± 1.00 | 73.41 ± 0.88 | 84.63 ± 1.22 |
| | Flights | 60.72 ± 1.29 | 61.99 ± 1.39 | 63.40 ± 0.26 | 59.66 ± 1.05 | 56.58 ± 0.44 | 63.49 ± 0.23 | 63.32 ± 0.19 | 56.05 ± 0.22 | 55.18 ± 1.04 | 83.87 ± 1.30 |
| | Can. Parl. | 51.61 ± 0.98 | 52.35 ± 0.52 | 58.15 ± 0.62 | 55.43 ± 0.42 | 60.01 ± 0.47 | 56.88 ± 0.93 | 56.63 ± 1.09 | 88.51 ± 0.73 | 51.37 ± 2.23 | 81.26 ± 3.92 |
| | US Legis. | 58.12 ± 2.94 | 67.94 ± 0.98 | 49.99 ± 4.88 | 64.87 ± 1.65 | 54.41 ± 1.31 | 52.12 ± 2.13 | 49.28 ± 0.86 | 56.57 ± 3.22 | 53.75 ± 5.92 | 55.74 ± 1.53 |
| | UN Trade | 58.71 ± 1.20 | 57.87 ± 1.36 | 59.98 ± 0.59 | 55.62 ± 3.59 | 60.88 ± 0.79 | 61.01 ± 0.93 | 59.71 ± 1.17 | 57.28 ± 3.06 | - | 86.28 ± 1.76 |
| | UN Vote | 65.29 ± 1.30 | 64.10 ± 2.10 | 51.78 ± 4.14 | 68.58 ± 3.08 | 48.04 ± 1.76 | 54.65 ± 2.20 | 45.57 ± 0.41 | 53.87 ± 2.01 | 48.93 ± 2.53 | 72.98 ± 1.22 |
| | Contact | 90.80 ± 1.18 | 88.87 ± 0.67 | 93.76 ± 0.40 | 88.85 ± 1.39 | 74.79 ± 0.38 | 90.37 ± 0.16 | 90.04 ± 0.29 | 94.14 ± 0.26 | 94.27 ± 0.76 | 88.69 ± 3.25 |
| | Avg. Rank | 6.54 | 7.38 | 5.69 | 5.77 | 7.0 | 5.54 | 5.31 | 4.23 | 5.08 | 2.46 |

## E.2 THEORETICAL TIME COMPLEXITY COMPARISON WITH SOTAs

Now, we give an in-depth analysis of the complexity of DyGFormer and FreeDyG as follows.

Suppose we are given $E$ query links: $\{(u_i, v_i, t_i)\}_{i=1}^E$. As we move to new timestamps, new connections and new nodes will emerge, resulting in the changes in some nodes' neighborhoods; then, both SimpleTLP and DyGFormer need to update historical neighbors for some nodes. (Note that for a node, both SimpleTLP and DyGFormer consider its historical neighbors, so both methods would need to perform an update if its neighborhood changes).

For each link $(u_i, v_i, t_i)$, DyGFormer and FreeDyG first need to obtain the neighbor co-occurrence information of the node pair $(u_i, v_i)$ as follows. Suppose $u_i$ has $p$ historical neighbors, $a_1, \ldots, a_p$, and $v_i$ has $q$ historical neighbors, $b_1, \ldots, b_q$, then the neighbor co-occurrence scheme of DyGFormer and FreeDyG requires counting (1) the number of times $a_1, \ldots, a_p$ are involved in interactions with $v_i$ before time $t_i$ and (2) the number of times $b_1, \ldots, b_q$ are involved in interactions with $u_i$ before time $t_i$. In other words, for an arbitrary link $(u_i, v_i, t_i)$, the neighbor co-occurrence framework of DyGFormer and FreeDyG makes the computation of $u_i$ depend on $v_i$'s information, as they have to count how many times historical neighbors of $u_i$ are also $v_i$'s historical neighbors, and vice versa. Thus, in order to make predictions for the aforementioned query links, DyGFormer and FreeDyG require at least $\mathcal{O}(E)$ complexity.

On the other hand, for SimpleTLP, we first consider the unique nodes among $u_1, \ldots, u_E, v_1, \ldots, v_E$. Suppose there are $n$ unique nodes, and let us denote them as $c_1, \ldots, c_n$, then we could compute their temporal representation as stated in Eq. 9. Moreover, in the process of obtaining the necessary components for Eq. 9, for any node $c_i$, we only use $c_i$'s related information (most recent temporal neighbors / links and its previous positional encoding). Thus, it takes SimpleTLP $\mathcal{O}(n)$ time to compute the representation for these $n$ nodes. To predict an arbitrary link $(u_i, v_i, t_i)$, we simply retrieve the representation for $u_i, v_i$ that we just computed and apply Eq. 10

In summary, DyGFormer and FreeDyG have $\mathcal{O}(E)$ since they process a node pair $(u, v)$-related information, while SimpleTLP's complexity linearly scales with the number of nodes.

## E.3 EMPIRICAL RUNTIME COMPARISON WITH SOTAs

We present the runtime comparison between SimpleTLP and DyGFormer (Yu et al., 2023), FreeDyG (Tian et al., 2024), recent SOTA methods, on *US Legis* and *UN Trade* datasets, in Table 11. As shown in Table 11, our running time is lower than SOTAs in terms of convergence time.

Table 11: Comparison in runtime (in seconds) and number of training epochs of SimpleTLP and DyGFormer across 5 runs on US Legis and UN Trade.

| Dataset | Method | Total time for 5 runs (in seconds) | Number of training epochs |
|---|---|---|---|
| US Legis | SimpleTLP | **[287.19, 257.18, 251.49, 248.92 , 305.09]** | **[21, 18, 17, 17, 21]** |
| | DyGFormer | [1251.97, 1697.56, 1112.50, 756.80, 1092.77] | [27, 37, 24, 16, 23] |
| | FreeDyG | [1019.91, 871.21, 1061.15, 1070.41, 868.03] | [17, 13, 18, 18, 13] |
| UN Trade | SimpleTLP | **[2922.14 ,2857.13 , 2450.84 , 2593.40, 2740.26]** | **[21, 21, 18, 18, 19]** |
| | DyGFormer | [12847.83, 21167.01, 20380.42, 9901.51, 39374.45] | [33, 55, 53, 25, 95] |
| | FreeDyG | - | - |

## E.4 EMPIRICAL RESULTS FOR ABLATION STUDY

Next, we conduct an ablation study to examine the ability of the learnable positional encoding in the link prediction task.

Specifically, we consider the following cases: (1) only leveraging LPE to make predictions (i.e., we omit node/edge features and only employ positional encoding learned by LPE to obtain link predictions); (2) omitting LPE module (i.e., only relying on node and link features).

First, we provide the performance comparison between SimpleTLP and only using LPE module (denoted as "Only LPE") to obtain link predictions under transductive and inductive settings with

random negative sampling strategy. As shown in Table 12, on most datasets, the performance obtained by only leveraging LPE is already competitive, which suggests the positional encoding is informative enough to support downstream tasks. Also, we can observe that LPE positional encodings can collaborate well with node and edge features to boost performance.

Table 12: Comparison between only LPE and SimpleTLP with random negative sampling strategy.

| Dataset | Transductive setting | | | | Inductive setting | | | |
|---|---|---|---|---|---|---|---|---|
| | AP | | ROC-AUC | | AP | | ROC-AUC | |
| | SimpleTLP | Only LPE | SimpleTLP | Only LPE | SimpleTLP | Only LPE | SimpleTLP | Only LPE |
| Wikipedia | **99.34 ± 0.04** | 99.07 ± 0.03 | **99.48 ± 0.03** | 99.42 ± 0.03 | **99.15 ± 0.04** | 98.77 ± 0.05 | **99.30 ± 0.03** | 98.99± 0.03 |
| Reddit | **99.37 ± 0.04** | 98.67 ± 0.07 | **99.49 ± 0.03** | 99.13 ± 0.05 | **98.02 ± 0.09** | 95.44 ± 0.29 | **98.49 ± 0.06** | 96.71±0.16 |
| MOOC | **86.94 ± 0.34** | 81.55 ± 0.53 | **89.28 ± 0.28** | 83.04 ± 0.44 | **88.49 ± 0.24** | 80.04 ± 0.40 | **90.63 ± 0.21** | 81.70± 0.39 |
| Lastfm | **96.06 ± 0.30** | 92.84 ± 0.54 | **97.52 ± 0.17** | 94.43 ± 0.38 | **96.25 ± 0.43** | 94.27 ± 0.36 | **97.66 ± 0.21** | 96.22± 0.22 |
| Enron | **93.96 ± 0.36** | 93.82 ± 0.44 | **95.98 ± 0.23** | 95.95 ± 0.31 | **89.17 ± 0.88** | 89.15 ±1.32 | 92.30 ± 0.57 | **92.33 ±0.79** |
| Social Evo. | **92.22 ± 0.25** | 91.56 ± 0.79 | **94.33 ± 0.21** | 93.76 ± 0.63 | **91.71 ± 0.32** | 90.42± 1.68 | **94.09 ± 0.24** | 93.28± 0.97 |
| UCI | 96.67 ± 0.47 | **96.83 ± 0.19** | 97.62 ± 0.23 | **97.89 ± 0.10** | **94.60 ± 0.41** | 94.60 ± 0.46 | 95.88 ± 0.17 | **96.17 ±0.22** |
| Flights | 98.94 ± 0.10 | **99.10 ± 0.03** | 99.38 ± 0.04 | **99.43 ± 0.06** | 97.43 ± 0.15 | **97.81 ±0.25** | 98.46 ± 0.07 | **98.62 ±0.11** |
| Can. Parl. | 98.24 ± 0.11 | **98.41 ± 0.06** | 98.97 ± 0.06 | **99.03 ± 0.03** | **92.25 ± 0.24** | 92.13 ± 0.24 | 95.06 ± 0.11 | **95.09 ±0.07** |
| US Legis. | 76.74 ± 0.60 | **77.36 ± 0.26** | 83.89 ± 0.43 | **84.36 ± 0.20** | 61.98 ± 1.31 | **63.47 ±0.63** | 66.18 ± 1.19 | **66.75± 0.57** |
| UN Trade | 75.84 ± 2.08 | **78.46 ± 0.97** | **83.17 ± 1.28** | 77.81 ± 1.78 | 76.80 ± 2.01 | **84.85± 0.89** | 83.95 ± 1.76 | **84.89± 1.39** |
| UN Vote | **73.40 ± 0.03** | 70.99 ± 0.11 | **79.59 ± 0.02** | 76.67 ± 0.09 | **75.09 ± 0.34** | 70.01± 0.69 | **82.01 ± 0.21** | 77.36 ±0.51 |
| Contact | **98.16 ± 0.09** | 97.73 ± 0.04 | **98.72 ± 0.06** | 98.40 ± 0.03 | **97.25 ± 0.07** | 96.38± 0.09 | **97.99 ± 0.07** | 97.37± 0.07 |

Second, we present the comparison between SimpleTLP and only using node/edge features, i.e., omitting the LPE module from SimpleTLP (denoted as "W/o LPE"). As shown in Table 13, SimpleTLP performs the best on all datasets, further suggesting that performance would be negatively affected without employing LPE.

Table 13: Comparison between not using LPE and SimpleTLP with random negative sampling strategy.

| Dataset | Transductive setting | | | | Inductive setting | | | |
|---|---|---|---|---|---|---|---|---|
| | AP | | ROC-AUC | | AP | | ROC-AUC | |
| | SimpleTLP | W/o LPE | SimpleTLP | W/o LPE | SimpleTLP | W/o LPE | SimpleTLP | W/o LPE |
| Wikipedia | **99.34 ± 0.04** | 96.52 ± 0.06 | **99.48 ± 0.03** | 96.15 ± 0.08 | **99.15 ± 0.04** | 96.11 ± 0.09 | **99.30 ± 0.03** | 95.64 ± 0.11 |
| Reddit | **99.37 ± 0.04** | 97.07 ± 0.08 | **99.49 ± 0.03** | 96.96 ± 0.07 | **98.02 ± 0.09** | 95.10 ± 0.06 | **98.49 ± 0.06** | 95.02 ± 0.06 |
| MOOC | **86.94 ± 0.34** | 81.89 ± 0.32 | **89.28 ± 0.28** | 83.24 ± 0.29 | **88.49 ± 0.24** | 80.65 ± 0.28 | **90.63 ± 0.21** | 82.19 ± 0.26 |
| LastFM | **96.06 ± 0.30** | 74.63 ± 0.41 | **97.52 ± 0.17** | 73.48 ± 0.40 | **96.25 ± 0.43** | 82.08 ± 0.46 | **97.66 ± 0.21** | 80.74 ± 0.37 |
| Enron | **93.96 ± 0.36** | 82.49 ± 0.85 | **95.98 ± 0.23** | 84.18 ± 0.53 | **89.17 ± 0.88** | 76.14 ± 0.82 | **92.30 ± 0.57** | 76.63 ± 0.65 |
| Social Evo. | **92.22 ± 0.25** | 91.17 ± 0.21 | **94.33 ± 0.21** | 93.30 ± 0.17 | **91.71 ± 0.32** | 89.38 ± 0.30 | **94.09 ± 0.24** | 91.85 ± 0.24 |
| UCI | **96.67 ± 0.47** | 91.79 ± 4.08 | **97.62 ± 0.23** | 90.51 ± 3.45 | **94.60 ± 0.41** | 89.58 ± 3.41 | **95.88 ± 0.17** | 87.88 ± 3.10 |
| Flights | **98.94 ± 0.10** | 90.91 ± 0.02 | **99.38 ± 0.04** | 91.03 ± 0.01 | **97.43 ± 0.15** | 82.72 ± 0.05 | **98.46 ± 0.07** | 82.05 ± 0.04 |
| Can. Parl. | **98.24 ± 0.11** | 67.19 ± 1.39 | **98.97 ± 0.06** | 76.51 ± 1.37 | **92.25 ± 0.24** | 52.47 ± 0.88 | **95.06 ± 0.11** | 50.99 ± 1.72 |
| US Legis. | **76.74 ± 0.60** | 60.98 ± 5.43 | **83.89 ± 0.43** | 65.76 ± 7.15 | **61.98 ± 1.31** | 52.59 ± 2.13 | **66.18 ± 1.19** | 53.75 ± 1.07 |
| UN Trade | **75.84 ± 2.08** | 62.04 ± 0.72 | **83.17 ± 1.28** | 65.91 ± 0.62 | **76.80 ± 2.01** | 61.84 ± 0.53 | **83.95 ± 1.76** | 63.54 ± 0.54 |
| UN Vote | **73.40 ± 0.03** | 52.29 ± 0.44 | **79.59 ± 0.02** | 52.55 ± 0.61 | **75.09 ± 0.34** | 50.74 ± 0.82 | **82.01 ± 0.21** | 49.39 ± 0.75 |
| Contact | **98.16 ± 0.09** | 91.39 ± 0.05 | **98.72 ± 0.06** | 93.57 ± 0.04 | **97.25 ± 0.07** | 89.71 ± 0.05 | **97.99 ± 0.07** | 92.24 ± 0.05 |

Combining the results from Tables 12 and 13, we discern that (1) in most cases, the full design of SimpleTLP is the best and removing any component can lead to suboptimal results, (2) in a few cases, only using LPE can also perform competitively, which suggests that the Learnable Positional Encoding module in our SimpleTLP is relatively dominating than input node and edge features.

## E.5 EMPIRICAL RESULTS FOR PARAMETER ANALYSIS

### E.5.1 PARAMETER ANALYSIS FOR $t_{gap}$

Here, we examine how varying the values of $t_{gap}$ affects the performance of SimpleTLP under both *transductive* and *inductive* settings with random negative sampling.

We conduct the parameter analysis on two datasets: *Enron* and *UN Vote*, and adjusting $t_{gap}$ to different values: $\{10, 50, 1000, 2000\}$. Due to the page limit, we report the results in Table 14 for *transductive* and Table 15 for *inductive* setting, respectively. The best results are emphasized with **bold**. Firstly, it worth noting that the ratio #Links / (Unique Steps) of *UN Vote* is much larger than that of *Enron*, since *UN Vote* has significantly more links and fewer number of unique steps than

Table 14: AP and ROC-AUC with various values of $t_{gap}$ under *transductive setting* and random negative sampling.

| Dataset | Metric | $t_{gap} = 10$ | $t_{gap} = 50$ | $t_{gap} = 1000$ | $t_{gap} = 2000$ |
|---------|--------|----------------|----------------|------------------|------------------|
| Enron | AP | $93.77 \pm 0.28$ | $93.82 \pm 0.11$ | $\mathbf{93.96 \pm 0.36}$ | $93.70 \pm 0.43$ |
|  | ROC-AUC | $95.83 \pm 0.14$ | $95.86 \pm 0.07$ | $\mathbf{95.98 \pm 0.23}$ | $95.83 \pm 0.21$ |
| UN Vote | AP | $\mathbf{73.40 \pm 0.03}$ | $73.38 \pm 0.07$ | $73.39 \pm 0.08$ | $73.28 \pm 0.33$ |
|  | ROC-AUC | $\mathbf{79.59 \pm 0.02}$ | $79.54 \pm 0.08$ | $79.57 \pm 0.06$ | $79.53 \pm 0.22$ |

Table 15: AP and ROC-AUC with various values of $t_{gap}$ under *inductive setting* and random negative sampling.

| Dataset | Metric | $t_{gap} = 10$ | $t_{gap} = 50$ | $t_{gap} = 1000$ | $t_{gap} = 2000$ |
|---------|--------|----------------|----------------|------------------|------------------|
| Enron | AP | $88.43 \pm 0.52$ | $88.76 \pm 0.62$ | $\mathbf{89.17 \pm 0.88}$ | $88.41 \pm 0.21$ |
|  | ROC-AUC | $91.83 \pm 0.37$ | $92.02 \pm 0.37$ | $\mathbf{92.30 \pm 0.57}$ | $91.86 \pm 0.24$ |
| Un Vote | AP | $\mathbf{75.09 \pm 0.34}$ | $75.01 \pm 0.29$ | $74.92 \pm 0.48$ | $74.73 \pm 0.44$ |
|  | ROC-AUC | $\mathbf{82.01 \pm 0.21}$ | $79.54 \pm 0.08$ | $81.85 \pm 0.32$ | $81.77 \pm 0.16$ |

*Enron*, as stated in Table 6, implying *UN Vote* is much denser that *Enron*. From Tables 14 and 15, we can see that the best performance on *UN Vote* is derived from $t_{gap} = 10$, which is the smallest value, while the best result on *Enron* is obtained by a much larger value of $t_{gap}$, which is 1000. These findings indicate that a small $t_{gap}$ value functions effectively for dense graphs, whereas a larger $t_{gap}$ value is suited for sparse graphs. The detailed configurations are reported in Table 22 further suggest this finding, as the $t_{gap}$ values associated with the sparse dataset (*Wikipedia, Reddit, MOOC, LastFM, Enron, Social Evo., UCI*) mostly are in the range of $[1000, 2000]$, while the $t_{gap}$ values employed for *Can. Parl., US Legis., UN Trade, UN Vote,* and *Contact* are only less than or equal to 10.

We report the results for the parameter analysis regarding how the performance change with different values of $t_{gap}$ Tables 14 and 15.

### E.5.2 PARAMETER ANALYSIS FOR $K$

Next, we conduct parameter analysis for the neighborhood size, $K$, with random negative sampling, and report the results under *transductive* and *inductive* settings in Table 16 and Table 17, respectively.

Table 16: AP and ROC-AUC with various values of $K$ under *transductive setting* and random negative sampling.

| Dataset | Metric | $K = 5$ | $K = 20$ | $K = 50$ | $K = 100$ |
|---------|--------|---------|----------|----------|-----------|
| Enron | AP | $93.32 \pm 0.25$ | $\mathbf{93.96 \pm 0.36}$ | $93.53 \pm 0.51$ | $93.22 \pm 0.38$ |
|  | ROC-AUC | $95.50 \pm 0.16$ | $\mathbf{95.98 \pm 0.23}$ | $95.71 \pm 0.31$ | $95.48 \pm 0.19$ |
| UN Vote | AP | $72.08 \pm 0.73$ | $73.40 \pm 0.03$ | $\mathbf{73.49 \pm 0.10}$ | $73.31 \pm 0.13$ |
|  | ROC-AUC | $78.92 \pm 0.33$ | $79.59 \pm 0.02$ | $\mathbf{79.69 \pm 0.10}$ | $79.58 \pm 0.08$ |

According to the tables, we can discern that: (1) in a few cases, increasing $K$ can slightly increase the performance, but it is prone to cost more computational complexity; (2) in most cases, increasing $K$ does not add much to the performance, which suggests that close neighbors have already had enough information to support downstream tasks.

### E.5.3 PARAMETER ANALYSIS FOR $L$

Here, we report the parameter analysis for $L$ as follows on Can. Parl. dataset in Table .

From Table 18, we can observe that our proposed SimpleTLP just relies on looking back a few historical information to make the correct information, as $L$ increases, the performance can be

Table 17: AP and ROC-AUC with various values of $K$ under *inductive setting* and random negative sampling.

| Dataset | Metric | $K = 5$ | $K = 20$ | $K = 50$ | $K = 100$ |
|---------|--------|---------|----------|----------|-----------|
| Enron | AP | $89.72 \pm 0.50$ | $\mathbf{94.60 \pm 0.41}$ | $88.99 \pm 1.04$ | $88.29 \pm 0.58$ |
| | ROC-AUC | $92.65 \pm 0.33$ | $\mathbf{95.88 \pm 0.17}$ | $92.11 \pm 0.65$ | $91.64 \pm 0.36$ |
| UN Vote | AP | $\mathbf{77.76 \pm 0.49}$ | $75.09 \pm 0.34$ | $75.10 \pm 0.34$ | $73.92 \pm 0.75$ |
| | ROC-AUC | $83.73 \pm 0.23$ | $\mathbf{83.95 \pm 1.76}$ | $81.96 \pm 0.25$ | $80.94 \pm 0.82$ |

Table 18: AP and ROC-AUC with various values of $L$ under *transductive setting* and *inductive setting* with random negative sampling.

| Dataset | | Transductive AP | Transductive ROC-AUC | Inductive AP | Inductive ROC-AUC |
|---------|---|-----------------|----------------------|--------------|-------------------|
| Can. Parl | $L = 20$ | $98.24 \pm 0.11$ | $98.97 \pm 0.06$ | $92.25 \pm 0.24$ | $95.06 \pm 0.11$ |
| | $L = 50$ | $98.26 \pm 0.06$ | $98.94 \pm 0.03$ | $92.15 \pm 0.34$ | $95.11 \pm 0.17$ |
| | $L = 100$ | $98.11 \pm 0.21$ | $98.85 \pm 0.11$ | $92.16 \pm 0.98$ | $95.25 \pm 0.56$ |
| | $L = 200$ | $97.04 \pm 0.22$ | $98.37 \pm 0.09$ | $91.96 \pm 0.59$ | $95.19 \pm 0.38$ |

downgraded, and the best performance exists between 20 and 50, which discovery demonstrates that the near past history plays a more important role in the current decision making than the long past history.

### E.5.4 PARAMETER ANALYSIS FOR $\alpha_{neg}$

We present the parameter analysis for $\alpha_{neg}$ on Can. Parl. dataset in Table 19.

Table 19: AP and ROC-AUC with various values of $\alpha_{neg}$ under *transductive setting* and *inductive setting* with random negative sampling.

| Dataset | | Transductive AP | Transductive ROC-AUC | Inductive AP | Inductive ROC-AUC |
|---------|---|-----------------|----------------------|--------------|-------------------|
| Can. Parl | $\alpha_{neg} = 0.05$ | $98.26 \pm 0.13$ | $98.98 \pm 0.05$ | $92.24 \pm 0.09$ | $95.09 \pm 0.11$ |
| | $\alpha_{neg} = 0.1$ | $98.28 \pm 0.09$ | $98.98 \pm 0.05$ | $92.35 \pm 0.21$ | $95.09 \pm 0.09$ |
| | $\alpha_{neg} = 0.3$ | $98.24 \pm 0.11$ | $98.97 \pm 0.06$ | $92.25 \pm 0.24$ | $95.06 \pm 0.11$ |
| | $\alpha_{neg} = 0.5$ | $98.28 \pm 0.13$ | $98.97 \pm 0.05$ | $91.48 \pm 0.23$ | $94.75 \pm 0.16$ |
| | $\alpha_{neg} = 0.95$ | $98.24 \pm 0.18$ | $98.95 \pm 0.09$ | $91.09 \pm 0.49$ | $94.53 \pm 0.32$ |

In general, Table 19 shows (1) our proposed method SimpleTLP is robust towards different choices of $\alpha_{neg}$, and (2) usually a smaller weight of negative pairs can help the model concentrate more on the positive pairs and boost the performance.

### E.5.5 PARAMETER ANALYSIS FOR $\alpha_{pe}$

We present the parameter analysis for $\alpha_{neg}$ on Enron dataset in Table 20.

Table 20: AP and ROC-AUC with various values of $\alpha_{pe}$ under *transductive setting* and *inductive setting* with random negative sampling.

| Dataset | | Transductive AP | Transductive ROC-AUC | Inductive AP | Inductive ROC-AUC |
|---------|---|-----------------|----------------------|--------------|-------------------|
| Enron | $\alpha_{pe} = 0.05$ | $92.81 \pm 0.82$ | $95.32 \pm 0.38$ | $86.49 \pm 1.52$ | $90.91 \pm 0.61$ |
| | $\alpha_{pe} = 0.3$ | $93.96 \pm 0.36$ | $95.98 \pm 0.23$ | $89.17 \pm 0.88$ | $92.30 \pm 0.57$ |
| | $\alpha_{pe} = 0.5$ | $93.94 \pm 0.10$ | $95.90 \pm 0.06$ | $88.92 \pm 0.39$ | $92.12 \pm 0.22$ |
| | $\alpha_{pe} = 0.95$ | $94.00 \pm 0.28$ | $95.92 \pm 0.17$ | $88.67 \pm 0.34$ | $91.91 \pm 0.30$ |

$\alpha_{neg}$ stands for the weight of the loss function of learnable positional encoding. As shown in the Table 20, we can observe that (1) our SimpleTLP is robust towards different choices of $\alpha_{pe}$, and (2) a considerable weight, e.g., 0.5, is optimal, too large or too small weight can induce suboptimal results.

E.6 DIFFERENT POSITIONAL ENCODING INITIALIZATION

Table 21: Comparison in Positional Encoding initialization with Laplacian and Random Walk Positional Encoding under *transductive* and *inductive* settings with random negative sampling.

| Dataset | Positional Encoding | Transductive AP | Transductive ROC-AUC | Inductive AP | Inductive ROC-AUC |
|---------|---------------------|-----------------|----------------------|--------------|-------------------|
| Enron | Random Walk PE | $93.79 \pm 0.67$ | $95.88 \pm 0.40$ | $89.11 \pm 0.83$ | $92.43 \pm 0.64$ |
|       | Laplacian PE | $93.96 \pm 0.36$ | $95.98 \pm 0.23$ | $89.17 \pm 0.88$ | $92.30 \pm 0.57$ |

In this section, we demonstrate that SimpleTLP can be equipped with different positional encoding method, other than Laplacian PE. Specifically, our work aims to introduce a general framework on how to approximate positional encodings at a current time solely based on information from previous timestamps to outperform in temporal link prediction task, and we do not want to constraint the type of positional encodings. In Table 21, we provide the comparison between initializing the positional encodings at initial timestamp with Laplacian PE and Random Walk PE (Dwivedi et al., 2022) under two settings, *transductive* and *inductive*, with random negative sampling.

Based on Table 21, we can discern that (1) SimpleTLP can handle different positional encodings and perform robustly; (2) different positional encodings bring slightly different performances and finding a powerful positional encoding is a promising future direction to trigger more interesting works.

# F REPRODUCIBILITY

## F.1 DIFFERENT NEGATIVE SAMPLING STRATEGIES (NSS)

In brief, random NSS samples possible node pairs uniformly at random, historical NSS samples negative edges from the edges occurring in past timestamps but are absent in the present time, and inductive NSS samples edges that are unseen during the training time. We refer readers to (Poursafaei et al., 2022) for more details regarding these three sampling strategies.

**Configuration and implementation details.** Following the training procedure of (Yu et al., 2023), SimpleTLP is trained with a maximum of 200 epochs, and we employ early stopping with patience 10 during the training process. We leverage the Adam optimizer with learning rate of 0.0001. For a more detailed description of the model's implementation, computational resources, and configurations of hyper-parameters over all 13 datasets, we refer readers to Appendix F.2, F.3.

Across Table 1, Table 2, Table 7, Table 8, Table 9, and Table 10, the model with the best performance, in terms of metric scores AP and AUC-ROC, on the validation set will be selected for testing. We run SimpleTLP 5 times with different random seeds from 0 to 4 and report the average metric score. Results from all other baselines are also obtained in the same manner (Yu et al., 2023).

## F.2 MODEL CONFIGURATIONS, HYPER-PARAMETERS, AND COMPUTING RESOURCES

We first report the configuration and hyper-parameters that are unchanged for all 13 datasets:

- Dimension of time encoding: $d_T = 100$.
- Dimension of node encoding: $d_N = 172$.
- Dimension of edge encoding: $d_E = 172$.
- Dimension of positional encoding: $d_P = 172$ (only for *Social Evo.*, $d_P = 72$).
- Hyper-parameters for time encoding function: $\alpha = 10, \beta = 10$.
- Weight of negative samples in positional encoding loss: $\alpha_{neg} = 0.3$.
- Weight of positional encoding loss in objective loss function of SimpleTLP $\alpha_{pe} = 0.5$.

The experiments are coded by Python and are performed on a Linux machine with a single NVIDIA Tesla V100 32GB GPU. The code will be released upon paper's publication.

Next, for reproducibility, we present detailed hyper-parameters across 13 datasets in Table 22.

Table 22: Configurations of the number of recent snapshots for computing LPE, time window for node features, recent interactions for edge features, and batch size over 13 datasets

| Dataset | $L$ | $t_{gap}$ | $K$ | Batch size |
|---------|-----|-----------|-----|------------|
| Wikipedia | 100 | 1000 | 15 | 128 |
| Reddit | 100 | 1000 | 20 | 200 |
| MOOC | 100 | 2000 | 30 | 128 |
| Lastfm | 100 | 1000 | 30 | 128 |
| Enron | 100 | 1000 | 20 | 64 |
| Social Evo. | 100 | 1000 | 20 | 128 |
| UCI | 200 | 500 | 30 | 100 |
| Flights | 100 | 1000 | 30 | 128 |
| Can. Parl. | 20 | 2 | 10 | 64 |
| US Legis. | 50 | 2 | 10 | 200 |
| UN Trade | 200 | 6 | 30 | 200 |
| UN Vote | 100 | 10 | 20 | 128 |
| Contact | 200 | 10 | 20 | 128 |

### F.3 RUNNING SIMPLETLP ON CONTINUOUS TIME DYNAMIC GRAPHS

The definition of discrete time and continuous time dynamic graphs can be referred to the survey paper (Kazemi et al., 2020), where

- Continuous Time Dynamic Graph (CTDG) is represented as $((v_2, v_5), t_1)$, $((v_1, v_2), t_2)$, ..., as shown in its Example 2;

- Discrete Time Dynamic Graph (DTDG) is represented as a set of $G_1, G_2, \ldots, G_T$, where $G_t = (V_t, E_t)$ is the graph at snapshot $t$, $V_t$ is the set of nodes in $G_t$, and $E_t$ is the set of edges in $G_t$.

In this viewpoint, our method is designed for discrete time. However, we can see a clear transformation between CTDG and DTDG: if we aggregate edges that happen at the same timestamp $t$ into a set, then it is the graph snapshot $G_t$. This is not invented by us, a similar example can be seen in Example 3 of the survey paper (Kazemi et al., 2020).

Because the introduction and theoretical derivation on the snapshot level are much easier to understand by people, and using "a set of continuous-time edges" every time when we do theoretical derivation and illustration can be wordy and hurt the presentation. Since graph snapshot and relevant concepts are already existing in the community and cover the meaning, then we choose to follow them.

Then, in this section, we elaborate how SimpleTLP can be evaluated on Continuous Time Dynamic Graphs. Firstly, we explain the data batching procedure for Continuous Time Dynamic Graphs as follows. Suppose we are given a CTDG with $T$ interactions, $G = \{u_i, v_i, t_i\}_{i=1}^T$, where $t_1 \leq \ldots t_T$ and $(u_i, v_i, t_i)$ denotes the link occurrence between $u_i, v_i$ at time $t_i$. Let the batch size be $B$, then a data batch would be $B$ consecutive interactions from $\{u_i, v_i, t_i\}_{i=1}^T$. Specifically, suppose $T = B \cdot K$ (for some $K$), then our 1st data batch would be the first $B$ link occurrences of $G$: $\{u_i, v_i, t_i\}_{i=1}^B$. The 2nd data batch would be the next $B$ consecutive interactions, $\{u_i, v_i, t_i\}_{i=B+1}^{2B}$, and so on. In this way, now we obtain $K$ data batches. Moreover, as a batch is a stream of $B$ events, so we can regard a data batch as a mini temporal graph with $B$ link occurrences. This data batching technique is also employed by other baselines.

Next, we explain the evaluation process of SimpleTLP. Consider an arbitrary data batch $\{u_j, v_j, \tau_j\}_{j=1}^B$. If this is the first data batch, then we first extract all links occurrence (without the timestamps), $\{u_j, v_j\}_{j=1}^B$, transform this to a static graph, and obtain the Laplacian eigenvectors of this graph to initialize the positional encoding for each node. If a node of $G$ does not belong to this graph then we initialize the positional encoding of that node with a zero vector. Here, let the "initial" positional encoding for node $u$ as $\mathbf{p}_u^1$. As we advance to the 2nd data batch, we first compute the approximated positional encoding for all nodes, $\widetilde{\mathbf{p}}^2$, as stated in Eq. 1, 2, based on $\mathbf{p}^1$. In this way, following Eq. 1, 2, we can iteratively compute the approximate positional encoding for the $k$-th batch,

$\widetilde{\mathbf{p}}^k$, based on the sequence of previous $L$ positional encodings, $\mathbf{p}_.^{k-L}, \ldots, \mathbf{p}_.^{k-1}$, corresponding to previous $L$ data batches.

For our current data batch (suppose this is the $k$-th batch), $\{u_j, v_j, \tau_j\}_{j=1}^B$, we now describe how to obtain the link predictions. For each query $(u_j, v_j, \tau_j)$, we obtain $\mathbf{h}_{u_j, N||E}^{\tau_j}$, as defined in Eq. 6, and $\mathbf{h}_{u_j, P}^{\tau_j}$, as stated in Eq. 8, using $\widetilde{\mathbf{p}}_{u_j}^k$. Finally, we combine $\mathbf{h}_{u_j, N||E}^{\tau_j}, \mathbf{h}_{u_j, P}^{\tau_j}$ to obtain the temporal representation at time $\tau_j$ for $u_j$, $\mathbf{h}_{u_j}^{\tau_j}$. Similarly, we obtain the temporal representation for $v_j$, $h_{v_j}^{\tau_j}$. Finally, following Eq. 10, we obtain the link prediction for $(u_j, v_j, \tau_j)$ using $\mathbf{h}_{u_j}^{\tau_j}, \mathbf{h}_{v_j}^{\tau_j}$.

In summary, we ideally would want to obtain the positional encoding of a node $u$, $\mathbf{p}_u^t \in \mathbb{R}^{d_P}$ for every temporal graph snapshot. However, for training efficiency, our implementation computes the positional encoding for the graph corresponding to a data batch.

