# OpenReview forum: "Is Attention All You Need for Temporal Link Prediction? A Lightweight Alternative via Learnable Positional Encoding and MLPs"
_ICLR.cc/2025/Conference — Submitted to ICLR 2025_

### Official Review · Reviewer_bMjN · 2024-10-21

**Soundness:** 3
**Presentation:** 3
**Contribution:** 3
**Rating:** 6
**Confidence:** 4

**Summary:**

The paper is about temporal link prediction, which is an interesting topic. The authors discover that some deliberately-designed simple positional encoding can enable MLPs to exploit attributed graph information to achieve SOTA performance than complex graph transformers. The paper is well written and well organized. However, there are several concerns in the current version of the paper that addressing them will increase the quality of this paper.

**Strengths:**

1 Cutting-edge research directions.

2 Clear writing logic.

3 Sufficient experimental results.

**Weaknesses:**

1 The authors may consider improving some statements in the abstract, especially in the contribution section.  Some contributions only state the process without stating the results (e.g. ‘‘verify the roles of ...’’ without saying what role they played), which is difficult to convey effective information in the abstract.

2 In my understanding, dynamic graphs can be divided into two categories, namely CTDG (Continuous-Time Dynamic Graph) and DTDG (Discrete-Time Dynamic Graph). CTDG corresponds to the concept of “temporal graph”, while the graph with static snapshots mentioned in the paper is actually DTDG. The author should further investigate related work and use professional terms accurately.

3 For this kind of dynamic graph learning based on static snapshots, will it be limited by the number of nodes? One thing that makes me curious is that the author mainly compares small-scale datasets, but also obtains good results on the large-scale datasets of TGB. So how does this learning model based on adjacency matrix adapt to large-scale datasets?

**Questions:**

As above.

---

> ### Author Response · Authors · 2024-11-23
> **Response from Authors**
>
> Thanks very much for your appreciation of our method’s research motivation, paper writing, and sufficient experimental results.
>
> Your suggestions are practical and beneficial for enhancing the quality of our paper. We have organized our responses in a Q&A format below.
>
> > **W1**. The authors may consider improving some statements in the abstract, especially in the contribution section. Some contributions only state the process without stating the results (e.g. ‘‘verify the roles of ...’’ without saying what role they played), which is difficult to convey effective information in the abstract.
>
> Thanks for the suggestion for highlighting our contribution. We have specified the concrete contributions in the abstract, the new texts are marked by orange color, and the updated paper is uploaded.
>
> > **W2**. In my understanding, dynamic graphs can be divided into two categories, namely CTDG (Continuous-Time Dynamic Graph) and DTDG (Discrete-Time Dynamic Graph). CTDG corresponds to the concept of “temporal graph”, while the graph with static snapshots mentioned in the paper is actually DTDG. The author should further investigate related work and use professional terms accurately.
>
> **First, please allow us to clarify the definition**.
>
> The definition of discrete time and continuous time dynamic graphs can be referred to the survey paper [1], where
>
> - Continuous Time Dynamic Graph (CTDG) is represented as $((v_2, v_5), t_1)$, $((v_1, v_2), t_2)$, …, as shown in its Example 2;
>
> - Discrete Time Dynamic Graph (DTDG) is represented as a set of $G_1$, $G_2$, …, $G_T$, where $G_t = (V_t, E_t)$ is the graph at snapshot $t$, $V_t$ is the set of nodes in $G_t$, and $E_t$ is the set of edges in $G_t$.
>
> In this viewpoint, our method is designed for continuous time.
>
> However, we can see a clear transformation between CTDG and DTDG: if we aggregate edges that happen at the same timestamp $t$ into a set, then it is the graph snapshot $G_t$, the example can be seen in Example 3 of the survey paper [1].
>
> **Second, we want to emphasize the reason why we bother to include “discrete” or “snapshot” in the paper**.
>
> Because the introduction and theoretical derivation on the snapshot level is much easier to be understood by the readers. Definitely, we can always use the term “a set of continuous time edges” every time when we do theoretical derivation and illustration, but it can be wordy and hurt the presentation. Since graph snapshot and relevant concepts are already existing in the community and cover the meaning, then we choose to follow them.
>
> Reference
>
> [1] Kazemi et al., Representation Learning for Dynamic Graphs: A Survey, JMLR 2020
>
> > **W3**. For this kind of dynamic graph learning based on static snapshots, will it be limited by the number of nodes? One thing that makes me curious is that the author mainly compares small-scale datasets, but also obtains good results on the large-scale datasets of TGB. So how does this learning model based on adjacency matrix adapt to large-scale datasets?
>
> Based on the above answer, we hope we reach the consensus that the computational procedure of our SimpleTLP is categorized into the CTDG. Also, because of the existence of transition between CTDG and DTDG, and DTDG is more friendly to less notation and easy understanding, we employ the snapshot definition for the illustration.
>
> Next, we want to give a detailed explanation of why our SimpleTLP can deal with the large-scale datasets and get good results.
>
> SimpleTLP can be adapted to large-scale graphs due to our efficient data batching procedure (more details can be found in Appendix D.2 and E.3). In other words, our data batching procedure allows SimpleTLP to process $B$ interactions at one time, where $B$ is the batch size of training. For an arbitrary batch, our computational complexity linearly scales with the number of unique nodes in the data batch, which is at most $2B$ nodes. Then before we proceed to a new data batch, we update the positional encodings via Eq (3), which takes $\mathcal{O}(|V|)$ time, where $|V|$ is the total number of nodes in the given temporal graph. As the runtime complexity for each process linearly scales with the number of nodes in the given temporal graph, or the number of unique nodes in a data batch, SimpleTLP is able to scale up, as theoretically analyzed in Section 4 and empirically demonstrated in Appendix D.3.

---

> > ### Comment · Reviewer_bMjN · 2024-11-25
> >
> > I appreciate the authors' response, I think the paper has merit and I will maintain my positive score.

---

> > > ### Author Response · Authors · 2024-11-25
> > >
> > > Thanks very much for your appreciation! We are glad to receive you helpful suggestions for the paper. We will add above answers to the paper and happy to answer any further question you may have.

---

### Official Review · Reviewer_D3ww · 2024-10-25

**Soundness:** 2
**Presentation:** 3
**Contribution:** 4
**Rating:** 6
**Confidence:** 4

**Summary:**

This paper introduces a positional encoding for graphs that integrates the graph structure, enabling it to be used with MLP models to outperform graph Transformers in temporal link prediction. The extensive experimental results demonstrate the effectiveness of the proposed positional embedding, even surpassing graph Transformers. This finding could significantly impact future research directions. However, the meaning of Theorem 1 requires clearer explanation, as detailed below.

**Strengths:**

1. The paper conducts extensive experiments to evaluate the performance, including experiments with multiple baseline methods and experimental settings.
2. Across multiple evaluation scenarios, although the method does not achieve the best results in every case, considering the number of datasets and baselines, as well as the significant improvements in the majority of scenarios, the overall experimental performance of this method is strong. Its notably lower average rank compared to other methods further supports this conclusion.
3. Table 3 is impressive, as it offers new insights that may inspire a shift in understanding for future research in the transformer-GNN field. This could significantly influence subsequent work in this area that the Transformer block not being necessary, where the well-designed positional encoding may be more important than people think.

**Weaknesses:**

1. The logical relationships in the introduction could be further developed. The introduction does not adequately discuss the significance of positional encoding for the transformer-based temporal GNN methods, nor does it clarify what can be learned. Consequently, the motivation for adding positional encoding to MLPs to level the approach against transformer-based methods presents a conceptual leap that requires further justification.
2. The meaning of Theorem 1 in Section 4 is unclear. First, it is uncertain whether the assumption that the temporal graph \( G \) is 'slowly changing' is valid or meaningful. Even if valid, this assumption greatly restricts the applicability of the method. Second, Equation 14 presents a relationship between the estimates of \( p \) at times \( t \) and \( t'' \); however, its connection to the model’s approximation or learning capability for \( p \) remains unknown. Overall, the significance of this theorem is unclear.

**Questions:**

1. How is equation (3) designed? What is the relationship of $p^t_u$ and $\tilde{p}^t_u$ during this design for the purpose of prediction and update?
2. What is the relationship of equation (3) and equation (8)? is that mean after the timestamp t is revealed, the positional encoding of (3) is updated following equation (8)?
3. In section 3.2, why is node encoding and link encoding modeled separately? Each interaction involving node \(u\) contains both the node features of \(u'\) and the edge features between \(u\) and \(u'\). What is the rationale for modeling these two aspects as separate variables and what would be the benefit of this approach?
4. Please explain the theorem 1 mentioned in weakness 2.

---

> ### Author Response · Authors · 2024-11-23
> **Response from Authors (Part I)**
>
> Thanks very much for your appreciation of our method’s empirical performance and extensive experimental designs.
>
> We appreciate your constructive suggestions, which helps refine our paper. Our responses are provided below in a Q&A format.
>
> > **Q1**. How is equation (3) designed? What is the relationship of $p^t_u$ and $\tilde{p}_u^t$ during this design for the purpose of prediction and update?
>
> **The short answer is**, $p^t_u$ is positional encoding of node $u$ to be learned at time $t$, and $\tilde{p}_u^t$  is the approximated positional encoding of node $u$ for time $t$. The approximation is the prerequisite to compute the true positional encoding. Their existence is because of the problem setting of temporal link prediction, as we are not allowed to observe the interactions at time $t$ when we make the prediction for $t$.
>
> **The detailed reasoning is stated below**.
>
> - Firstly, we state the formal usage of $p^t_u$ and $\tilde{p}^t_u$. When the model makes predictions for some queried links at a certain timestamp $t$, the model is not allowed to leverage the interactions or structural information occurring at time $t$. So $\tilde{p}^t_u$, the approximated positional encoding, is constructed only based on the positional encodings of previous timestamps (timestamps before $t$) and is leveraged to help the model “approximate” the structural information occurring at time t.
>
> - Secondly, the interaction update between them goes as follows. After processing all interactions (e.g., finished the predication) at time $t$, the model is then allowed to leverage the structural information occurring at time $t$, so we proceed to update the positional encoding that reflects the structural information including time $t$. Note that, earlier, we computed the approximated positional encoding only based on structural information at previous timestamps before $t$, $\bar{t}$ (where $\bar{t} < t$). Now that since the model’s knowledge is updated with new information occurring at time $t$, we combine this new information (structural information at time t) with the information occurring before $t$ captured by $\tilde{p}^t_u$ to obtain the positional encoding $p^t_u$.
>
> - In summary, $\tilde{p}^t_u$ is the approximated positional encoding at time t, without “peeking” at interactions occurring at t, while $p^t_u$ is the positional encoding that also accounts for information occurring at t.
>
> Therefore, given these reasonings, Eq (3) describes how to obtain the positional encoding $p^t_u$ AFTER the model updates its knowledge about interactions occurring at time $t$ by combining the approximated positional encoding of the node itself $\tilde{p}^t_u$ and the positional encoding of $u$’s neighboring nodes (now including those that interact with $u$ at time $t$ as the model’s knowledge is updated).
>
> To be more specific, through the lens of the message passing architecture, we can view Eq (3) as a way of neighbor aggregation ($\hat{q}^t_u$ is sum-aggregation over $u$’s temporal neighbors) and self-update (combining with $u$’s own approximated positional encoding $\tilde{p}^t_u$), thus in this way, the model is aware of new structural information occurring at time $t$ through this “Message Passing”-based update.

---

> > ### Author Response · Authors · 2024-11-23
> > **Response from Authors (Part II)**
> >
> > > **Q2**. What is the relationship of equation (3) and equation (8)? is that mean after the timestamp t is revealed, the positional encoding of (3) is updated following equation (8)?
> >
> > Eq (3) will not affect Eq (8), as Eq (3) is used to obtain the positional encoding $p^t_u$, while Eq (8) computes $h^t_{u, P}$ only based on $\tilde{p}^{t}_u$.
> >
> > To be more specific, we clarify the computational process of our SimpleTLP as follows.
> >
> > Given some queried links at time $t$, $(u, v, t)$, in order to make predictions for this queried link as in Eq (10), we will need to obtain $h^{t}_u$ and $h^{t}_v$. For each $u$, in order to obtain its temporal representation, $h^t_u$, we need the following procedure.
> >
> > - **Positional - related information**:
> >   - We sequentially perform Eq (1, 2) to first obtain the approximated positional encoding $\tilde{p}^t_{.}$, (“.” here denotes any node index that is involved in the computational process). Then, we proceed to Eqs (7, 8) to obtain $h^t_{u, P}$.
> >   - It is noteworthy that this process only involves the positional encodings at timestamps prior to $t$, $p_u^{t_i}$ (where $t_i < t$).
> >
> > - **Node / Link - related information**:
> >   - We sequentially execute Eqs (4 - 6) to obtain $h^t_{u, N || E}$.
> >
> > - **Temporal representation**:
> >   - As stated in Eq (9), we combine $h^t_{u, P}$ and $h^t_{u, N || E}$ to obtain the temporal representation, $h^t_u$.
> >
> > - **Link prediction**:
> >   - After obtaining $h^t_u, h^t_v$, SimpleTLP returns the prediction for the queried link as in Eq (10).
> >
> > - **Positional Encoding update**:
> >   - Now that the model has processed all interactions occurring at $t$, we can reveal these new interactions to SimpleTLP so that the model can move on and has more information to process future links occurring at time $t’$ ($t’ > t$).
> >   - In the future timestamp $t’$ ($t’ > t$), if we repeat this process, then we would need to access the positional encoding $p^t_u$ to obtain positional - related information (described in Eq (1)) for computing temporal representation at time $t’$.
> > Finally, before moving onto new timestamps, we update the positional information INCLUDING  time $t$ via Eq (3).
> >
> > > **Q3**. In section 3.2, why is node encoding and link encoding modeled separately? Each interaction involving node (u) contains both the node features of (u') and the edge features between (u) and (u'). What is the rationale for modeling these two aspects as separate variables and what would be the benefit of this approach?
> >
> > We compute node and edge representations separately, as node and edge features represent different views of information.
> >
> > - **From the perspective of physical meaning**
> >
> >   - The edge representation $h^t_{u, E}$ consists of historical edge raw features and time encodings, which indicates how frequently $u$ involves an interaction for the previous $K$ recent timestamps. In other words, $h^t_{u, E}$ not only aggregates neighboring edge features but also encodes how “active” a node is in forming interactions in recent timestamps, which is beneficial for link prediction, as if a node is highly active in forming interactions, then there is a high probability that it would also keep forming interactions in the future.
> >
> >   - The aggregated node representation $h^t_{u, N}$ only summarizes the node raw features of $u$’s temporal neighborhood in a time interval $[t - t_{gap}, t]$, encoding the “overall” node raw features of $u$’s temporal neighborhood.
> >
> > - **From the perspective of information aggregation manner**
> >
> >   - For any queried link at time $t$, $h^t_{u, N}$ can also aggregate information at timestamp $t$ (the node raw features of $u$ and its neighbors at timestamp $t$) while for link encodings (edge features and time encodings in $h^t_{u, E}$), the model is not allowed to receive information at timestamp $t$, so we find it more convenient to compute these two separately.
> >
> > - **From the perspective of datasets**
> >   - Some datasets do not support node raw features, or edge raw features, then in this case we simply set $s^t_u = 0$ or $e^t_{u, v} = 0$. So if we compute node and edge representations altogether, then we would combine the zero-vectors of raw node features with well-defined edge raw features (or vice versa), potentially making the input raw features less distinguishable (as now all input raw feature vectors share similar proportions, which are the zero-vectors).

---

> > > ### Author Response · Authors · 2024-11-23
> > > **Response from Authors (Part III)**
> > >
> > > > **Q4**. Please explain the theorem 1 mentioned in weakness 2.
> > >
> > > First of all, please allow us to introduce Theorem 1’s intuition.
> > >
> > > In short, it shows that when the temporal graph is slowly evolving with respect to time, the learned positional encoding at previous timestamp can well approximate the next close future timestamp’s positional encodings.
> > >
> > > **Theoretically**, the significance of Theorem 1 is to demonstrate the effectiveness of our proposed LPEs in the inductive setting, where future graph structural information is hard to observe, such that the “slowly changing” assumption for the coming snapshots is needed for this theoretical derivation. Moreover, defining a degree of change over temporal snapshots is an interesting but largely open question, which may involve the random matrix theory and relevant knowledge [1], and so far no clear definition, to the best of our knowledge. Therefore, _assuming the temporal smoothness [2], i.e., next snapshot will not change dramatically from the previous snapshot_, is a greedy quick-win solution for now, and we are more than happy to dive in this topic for our future research direction.
> > >
> > > **Empirically**, Theorem 1 does not necessarily suggest that we do not need to update the positional encodings. Actually, we update the positional encoding to avoid the possible unpredictable changes, the detailed procedures are as we answered in Q3, and the comprehensively outperformance demonstrates our effectiveness.
> > >
> > >
> > > Reference
> > >
> > > [1] Fu et al., Fairness-Aware Clique-Preserving Spectral Clustering of Temporal Graphs, WWW 2023
> > >
> > > [2] Chi et al., Evolutionary Spectral Clustering by Incorporating Temporal Smoothness, KDD 2007
> > >
> > > > **W1**. The logical relationships in the introduction could be further developed. The introduction does not adequately discuss the significance of positional encoding for the transformer-based temporal GNN methods, nor does it clarify what can be learned. Consequently, the motivation for adding positional encoding to MLPs to level the approach against transformer-based methods presents a conceptual leap that requires further justification.
> > >
> > > Based on the above Q&A, we in-depth introduce the technical design of our learnable positional encoding, and the theoretical meaning of learnable positional encodings.
> > >
> > > If you appreciate our explanation, we promise to add all the above discussion into the updated paper correspondingly and distillate the significant discovery into the introduction.

---

> > > > ### Comment · Reviewer_D3ww · 2024-11-26
> > > >
> > > > Thanks for the detailed response. After reading the explanation of the overall procedure in response Q2, the approach is much clearer.   Considering the challenges of making reasonable assumptions in dynamic graphs, the perspective of the theoretical analysis is also helpful for this field. I would increase my score accordingly.

---

> ### Author Response · Authors · 2024-11-26
>
> Thanks very much for your appreciation, and we are quite glad to know that your concerns are addressed!
>
> We promise to involve the above answer to our updated paper, and we are more than willing to answer any further question you may have. Thanks again!

---

### Official Review · Reviewer_XPj9 · 2024-10-27

**Soundness:** 3
**Presentation:** 2
**Contribution:** 2
**Rating:** 5
**Confidence:** 4

**Summary:**

The author focuses on link prediction tasks in temporal graphs, which have widespread applications in real-world systems. The authors propose a lightweight method that does not incorporate an attention mechanism: they first learn a positional encoding through discrete fourier transform and then apply a multi-layer perceptron (MLP) to encode these encodings for performance improvements. Additionally, the authors construct a loss function to supervise the positional encoding learning process. Extensive experimental results highlight the effectiveness of their proposed method, providing a thorough illustration of the model.

**Strengths:**

The paper reveals several strengths that deserve recognition:

* **Interesting Motivation.** The motivation for learning a positional encoding for the target node in temporal graphs, which can capture structural information, is intriguing.
* **Extensive Experiments.** The authors conduct a wide range of experiments to evaluate the effectiveness of their model, particularly across various link prediction settings, metrics, and negative sampling strategies.
* **Well-Written.** The main body of the paper is relatively well-organized and easy to read.

**Weaknesses:**

However, there are several weaknesses in the paper:

* **Ambiguous Motivation.** The authors claim to “design a module that is better than graph transformers” (Lines 64-65). However, why do you achieve this by finding a learnable positional encoding? Are there existing methods for positional encoding in static graphs? What makes static graph positional encoding insufficient for dynamic graphs? How does your proposed method differ from those used in static graphs?

* **Concerns about ‘Lightweight.’** The authors assert that their model is lightweight due to the absence of an attention mechanism, but **technically**, their model seems to be not appear lightweight. For example, the introduction of $L$ in positional encoding seems to introduce significant additional computation. From Table 18, $L$ appears to be quite large, potentially exceeding the number of samples per node. For a given node, the time complexity of the proposed method for generating its representation is $\mathcal{O}(Ln)$, where $n$ is the number of sampled neighbors. If an attention mechanism were used, the model complexity would be $\mathcal{O}(n^2)$, yet unfortunately, $L > n.$
* **Redundant Experiments.** Although the authors have designed numerous experiments to validate their model's effectiveness, these experiments do not address my key concerns. For instance, a deeper analysis of the important parameters $L$, $\alpha_{neg}$ in Equation 12 and $\alpha_{pe}$ in Equation 13 would be beneficial. Additionally, visualizations of the positional encodings or a case study could enhance the persuasiveness of your motivation.
* **Concerns about Efficiency.** The authors introduce an auxiliary loss to learn the positional encoding, which could significantly impact the model's convergence speed. However, the authors state in Table 10 that their model has a relatively fast convergence speed. What explains this discrepancy? The trade-off between effectiveness and efficiency (the selection of $L$) should be analyzed in the paper.
* Since PINT [1] also improves link prediction performance in temporal graphs through positional encoding, incorporating PINT—whether through empirical analyses or comparative discussions—could enhance your representation.

[1] Provably Expressive Temporal Graph Networks, NeurIPS 2022.

**Questions:**

Q1: How does your approach build upon or differ from existing methods for learnable positional encoding in static graphs?

Q2: Can you clarify the rationale behind your model's claim of being lightweight, considering the additional computations introduced by $L$?

Q3: What specific insights do you expect to gain from a deeper analysis of the parameters?

Q4: Can you explain the impact of the auxiliary loss on the convergence speed of your model, and how this relates to the overall efficiency of the method?

---

> ### Author Response · Authors · 2024-11-23
> **Response from Authors (Part I)**
>
> Thanks very much for your appreciation of our method’s interesting motivation, extensive experiments, and our paper’s clear writing.
>
> Your suggestions are actionable, and answering them helps improve the quality of our paper. Based on your concern about the weakness, we prepare the answer in the format of Q&A below.
>
> Thanks very much for your appreciation of our method’s interesting motivation, extensive experiments, and our paper’s clear writing.
>
> Your suggestions are actionable, and answering them helps improve the quality of our paper. Based on your concern about the weakness, we prepare the answer in the format of Q&A below.
>
> > **Q1**. How does your approach build upon or differ from existing methods for learnable positional encoding in static graphs?
>
> To the best of knowledge, the only work for learnable positional encoding is LSPE [1], which is designed for static graph neural network and graph classification tasks.
>
> In short, different from [1], our learnable positional encoding first attempts to graph data and also accounts for its temporal dependencies via Eq (1) and Eq (2).
>
> Therefore, next we would provide two directions of explanations for answering "What makes static graph positional encoding insufficient for dynamic graphs?" and "How does our proposed method differ from those used in static graphs?"
>
> **1. The insufficiency of static methods**
>
> On the one hand, if we apply positional encoding methods on static graphs (Laplacian, Random Walk Positional Encoding, etc.), everytime the model updates its knowledge, i.e., receive interactions of new timestamp, static methods would have to re-compute the positional encoding (based on Laplacian, Random Walk, etc.) which is a computational burden.
>
> On the other hand, applying static methods to the temporal setting relies on taking the time individually, for the task that requires the temporal patterns to be encoded, static methods can produce suboptimal results. Just take the temporal link prediction task, the prediction (or the label) indicating whether two entities would be connected or not depends on a series of historical behavior, where static methods are usually not capable to capture that, as shown in Table 1 of [2], Table 1 and Table 2 of [3], and Table 2 of [4].
>
> **2. Difference between our method and static methods**
>
> Motivated by point 1, we propose an efficient method to learn the positional encoding for future timestamps, without re-compute the positional encoding everytime the model’s knowledge is updated, avoiding the computational burden. To be more specific, LSPE proposes learnable positional encoding (LPE) as follows.
>
> On the one hand, LSPE initializes their LPE with well-defined positional encodings such as Laplacian, or Random Walk Positional Encoding, then obtains the final LPE via multiple layers of Message Passing. Their LPE of a node $u$ at the $(l + 1)-$th Message Passing layer is computed by aggregating the concatenation of (1) $u$’s adjacent edge encodings at $l-$th Message Passing layer and (2) LPEs (from $l-$th Message Passing layer) of $u$’s neighbors.
>
> On the other hand, our proposed LPEs only execute 1 round of aggregating the approximated positional encodings of neighboring nodes. Moreover, LSPE proposed the loss function for their LPE is designed to enforce the LPEs to have a centered and unit norm with zero mean, and better approximate the Laplacian eigenvectors. On the other hand, our loss function for our proposed LPE is designed to reflect the nature of node positional encodings (i.e., the more closer 2 nodes are, the more similar their positional encodings are) by minimizing the 2-norm LPEs difference of positive samples, and maximizing the 2-norm LPEs difference of negative samples.
>
>
> Reference
>
> [1] Dwivedi et al., Graph Neural Networks with Learnable Structural and Positional Representations, ICLR 2022
>
> [2] Trivedi et al., DyRep: Learning Representations over Dynamic Graphs, ICLR 2019
>
> [3] Xu et al., Inductive Representation Learning on Temporal Graphs, ICLR 2020
>
> [4] Wang et al., TCL: Transformer-based Dynamic Graph Modelling via Contrastive Learning, arXiv 2021

---

> > ### Author Response · Authors · 2024-11-23
> > **Response from Authors (Part II)**
> >
> > > **Q2**. Can you clarify the rationale behind your model's claim of being lightweight, considering the additional computations introduced by $L$?
> >
> > In general, we want to clarify two aspects: (1) the introduction of $L$ is indispensable in temporal graph representation learning, and (2) our method already reduces the large time complexity given the necessary existence of $L$.
> >
> > - First of all, the definition $L$ is the number of past timestamps, i.e., $L$ most recent distinct timestamps prior to the current time $t$. In the temporal graph learning setting like temporal link prediction, the label (i.e., whether two entities connect or not in the future) largely depends on their historical behaviors, such that looking back is unavoidable. For example, SOTA methods such as GraphMixer, DyGFormer, and FreeDyG, obtain temporal node representation for a node $u$ by leveraging information from $u$’s most recent historical neighbors / interactions.
> >
> > - Second, we want to clarify that the complexity of obtaining the representation for a node $u$ at time $t$, given that we aggregate information from $u$’s $n$ temporal neighbors, is not $\mathcal{O}(Ln)$, but it only takes $\mathcal{O}(n)$. Furthermore, our empirical running time of each epoch and the model convergence speed is superior to SOTA lightweight baselines, as shown in Table 10 in Appendix D.3.
> >
> > Next, we will disentangle why our time complexity of obtaining node embedding is not related with $L$ but linearly with $n$, i.e., number of temporal neighbors.
> >
> > - We can achieve this by first pre-compute the approximated positional encoding (via Eq (1, 2)) for all nodes in the graph, then proceed to compute temporal representation for a batch of queried links by simply retrieving the $n$ approximated positional encodings that we just computed earlier for $n$ neighbor nodes of $u$.
> >
> > - Then, pre-computation of approximated positional encodings takes $\mathcal{O}((L logL) |V|)$, where $|V|$ is the total number of nodes in the temporal graph, and for computing the temporal representation for node $u$, we just need to retrieve $n$ approximated positional encodings (where $n$ is the number of sampled neighbors for $u$), which only takes $\mathcal{O}(n)$ time, giving our SimpleTLP the advantage of being lightweight.
> >
> > For more details regarding our complexity, please see our Complexity Analysis in Section 4, where we prove that the time complexity linearly scales with the number of nodes in the graph, and our complexity does not contain the term $\mathcal{O}(LK)$, where $K$ is equivalent to $n$ in your context, and $n$ in our Section 4 is the number of queried nodes.
> >
> > Moreover, with the attention mechanism, even if we first pre-compute the attention scores, which takes $\mathcal{O}(|V| n^2)$ time, then for each queried node, we can simply retrieve the attention scores for its $n$ sampled neighbors, taking $\mathcal{O}(n)$ time. However, in this case, our pre-computation of approximated positional encoding, taking $\mathcal{O}(|V| L \log L)$, is still more lightweight than the pre-computation using attention, which is $\mathcal{O}(|V| n^2)$.
> >
> > Finally, when we consider larger datasets, and we might need to sample more neighbors, i.e $n$ could become large, then in this case, our SimpleTLP has the advantage over the attention mechanism, as the attention mechanism would take $\mathcal{O}(n^2)$ for each queried node.

---

> ### Author Response · Authors · 2024-11-23
> **Response from Authors (Part III)**
>
> > **Q3**. What specific insights do you expect to gain from a deeper analysis of the parameters?
>
> We understand your concern. As you requested, we did extra parameter analysis experiments for the parameters $L$, $\alpha_{neg}$, and $\alpha_{pe}$.
>
> **[Parameter Analysis for $L$]**
>
> We show the parameter analysis for $L$ as follows on Can. Parl. dataset below.
>
> For the below table we can observe that our proposed SimpleTLP just relies on looking back a few historical information to make the correct information, as $L$ increases, the performance can be downgraded, and the best performance exists between 20 and 50, which discovery
> - demonstrates that the near past history plays a more important role in the current decision making than the long past history,
> - addresses your concern again that our proposed SimpleTLP method may needs $L$ to be quite large and effect the efficiency.
>
> | Dataset |  | Transductive AP | Transductive AUC | Inductive AP | Inductive AUC |
> | -- | -- | -- | -- | -- | -- |
> | Can. Parl. | $L = 20$ | 98.24 ± 0.11 | **98.97 ± 0.06** | **92.25 ± 0.24** | 95.06 ± 0.11
> | |$ L = 50$ |  **98.26 ± 0.06** |98.94 ± 0.03 |92.15 ± 0.34 | **95.11 ± 0.17** |
> | |$ L = 100$ | 98.11 ± 0.21 | 98.85 ± 0.11 |  92.16 ± 0.98 | 95.25 ± 0.56 |
> | |$ L = 200$ |  97.04 ± 0.22 |  98.37 ± 0.09 | 91.96 ± 0.59 |  95.19 ± 0.38 |
>
> **[Parameter Analysis for $\alpha_{neg}$]**
>
> We present the parameter analysis for $\alpha_{neg}$ on Can. Parl. dataset as follows:
>
> In general, the below table shows
> - our proposed method SimpleTLP is robust towards different choices of $\alpha_{neg}$,
> - usually a smaller weight of negative pairs can help the model concentrate more on the positive pairs and boost the performance.
>
> | Dataset |  | Transductive AP | Transductive AUC | Inductive AP | Inductive AUC |
> | -- | -- | -- | -- | -- | -- |
> | Can. Parl. | $\alpha_{neg} = 0.05$ | 98.26 ± 0.13 |98.98 ± 0.05 |92.24 ± 0.09 |95.09 ± 0.11 |
> | | $\alpha_{neg} = 0.1$ | **98.28 ± 0.09** | **98.98 ± 0.05** | **92.35 ± 0.21** | **95.09 ± 0.09** |
> | | $\alpha_{neg} = 0.3$ | 98.24 ± 0.11 | 98.97 ± 0.06 | 92.25 ± 0.24 | 95.06 ± 0.11
> | | $\alpha_{neg} = 0.5$ |  98.28 ± 0.13 |98.97 ± 0.05 | 91.48 ± 0.23 |94.75 ± 0.16 |
> | | $\alpha_{neg} = 0.95$ | 98.24 ± 0.18 |98.95 ± 0.09 |91.09 ± 0.49 |94.53 ± 0.32 |
>
> **[Parameter Analysis for $\alpha_{pe}$]**
>
> We present the parameter analysis for $\alpha_{pe}$ on Enron dataset as follows:
>
> $\alpha_{pe}$ stands for the weight of the loss function of learnable positional encoding. As shown in the following table, we can observe that
> - our SimpleTLP is robust towards different choices of $\alpha_{pe}$,
> - a considerable weight, e.g., 0.5, is optimal, too large or too small weight can induce suboptimal results..
>
> | Dataset |  | Transductive AP | Transductive AUC | Inductive AP | Inductive AUC |
> | -- | -- | -- | -- | -- | -- |
> | Enron | $\alpha_{pe}$ = 0.1 | 92.81 ± 0.82 |95.32 ± 0.38 |86.49 ± 1.52 |90.91 ± 0.61 |
> | | $\alpha_{pe}$ = 0.5 | 93.96 ± 0.36	| **95.98 ± 0.23** |	**89.17 ± 0.88** |	**92.30 ± 0.57**|
> | | $\alpha_{pe}$ = 0.7 | 93.94 ± 0.10 |95.90 ± 0.06 |88.92 ± 0.39 | 92.12 ± 0.22 |
> | | $\alpha_{pe}$ = 0.95 | **94.00 ± 0.28** | 95.92 ± 0.17 | 88.67 ± 0.34 | 91.91 ± 0.30 |
>
> Moreover, we are very interested in your suggested visualization experiments and more than willing to demonstrate it to improve our paper. Could please describe what kind of specific experimental settings you are referring to?
>
>
>
> > **Q4**. Can you explain the impact of the auxiliary loss on the convergence speed of your model, and how this relates to the overall efficiency of the method?
>
> **Theoretically**, auxiliary loss does not always negatively affect the convergence speed of the model. As demonstrated in [1], the authors state that a suitable auxiliary loss can accelerate the convergence speed.
>
> **Empirically**, we did the convergence speed comparison with SOTA baselines, as shown in Table 10 of Appendix D.3, no matter for _the running time of each epoch_ or _the number of epochs before convergence_, our method is faster than those SOTA baselines.
>
> Reference
>
> [1] Du et al., Adapting Auxiliary Losses Using Gradient Similarity, arXiv 2020

---

> ### Author Response · Authors · 2024-11-23
> **Response from Authors (Part IV)**
>
> > Additional experiment of PINT as a baseline.
>
> As requested, we incorporate PINT through empirical analysis and comparative discussion as follows.
>
> **Empirical Comparison**
>
> We present the empirical results for PINT as follows.
>
> | Dataset | Method | Transductive AP | Transductive AUC | Inductive AP | Inductive AUC |
> | -- | -- | -- | -- | -- | -- |
> | Enron | PINT | 81.60 ± 0.67 | 84.50 ± 0.60 | 67.01 ± 2.76 | 70.41 ± 2.34 |
> | | SimpleTLP (Ours) | **93.96 ± 0.36** | **95.98 ± 0.23** | **89.17 ± 0.88** | **92.30 ± 0.57**
> |UCI | PINT | 95.77 ± 0.11 | 94.89 ± 0.15 | 94.14 ± 0.05 | 92.51 ± 0.07
> | | SimpleTLP (Ours)| **96.67 ± 0.47** | **97.62 ± 0.23** | **94.60 ± 0.41** | **95.88 ± 0.17**
>
> Currently, we have obtained the results for UCI and Enron as they are small datasets. PINT also evaluates their method on larger datasets, which are Reddit, Wikipedia, and LastFM, but PINT has not finished its empirical evaluation on these large datasets, so now we only present the results for small datasets. We will post PINT’s results when they are ready. We adopt PINT’s official implementation (via the Github link provided in the PINT paper) and use the exact hyperparameters presented in the PINT paper to reproduce their results on these datasets. Moreover, PINT and SimpleTLP use the same data split.
>
> As shown in the above table, our SimpleTLP outperforms PINT in all experimental settings (transductive and inductive).
>
> **Comparative Discussion**
>
> PINT proposes relative positional features that count the number of temporal walks of a given length between 2 nodes. To be more specific, temporal walks are defined as follows. An $(L- 1)$ temporal walk is $W = \{(w_1, t_1), (w_2, t_2), \dots, (w_L, t_L)\}$, where $t_1 > t_2 > \dots t_L$ and $(w_{i - 1}, w_i, t_i)$ is an interaction in the give temporal graph. In summary, PINT proposes a relative positional encoding that represents pair-wise node “distance” (here the “distance” is the number of $L$-length temporal walks, assuming $L$ is given). On the other hand, our SimpleTLP learnable positional encodings (LPE) fall into the category of global positional encodings, which encode the position of the node relative to the global structure of the graph.

---

> ### Comment · Reviewer_XPj9 · 2024-11-25
>
> Thank you very much for the author's rebuttal. You have addressed most of the concerns, but I still have some questions regarding Q2.
>
> 1. You generate positional encoding by selecting $L$ different historical event timestamps. However, in dynamic graphs, a large number of interactions can occur simultaneously. Additionally, I noticed that the values of L you chose in Q3 are relatively large (even up to 100). Does LSPE maintain high efficiency and performance in dynamic graphs with a large number of interactions occurring at the same timestamps?
> 2. LSPE pre-computes the positional encoding, making the model to be not end-to-end. However, the authors mentioned that their method is "learnable." How is this part optimized, and what is the objective?
> 3. Default model parameters should be included in the section of model configuration.
>
> After re-reading your manuscript, I noticed that most of the rebuttal content and experimental results (not limited to the response to me) have not been incorporated into the paper. I believe it would be great to see this information included in your revised paper.

---

> > ### Author Response · Authors · 2024-11-25
> > **Addressing the further questions (Part I)**
> >
> > We are quite glad to know that most of your concerns are addressed.
> >
> > We are also more than happy to address your further questions as follows.
> >
> > > **Q1**. You generate positional encoding by selecting $L$ different historical event timestamps. However, in dynamic graphs, a large number of interactions can occur simultaneously. Additionally, I noticed that the values of L you chose in Q3 are relatively large (even up to 100). Does LSPE maintain high efficiency and performance in dynamic graphs with a large number of interactions occurring at the same timestamps?
> >
> > Thanks for the consideration. From Q3 of the last round, we can see that **$L$ does not have to be quite large for achieving the best performance**, because the near past history usually contains more information for the current time decision process.
> > Moreover, we understand your concern that **some extreme cases can happen**, for example, at a certain timestamp, a surge of connection events may happen. For this case, we believe that our SimpleTLP can also better handle it than SOTA baselines, and we will explain the detailed reasoning from two aspects.
> >
> > **From the theoretical perspective**, suppose we are given $E$ queries links {$(u_i, v_i, t)$} (these $E$ links occurred at the same timestamp $t$). Our SimpleTLP first considers the unique nodes among all nodes involved in the given $E$ queried links (i.e, unique nodes among $u_1, \dots, u_E, v_1, \dots, v_E)$. Suppose there are $n$ unique nodes, and we index them as $c_1, \dots, c_n$. Then, we compute the node temporal representation for these $n$ nodes by first extracting information for (1) node, (2) link, and (3) positional related information and then compute the node representations via Eqs (1 - 2, 4 - 9). As analyzed in our Complexity Analysis (Section 4), the computation process for obtaining the temporal node representations for these $n$ nodes $c_1, \dots, c_n$ takes $\mathcal{O}(n(t_{gap} + K + L \log(L)))$, which linearly scales with $n$. In short, despite the large number of interactions, our method’s complexity only linearly scales with the number of unique nodes.
> >
> > **From the empirical perspective**, as shown in Table 5 in Appendix C, we evaluate our SimpleTLP on datasets that have a large number of interactions occurring at the same timestamps. For example, *UN Vote* dataset has a total **1,035,742** links and **72** unique timestamps, so in average, there could be **14,385** (this number is much larger than the $L$ value used) interactions occurring at the same timestamp, or *Flights* dataset has **1,927,145** links and **122** unique timestamps, resulting in an average of **15,796** (again, this number is also much larger than the $L$ value used) interactions occurring at the same timestamps.

---

> > > ### Author Response · Authors · 2024-11-25
> > > **Addressing the further questions (Part II)**
> > >
> > > > **Q2**. LSPE pre-computes the positional encoding, making the model to be not end-to-end. However, the authors mentioned that their method is "learnable." How is this part optimized, and what is the objective?
> > >
> > > Firstly, please allow us to clarify the definition of positional encoding pre-computation process.
> > >
> > > - Suppose we are given $E$ queried links {$(u_i, v_i, t)$}, then instead of obtaining the approximated positional encoding for all $u_1, \dots u_E, v_1, \dots v_E$, we first compute the approximated positional encoding for the unique nodes among $u_1, \dots, u_E, v_1, \dots, v_E$ (suppose the number of unique nodes is $n$) then we proceed to compute the temporal node representations (defined in Eq (9)), and we refer to this process as “pre-computation”.
> > >
> > > - The reason we call it “pre” is because we execute this process before diving into the processing of the queried links.
> > >
> > >   - Technically, we only need to compute approximated PEs for these $n$ unique nodes. However, in practice, for the sake of efficient CUDA computations, we would like to keep $n$ consistent (because $n$ could be varied for different batches of queried links), so instead of “pre-computing” approximated PEs for these $n$ unique nodes, we “pre-compute” approximated PEs for all $|V|$ nodes in the temporal graph (where $|V|$ is the total number of nodes in the temporal graph). To be more specific, computing the approximated PEs involving matrix multiplications, so in order to better utilize CUDA, we want the number of unique nodes to be consistent through different data batches of queried links, and thus compute $|V|$ approximated PE vectors.
> > >
> > >   - This is a more effective way to implement the model, as now we only need to retrieve $n$ PE vectors from the aforementioned “pre-computation” process, and proceed with computing the temporal representation for nodes involved in the queried links. Finally, as stated in our Optimization Section (Section 3.3), the temporal node representations and approximated PEs are optimized via the loss function (Eq (13)).
> > >
> > > In summary, “pre-computation” is **just an effective way to implement the model**, and this “pre-computation” process **does not “freeze” or excluding any representation from being optimized** along with the model.

---

> > > > ### Author Response · Authors · 2024-11-25
> > > > **Addressing the further questions (Part III)**
> > > >
> > > > > **Q3**. Default model parameters should be included in the section of model configuration
> > > >
> > > > We reported all hyper-parameters in Appendix F.2.
> > > >
> > > > This includes (1) architecture’s parameters such as $L, K, time_{gap}, \alpha_{pe}, \alpha_{neg}$, dimensions of representations, .etc, (2) Adam optimizer’s parameters (learning rate,. etc), and (3) computational resource configuration (GPUs and compilers information).
> > > >
> > > > It is also noteworthy that our model does not stack multiple layers of neural architectures, and the implementation for temporal representation is exactly similar to the process in Section 3.
> > > >
> > > > > **Adding new contents to the paper**
> > > >
> > > > Thanks for your appreciation of our new contents.
> > > >
> > > > After gaining your acknowledgement and confirmation, we added them for sure. The updates are marked by orange color, and the new paper is uploaded.
> > > >
> > > > The updates are listed below for your information.
> > > > - We present comparative discussion and empirical comparison between our method and PINT in Appendix C.
> > > > - We report additional parameter analysis for $L, \alpha_{neg}, \alpha_{pe}$ in Appendix E.5.3, E.5.4, and E.5.5, respectively.

---

> ### Author Response · Authors · 2024-11-27
> **Gentle Reminder from Authors: for the PDF Upload Due**
>
> Dear Reviewer XPj9,
>
> We sincerely thank you again for your appreciation of our last-round response answers.
>
> After gaining your appreciation and confirmation, we had added the new experiments with detailed descriptions to the paper, marked by the orange color. The new paper had been uploaded already.
>
> Since the due for uploading the pdf is close to the end in a few hours, i.e., 11:59 PM AOE Time 11/27, may we gently and kindly remind you to have a look at our updates?
>
> If there are any updates you suggest, we can make them before the upload due date.
>
> Thanks very much!
>
> Authors

---

> ### Author Response · Authors · 2024-11-30
> **Response from Authors (Part V)**
>
> Dear Reviewer XPj9,
>
> As requested, the new baseline PINT’s empirical results on other larger datasets are ready and we report them in the following table.
>
> | Dataset | Method | Transductive AP | Transductive AUC | Inductive AP | Inductive AUC |
> | -- | -- | -- | -- | -- | -- |
> | Enron | PINT | 81.60 $\pm$ 0.67 | 84.50 $\pm$ 0.60 | 67.01 $\pm$ 2.76 | 70.41 $\pm$ 2.34 |
> | | SimpleTLP (Ours) | **93.96 $\pm$ 0.36** | **95.98 $\pm$ 0.23** | **89.17 $\pm$ 0.88** | **92.30 $\pm$ 0.57**
> |UCI | PINT | 95.77 $\pm$ 0.11 | 94.89 $\pm$ 0.15 | 94.14 $\pm$ 0.05 | 92.51 $\pm$ 0.07
> | | SimpleTLP (Ours)| **96.67 $\pm$ 0.47** | **97.62 $\pm$ 0.23** | **94.60 $\pm$ 0.41** | **95.88 $\pm$ 0.17**
> | Wikipedia | PINT | 98.40 $\pm$ 0.08 | 98.22 $\pm$ 0.04 | 97.16 $\pm$ 0.09 | 96.75 $\pm$ 0.11
> | | SimpleTLP (Ours)| **99.34 $\pm$ 0.04** | **99.48 $\pm$ 0.03** | **99.15 $\pm$ 0.04** | **99.30 $\pm$ 0.03**
> | Reddit | PINT | 98.23 $\pm$ 0.08 | 98.16 $\pm$ 0.08 | 90.72 $\pm$ 0.87 | 90.45 $\pm$ 0.75
> | | SimpleTLP (Ours)| **99.37 $\pm$ 0.04** | **99.49 $\pm$ 0.03** | **98.02 $\pm$ 0.09** | **98.49 $\pm$ 0.06**
> | LastFM | PINT | 74.42 $\pm$ 2.96 | 73.83 $\pm$ 2.13 | 78.62 $\pm$ 3.29 | 77.27 $\pm$ 2.00 |
> | | SimpleTLP (Ours)| **96.06 $\pm$ 0.30** | **97.52 $\pm$ 0.17** | **96.25 $\pm$ 0.43** | **97.66 $\pm$ 0.21**
>
> As shown in the Table, the observation is consistent with our previous discovery, and our SimpleTLP outperforms PINT. As always, we promise to add them to our paper.

---

> ### Author Response · Authors · 2024-12-02
> **Gentle Reminder from Authors: for the Discussion Due**
>
> Dear Reviewer XPj9,
>
> Thanks very much for your time and helpful review!
>
> As requested, we have finished additional theoretical analysis and empirical experiments, and we updated the pdf file with colored parts. We are quite glad to learn that you acknowledge that most of your concerns have been addressed already.
>
> Moreover, we appreciate that you further propose three sub-questions for Q2 in the first round. Regarding that, we further carefully prepared the extended answer from theoretical and empirical perspectives.
>
> We enjoyed exchanging ideas with you very much. Since the discussion period will close in a few hours, we would like to gently ask if you need any further discussions for our second-around answers for your re-evaluation. We would wholeheartedly answer any of them until the last minute.
>
> Thanks very much!
>
> Authors

---

### Official Review · Reviewer_Ysht · 2024-11-05

**Soundness:** 3
**Presentation:** 4
**Contribution:** 3
**Rating:** 5
**Confidence:** 4

**Summary:**

The authors propose learnable position encoding (LPE) approach and node-link-positional encoding approach: the prior applies discrete Fourier transform on the (spatial) positional embeddings which are essentially the eigenvectors of graph Laplacian matrix for every graph snapshots, while the latter aggregates the information of node features and edge features in the neighborhood. Empirical results show the SOTA performance of the proposed SimpleTLP model.

**Strengths:**

1. This paper is well written and the experiments are solid.

2. The theory looks sound, but I did not check it very carefully.

3. The research problem is interesting. When it comes to the information aggregation in the neighborhood, either Mean-pooling or MLP is used.

**Weaknesses:**

1. The idea of positional encoding and discrete Fourier transform is not new in the field of dynamic graph learning. This paper applies DFT only to the positional embeddings {$\mathbf{p}_u^t$}, rather than to the final representations {$\mathbf{h}_u^t$}. I am curious about the benefits of doing so.

2. In the viewpoint of graph signal processing, the eigenvectors of graph Laplacian matrix represents the basis of low-frequency signals on the graph [1]. When the graph structures evolve over time, the basis would probably change, then it is not clear to me what kind of information the filter in Eq. (1) can learn. Please remind that the high-frequency noise in for example $\mathcal{G}^{t_i}$ might correspond to low-frequency signal in another snapshot $\mathcal{G}^t_j$ [2].

3. It seems that p_u^t  in Eq.(3) encodes the spatial positional information, h_{u,N}^t encodes the node features, and h_{u,E}^t encodes the edge features and the time intervals between recent interactions. I am not sure whether  h_{u,N}^t is capable of learning sequential patterns in the neighborhood. This is very important for personalized recommendation. To be specific, the sequence order of historical items (i.e., 1-hop neighboring nodes) for a target user (i.e., query node) will reflect her/his future behaviors.

4. Theorem 1 states that the positional embeddings would similar to each other when temporal graph is slowly changing, then one raising concern is whether DFT is necessary.  Personally, I believe that it would be more interesting to see that LPE has the ability to learn abrupt changes.

Minor comments are summarized as follows:

1. It would be better to recall the definitions of $f_T$ and $\mathbf{e}$ in line 247.

2. It seems that the computational complexity of eigenvalue-decomposition is not mentioned in complexity analysis. As far as I know, it is quite expensive to achieve orthogonalized (approximate) eigenvectors. This would largely limit the application of the proposed model. (I read D.6 which has addressed part of this issue.)


[1]  Sampling in paley-wiener spaces on combinatorial graphs. Transactions of the American Mathematical Society.

[2] Sparse representation on graphs by tight wavelet frames and applications. Applied and Computational Harmonic Analysis.

**Questions:**

Please see the comments above. Thanks.

---

> ### Author Response · Authors · 2024-11-23
> **Response from Authors (Part I)**
>
> Thanks very much for your review and appreciation of our method’s theoretical soundness, neural architecture design, and solid experiments!
>
> Addressing your concerns improved the quality of our paper, and we prepared the answer in the Q&A format below.
>
> > **W1**. The idea of positional encoding and discrete Fourier transform is not new in the field of dynamic graph learning. This paper applies DFT only to the positional embeddings {$\{\mathbf{p}\_{u}^{t}\}$}, rather than to the final representation {$\{\mathbf{h}\_{u}^{t}\}$}. I am curious about the benefits of doing so.
>
> First of all, positional encoding is a necessary prerequisite for the self-attention mechanism, which provides the position information in the sequence that will be attended. Instead of setting the positional encoding as hand-crafted and fixed [1], our method and efforts focus on (1) how to propose a learnable positional encoding that is expected to be more expressive than pre-defined, hand-crafted, and fixed positional encodings, (2) and how to extend it to the time-evolving setting, such that the positional encoding can convey temporal information and be robust across timestamps. To the best of our knowledge, these topics have not been studied by previous works.
>
> Second, Discrete Fourier Transform (DFT) is a classic concept. However, to the best of our knowledge, the only previous work that applies DFT in dynamic graph representation learning is FreeDyG [1], and it has been discussed in detail and set as the SOTA baseline for 13 classic datasets and with 8 other algorithms in both transductive and inductive settings using 3 different sampling strategies, where our SimpleTLP achieves the best performance consistently.
>
> The reason we can achieve the above performance is that we are not just simply using DFT, but we adapt it in a principled way to the dynamic graph setting, as shown in Eqs (1 - 2). To be more specific, our learnable positional encodings are constructed based on the reasoning that the evolving graph’s structure is associated with temporal dependencies. On the other hand, node / edge raw features are hand-crafted or domain-specific injected features, so these features are not necessarily related to temporal dependencies. Moreover, some datasets do not support node / edge features. Therefore, we only apply DFT on positional-related information so that we can effectively capture the aforementioned temporal dependencies
>
> Reference
>
> [1] Dwivedi et al., A Generalization of Transformer Networks to Graphs, DLG-AAAI 2021.
>
> > **W2**. In the viewpoint of graph signal processing, the eigenvectors of graph Laplacian matrix represents the basis of low-frequency signals on the graph [1]. When the graph structures evolve over time, the basis would probably change, then it is not clear to me what kind of information the filter in Eq. (1) can learn. Please remind that the high-frequency noise in for example $\mathcal{G}^{t_{i}}$ might correspond to low-frequency signal in another snapshot
> $\mathcal{G}^{t}$ [2].
>
>
> To the best of our knowledge, not all eigenvalues of the graph Laplacian correspond to low-frequency signals on the graph, but actually small eigenvalues correspond to low-frequencies while larger eigenvalues correspond to higher frequencies.
>
> We understand your concerns about whether learnable positional encoding can fit the time-evolving setting well. Therefore, we give the following in-depth explanation.
>
> **Theoretically**, we would like to provide more elaborations on the significance of our learnable filter in Eq (1). We first apply DFT on $u$’s sequence of approximated positional encodings {$p^{t’}_u$} (where $t'_1 \leq t' \leq t'_L$), and the the learnable filter filters out high-frequency noise across dimensions of the input approximate positional encodings, and finally the Inverse DFT transformed the filtered positional encodings back to the original domain. It is noteworthy that we are apply DFT on the sequence of approximated positional encodings of the node itself (i.e, the sequence consists of previous approximated positional encodings of the same node), instead of basis, formed by the positional encodings of every node, of the whole graph.
>
> **Empirically**, given (1) 13 classic dynamic graph datasets and with 10 algorithms in both
> transductive and inductive settings using 3 different sampling strategies and (2) the open worldwide large-scale dynamic graph benchmark, our method shows superior performance.

---

> ### Author Response · Authors · 2024-11-23
> **Response from Authors (Part II)**
>
> > **W3**. It seems that p_u^t in Eq.(3) encodes the spatial positional information, h_{u,N}^t encodes the node features, and h_{u,E}^t encodes the edge features and the time intervals between recent interactions. I am not sure whether h_{u,N}^t is capable of learning sequential patterns in the neighborhood. This is very important for personalized recommendation. To be specific, the sequence order of historical items (i.e., 1-hop neighboring nodes) for a target user (i.e., query node) will reflect her/his future behaviors.
>
> $h^t_{u, N}$ is capable of capturing sequential patterns.
>
> In the context of personalized recommendation, if a node $u$ interacts with $v$ many times (for example, a user purchases an item many times) then the raw node feature of $v$ is emphasized in $h^t_{u, N}$.
>
> Similarly, $u$’s node feature is also emphasized in $h^t_{v, N}$, making $h^t_{u, N}$ and $h^t_{v, N}$ closer in the embedding space.
>
> > **W4**. Theorem 1 states that the positional embeddings would similar to each other when temporal graph is slowly changing, then one raising concern is whether DFT is necessary. Personally, I believe that it would be more interesting to see that LPE has the ability to learn abrupt changes.
>
> In brief, DFT is necessary for the real-world temporal link prediction setting and is the key point for our model achieving the comprehensive outperformance, as shown in LPE ablation study in Appendix D.4.
> To further address your concern, we would like to go through Theorem 1 intuitively for you.
>
> In short, it shows that when the temporal graph is slowly evolving with respect to time, the learned positional encoding at previous timestamp can well approximate the next close future timestamp’s positional encodings.
>
> **Theoretically**, the significance of Theorem 1 is to demonstrate the effectiveness of our proposed LPEs in the inductive setting, where future graph structural information is hard to observe, such that the “slowly changing” assumption for the coming snapshots is needed for this theoretical derivation. Moreover, defining a degree of change over temporal snapshots is an interesting but largely open question, which may involve the random matrix theory and relevant knowledge [1], and so far no clear definition, to the best of our knowledge. Therefore, _assuming the temporal smoothness [1], i.e., next snapshot will not change dramatically from the previous snapshot_, is a greedy quick-win solution for now, and we are more than happy to dive in this topic for our future research direction.
>
> **Empirically**, Theorem 1 does not necessarily suggest that we do not need to update the positional encodings. Actually, we update the positional encoding to avoid the possible unpredictable changes, the detailed procedures are as we answered in Q3, and the comprehensively outperformance demonstrates our effectiveness.
>
> Reference
>
> [1] Chi et al., Evolutionary Spectral Clustering by Incorporating Temporal Smoothness, KDD 2007
>
> ---
>
> **[Minor comments are also addressed below]**
>
> > **M1**. It would be better to recall the definitions of $f_{T}$ and $\mathbf{e}$ in line 247.
>
> $f_T$ denotes the time encoder and $e$ denotes the edge feature. The explanation is added to the new paper and marked by orange color, and the updated paper is uploaded.
>
>
> > **M2**. It seems that the computational complexity of eigenvalue-decomposition is not mentioned in complexity analysis. As far as I know, it is quite expensive to achieve orthogonalized (approximate) eigenvectors. This would largely limit the application of the proposed model. (I read D.6 which has addressed part of this issue.)
>
> Thanks for reading our Appendix D.6 for the robust performance of our SimpleTLP across different positional encodings.
>
> No matter in the main body of the paper or Appendix D.6, the positional encodings are just initial inputs to trigger our method as stated in lines 028, 077, and 216. This computation is pre-computed and does not need to be recomputed in our method as time goes by. On the contrary, this is one of contributions to learn and update the positional encodings in the time-evolving setting.
>
> Since it is out of the scope of the main story of the proposed SimpleTLP and can be replaced with other positional encodings with the similar results, that's why we view it as a dispensable condition of executing SimpleTLP and did not include into the time complexity analysis of our SimpleTLP.

---

> ### Comment · Reviewer_Ysht · 2024-11-28
> **Discussion on Q2**
>
> First apologize for the late response, and appreciate for the authors' responses.
>
> Let us take a simple example to understand Q2: $L^{t_i}=[v_0, \dots, v_n]^T \textrm{diag}(\lambda_0, \dots, \lambda_n) [v_0, \dots, v_n]$ represents the Laplacian matrix at time $t_i$ where $\lambda_0\le\dots\le\lambda_n$, where $L^{t_j}=[v_n, \dots, v_0]^T \textrm{diag}(\lambda_0', \dots, \lambda_n') [v_n, \dots, v_0]$ signifies the Laplacian matrix at time $t_j$ where $\lambda_0'\le\dots\le\lambda_n'$. In this case, a graph signal $\mu = v_0$ corresponds to the low-frequency signal on graph $G^{t_i}$, but it corresponds to the high-frequency signal on graph  $G^{t_j}$.
>
> The example above explains that the informative representation $\mu$ on a graph snapshot might be noisy signal in another graph snapshot. From this view of point, the basis of low-frequency signals will be evolving, what is the accurate definition of low-frequency signals in the evolving graph?
>
> I highly recommend the authors to make a further study about what DFT can essentially learn from the positional embeddings on evolving graphs. I deeply believe that this can largely help the readers to understand how the proposed positional encoding works theoretically.

---

> > ### Author Response · Authors · 2024-11-30
> > **Further Reply from Authors for Q2**
> >
> > Thanks very much for your appreciation of most of our responses!
> >
> > We are more than willing to further address your remaining two concerns!
> >
> > > Discussion on Q2
> >
> > **Firstly, please allow us to clarify the definition of positional encoding**.
> >
> > We understand your example, but we need to point out that **the eigenvector $v_i$ is not the positional encoding of node $i$** in our modeling.
> >
> > Instead, according to lines 112 to 113, the positional encoding of node $i$ consists of all the $i$-th entries from eigenvectors $v_1, \ldots v_n$.
> >
> > Therefore, the positional encoding vector of a node is not a pure low-frequency and high-frequency signal vector. In other words, the case “$\mu = v_0$” you suggest does not exist in our modeling.
> >
> > **Secondly, we respect your concern about the change. Next, we introduce how our modeling deals with the change scenario**.
> >
> > For example, if the $i$-th eigenvector is a low-frequency signal on a temporal snapshot $t-1$, but corresponds to a high-frequency signal on another temporal snapshot $t$, there can be two root causes, and our design can cover them.
> >
> >   - [The change is caused by randomness] To be more specific, as elaborated in our last round response to W3, DFT is applied on the sequence of $u$’s positional encodings at previous timestamps and the learnable filter is designed to filter noises across dimensions in the spectral domain. If the signal changes from high-frequency to low-frequency (or vice versa) over time by randomness, then the learnable filter filters out noise in the spectral domain of the sequence.
> >
> >   - [The change is caused by entity activities] It is also notable that the change pattern may reflect the historical properties of a batch of certain nodes, which is part of evidence for the current time decision making. Because our SimpleTLP updates positional encoding at each timestamp as shown in Eq. 3, this pattern will be recorded into the representation learning process constrained by the loss functions as shown in Eq. 11 and Eq. 12.

---

> ### Comment · Reviewer_Ysht · 2024-11-28
> **Concerns about Q3**
>
> In the revised paper, let us recall Eq. (4) as follows:
>
> $h_u^t = s_u^t + MEAN( s_0, \dots, s_k ) $
>
> where $s_u^t$ denotes the user feature, and $s_i$ denotes the item feature. It seems to me that $h_u^t$ is invariant to the order of item sequence $[s_0,\dots, s_k]$. Even if applying time encoding $t_0$ to $s_0$ (the user interacts with item $s_0$ at time $t_0$), it is not clear to me what kind of sequential patterns $h_u^t$ is able to learn from the data  $[s_0,\dots, s_k]$, since only Mean-Pooling is used.

---

> > ### Author Response · Authors · 2024-11-30
> > **Further Reply from Authors for Q3 (Part I)**
> >
> > > Discussion on Q3
> >
> > To better address your concern, in the following bullet point details, we mainly discuss three aspects:
> > - (1) why the time encoding is important and what information they encode,
> > - (2) why we choose sum pooling, and
> > - (3) why the sum pooling did not hurt the performance.
> >
> > ```
> > (1) Why the time encoding is important and what information they encode
> > ```
> >
> > - Time encoding serves as a kind of positional encoding that encodes the temporal order of happened connection events between entities.
> >
> >   - Intuitively, the heuristic way to understand it is: the difference (between the timestamp event happened and the current timestamp) (1) makes the entries in the sequence distinguishable and (2) makes the sequences distinguishable from each other. Therefore, representation learning can be more informative.
> >
> >   - Mathematically, in Eq. 4, $h^t\_{u, N}$ is designed to summarize the node raw features of $u$’s 1-hop temporal neighborhood with the time information they connected.
> >
> > - Moreover, we designed complicated designs to capture more temporal dependencies for our proposed SimpleTLP, i.e.,  $h^t\_{u, E}$ in Eq. 5 and $h^t\_{u, P}$ in Eq. 8, where the temporal order is sensitive, i.e., changing it will disobey the historical connection pattern and make the output different.
> >
> >   - In general, $h^t\_{u, E}$ and $h^t\_{u, P}$ consist of time encoding of the most $K$ recent interactions {$t_1, \dots t_K$}, so if we replace (or disturb) some interactions in these $K$ interactions with interactions occur at different timestamps then the time encodings would change, resulting in changing $h^t_{u, P}$ and $h^t_{u, E}$.
> >
> >   - For example, for $h^t_{u, E}$ in Eq. 5, the time encoding with edges is aggregated by the learnable weight matrix. That means if we change the connection temporal order, the aggregation will be different.
> >
> >   - For example, for $h^t_{u, P}$ in Eq. 8, changing the time order will affect $u_i$’s positional encoding sequence, resulting in changes in approximated PEs of $u_i$ at time $t$, as shown in Eq. 2.
> >
> > ```
> > (2) Why we choose sum pooling over time encoding
> > ```
> >
> >   - Time information is natural. In the temporal graphs, nodes connect with each other at different timestamps, which means that each edge is associated with a timestamp. Therefore, when we retrieve the historical connection, the time order exists, and we can naturally adopt it as a kind of input to indicate the event being represented by deep learning models, there is no reason for us to discard it or disturb it.
> >
> >   - Taking another kind of source information (i.e., time) into the deep learning models will induce additional computational burden for sure. That's why we design to use the sum pooling in this specific part, i.e., Eq. 4., because we find it has a good balance between the effectiveness and efficiency. The detailed theoretical and empirical reasons are below.

---

> ### Author Response · Authors · 2024-11-30
> **Further Reply from Authors for Q3 (Part II)**
>
> ```
> (3) Why the sum pooling did not hurt the performance
> ```
>
> - **Theoretically**, the reason that we can utilize this design is that **\{$s^{t’}\_{u’}$\} is already constrained in a time interval $[t- t\_{gap}, t]$**.
>
>   - Since the time interval is small enough (i.e., all are recent interactions), any interaction ($u, u’, t’$) would tend to have the equal chance of affecting $u$ at time $t$.
>   - As shown in Table 22 and Table 6, the ratio of $t\_{gap}$ / (number of unique timestamps) for all datasets is at most $0.167$. The interval $[t- t\_{gap}, t]$ would contains at most $t\_{gap}$ distinct timestamps (since the granularity in Table 6 indicates that timestamps are integers). Therefore, with such a small ratio above, the interval $[t - t\_{gap}, t]$ contains timestamps that are very close to $t$.
>
> - **Empirically**, the aggregation manner is model-agnostic, which means we can use other pooling functions. Therefore, we prepared an additional pooling function, i.e., time weighted pooling function, to prove our theoretical guess.
>
>   - Time weighted pooling function is sensitive to the temporal order, i.e., the more recent timestamp will occupy more weights during the aggregation. In this way, if the order changes then the output will be different. It has the following mathematical equation.
>
>   - Suppose we have $M$ distinct timestamps in node $u$’s interactions during the time interval $[t - t_{gap}, t]$ and let’s denote them as $t_1, \dots t_M$ (where $t_1 < \dots < t_M$), then the weight for the node that interacts with $u$ at time $t_i$ would be $e^{-(t - t_i)} / \sum_{j} e^{-(t - t_j)}$. In other words, we obtain weights by applying softmax on $-(t - t_1), \dots, -(t - t_M)$.
>
>   - In this way, Eq. 4 would be $ h^t\_{u, N} = s^t\_u + \sum\_{(u, u’, t’\_j)} \alpha\_{t’_j} s^{t’_j}\_{u’} $, where $(u, u’, t’\_j)$ are interactions involving $u$ during the time interval $[t - t\_{gap}, t]$ and  $\alpha\_{t’_j} = e^{-(t - t’\_j)} / \sum\_{j = 1}^M e^{-(t - t\_j)}$
>
>   - With the above new question replacing Eq. 4, the experiments are shown in the following table, where we can see that sum pooling can achieve a very competitive result, which aligns our theoretical design: in a short time window, the equal contribution (i.e., sum pooling) can be effective and efficient. Moreover, handicraft design for the aggregation weight is not alway efficient and suitable for all cases.
>
> ---
>
> Table. Performance of Different Pooling Function on Eq. 4.
> | Dataset |  | Transductive AP | Transductive AUC | Inductive AP | Inductive AUC |
> | -- | -- | -- | -- | -- | -- |
> | Enron | Mean Pooling | 93.96 $\pm$ 0.36	| **95.98 $\pm$ 0.23** |	**89.17 $\pm$ 0.88** |	**92.30 $\pm$ 0.57** |
> | | Weighted Sum Pooling | **94.02 $\pm$ 0.19** | 95.95 $\pm$ 0.13 | 88.71 $\pm$ 0.39 | 92.08 $\pm$ 0.29 |

---

> ### Author Response · Authors · 2024-12-02
> **Gentle Reminder from Authors for the Discussion Due**
>
> Dear Reviewer Ysht,
>
> We want to express our sincere thanks for your time and review!
>
> We enjoyed the discussion with you, and we are very grateful for your appreciation of our first-round answer addressing most of your concern.
>
> Beyond that, for your two further questions, we took them seriously and prepared the detailed theoretical analysis and empirical demonstration.
>
> Since the author-reviewer discussion will end in a couple of hours, we kindly remind you and gently ask if you are satisfied with the second-round answer or need more discussions for your re-evaluation.
>
> We are more than willing to try our best to answer any further questions until the last minute.
>
> Thanks very much!
>
> Authors

---

### Author Response · Authors · 2024-11-23
**General Response from Authors**

Dear reviewers,

We want to sincerely and wholeheartedly thank you for your review, answering your questions and addressing your concerns improves the quality of our paper to a large extent.

In the rebuttal,

- We prepared the detailed answer for each of your raised questions and concerns in below boxes

- We finished the required extra experiments, and the results further improves the effectiveness of our proposed method

- For the required change of the paper, we uploaded the new version and marked the updates by orange color, e.g.,
  - New baseline theoretical analysis and empirical comparison is added to Appendix C.
  - New hyperparameter analysis experiments and analysis is added to Appendix E.5.3, E.5.4, and E.5.5.

- After obtaining your appreciation and acknowledgement of our rebuttal answer, we promise to add them to the paper, and we promise to release the code upon the publication.


Thanks again for your review!

Authors

---

### Meta-Review · Area_Chair_9DwW · 2024-12-20

**Metareview:**

The paper proposes an algorothm for link prediction on temporal graphs. The authors novel contribution lies in devising a lightweight method for learnable positional embeddings through discrete fourier transform and then applying an MLP to encode these embeddings. While the reviewers appreciate the fresh approach towards link prediction on temporal graphs, they maintain that some concerns remain unaddressed despite the rebuttal (there was post-rebuttal discussion where we revisisted and discussed the concerns). Particularly, the definition of low frequencies on temporally evolving graphs and the capacity of learning the sequential patterns (not the temporal patterns) remains ambigious despite the rebuttal. Hence, the reviewers are not in favor of acceptance in its current form.

**Additional Comments On Reviewer Discussion:**

The paper underwent extensive discussion during the rebuttal phase and in the post-rebuttal phase. Reveiwer Ysht maintains that some concerns remain unaddressed despite the rebuttal (summarize in meta-review). Other reviewers did not come forward to champion acceptance.

---

### Decision · Program_Chairs · 2025-01-22

Reject